# Phosphorylation-driven epichaperome assembly is a regulator of cellular adaptability and proliferation

The intricate network of protein-chaperone interactions is crucial for maintaining cellular function. Recent discoveries have unveiled the existence of specialized chaperone assemblies, known as epichaperomes, which serve as scaffolding platforms that orchestrate the reconfiguration of protein-protein interaction networks, thereby enhancing cellular adaptability and proliferation. This study explores the structural and regulatory aspects of epichaperomes, with a particular focus on the role of post-translational modifications (PTMs) in their formation and function. A key finding is the identification of specific PTMs on HSP90, particularly at residues Ser226 and Ser255 within an intrinsically disordered region, as critical determinants of epichaperome assembly. Our data demonstrate that phosphorylation of these serine residues enhances HSP90's interactions with other chaperones and co-chaperones, creating a microenvironment conducive to epichaperome formation. Moreover, we establish a direct link between epichaperome function and cellular physiology, particularly in contexts where robust proliferation and adaptive behavior are essential, such as in cancer and pluripotent stem cell maintenance. These findings not only provide mechanistic insights but also hold promise for the development of novel therapeutic strategies targeting chaperone assemblies in diseases characterized by epichaperome dysregulation, thereby bridging the gap between fundamental research and precision medicine.

Conventional wisdom, as crystallized in Beadle and Tatum's 1941 paradigm of "one gene–one enzyme–one function," has traditionally delineated targets as outcomes of protein expression changes or point mutations within proteins. However, it is increasingly apparent that protein dysfunctions in the context of many disorders, including cancer, neurodegenerative disorders, among others, are predominantly shaped by changes in interaction strengths and cellular mislocalization. These factors, in turn, can be modulated by variations in post-translational modifications (PTMs), stabilization of disease-associated protein conformations, and other protein-modifying mechanisms[1,2]. Within this complex context, Heat Shock Protein 90 (HSP90) emerges as a compelling exemplar, transcending the boundaries of conventional understanding[3].

Positioned as a versatile chaperone, often referred to as the guardian of the proteome, HSP90 assumes a pivotal task in the realm of maintaining cellular equilibrium by facilitating protein folding, stabilization, and degradation[4]. Under the canonical folding paradigm, HSP90 functions as a homodimer. Each protomer is composed of an N-terminal domain (NTD), a middle domain (MD), and a C-terminal dimerization domain (CTD)[4,5]. The NTD contains a nucleotide-binding pocket, where ATP binding and hydrolysis take place[6]. The chaperone cycle of HSP90 is coupled to a series of dynamic conformational changes accompanying its ATPase cycle. Beginning with NTD/MD and MD/CTD interdomain rotations and cross-monomer dimerization[7], HSP90 transitions from open to closed conformational states, while folding client proteins[8,9]. HSP70 and HOP (HSP70–HSP90 organizing

✉e-mail: chiosisg@mskcc.org; feixia.chu@unh.edu

protein) bring client proteins to HSP90 and form the loading complex[10]. Other co-chaperones participate at different stages of the HSP90 chaperone cycle and regulate its conformational changes along the chaperone and ATPase cycle[4]. Co-chaperones may have different preferences for client proteins, fine-tuning subcellular pools of HSP90 to mitigate stressors and maintain proteostasis[11]. These assemblies are further shaped by PTMs in HSP90, co-chaperones and client proteins[12]. Overall, the highly orchestrated interactions among these proteins—both chaperones and clients—are transient in the chaperone cycle under physiological conditions.

While this classical understanding portrays HSP90 as a dimeric entity that interacts dynamically with co-chaperones and client proteins, research has uncovered a spectrum of multimeric HSP90 forms, each sculpted by the cellular milieu and the presence of stress-inducing factors[3]. These multimers, whether homo-oligomeric or hetero-oligomeric, expand HSP90's functional repertoire, blurring the boundaries between traditional chaperone functions and newfound roles as holdases or scaffold proteins. In disease contexts, such as cancer and neurodegenerative disorders, HSP90's conformational adaptability gives rise to epichaperomes—distinctive hetero-oligomeric formations of tightly bound chaperone, co-chaperones and other factors[13–15]. This phenomenon goes beyond mere biochemical curiosity; it represents a fundamental mechanism by which cells respond to stressors, whether of genetic, proteotoxic or environmental nature[3,16–18]. Unlike chaperones which help proteins fold or assemble, epichaperomes exert a maladaptive influence, reshaping the assembly and connectivity of proteins pivotal for sustaining pathological traits. For example, in cancer, epichaperomes take on scaffolding functions not found in normal cells, altering the assembly and connectivity of proteins important for maintaining a malignant phenotype and enhancing their activity, which provides a survival advantage to cancer cells and tumor-supporting cells[13,19]. In Alzheimer's disease epichaperomes rewire the connectivity of, and thus negatively impact, proteins integral for synaptic plasticity, brain energetics, and immune response[15].

The revelation of HSP90's maladaptive multimeric epichaperomes has also profound implications for therapeutic interventions, including in the treatment of diverse disease states including cancers and of neurodegenerative disorders. Rather than a blanket inhibition of all HSP90 pools, targeting specific pathologic conformations of HSP90 as found in epichaperomes while sparing normal HSP90 functions holds the promise of enhancing the safety as well as the immunostimulatory and anticancer effects of HSP90 inhibitors[3].

Despite these important mechanistic and therapeutic implications, key factors facilitating HSP90 incorporation into epichaperomes—namely, the conformations that enable epichaperome formation and the structural elements that support the enrichment of such conformations—remain unknown. In this study, we use a combination of chemical biology, unbiased mass spectrometry techniques, and molecular dynamics simulations to elucidate the conformation of HSP90 populated in epichaperomes and to characterize the structural and molecular factors that support and favor the enrichment of such conformation, and in turn, the formation of epichaperome assemblies. Beyond these structural revelations, our findings demonstrate how these factors directly influence cellular behaviors, particularly in contexts where robust proliferation and adaptation are crucial, such as cancer and stem cell maintenance. This direct link between epichaperome function and fundamental cellular processes has translational relevance for therapeutic development.

## Results

### Pluripotent stem cells and cancer cells share epichaperomes

Epichaperomes nucleated through enhanced interactions between HSP90 and HSP70, namely the heat shock cognate 70 (HSC70) isoform, are a distinct feature of cancer cells[13,19]. Epichaperomes containing HSP90 are detected in induced pluripotent stem cells (iPSCs)[20], in leukemia stem cells[21,22], and in glioma cancer stem cells (CSCs)[23]. Hyperactivation of the transcription factor c-MYC required in generating iPSCs[24], maintaining embryonic stem cells (ESCs)[25] and CSCs[26], is also a driving factor in epichaperome formation in tumors, irrespective of the tumor type[13,27]. Notably, these epichaperomes are all sensitive to and can be disrupted by small molecules such as PU-H71 (zelavespib) or PU-AD (icapamespib) that bind to HSP90[13,23,28], suggesting that a similar composition, facilitated by a specific conformation of HSP90, may characterize epichaperomes in these distinct cellular contexts.

To test this hypothesis, we initially explored the composition of epichaperomes in selected cellular models, encompassing pluripotent stem cells and cancer cells. For pluripotent stem cells, we examined two mouse ESCs (E14 and ZHBTc4) and a human induced pluripotent cell line (hiPSC). Additionally, two cancer cell lines, well-characterized in terms of epichaperome composition and function, were chosen as representative epichaperome-positive (MDA-MB-468) and -negative/low (ASPC1) cancer cells (Fig. 1a–f and Supplementary Figs. 1, 2).

In contrast to folding chaperone complexes, which are inherently dynamic and short-lived[6], epichaperomes represent long-lasting hetero-oligomeric assemblies composed of tightly associated chaperones, co-chaperones, and various other factors. HSP90 is a major component found within epichaperomes along with other chaperones, co-chaperones, and scaffolding proteins like HSP70 (especially HSC70), CDC37, AHA1, and HOP[13]. Consequently, when we analyzed cell homogenates containing epichaperomes using native PAGE followed by immunoblotting with antibodies specific to epichaperome constituent chaperones and co-chaperones, we observed a range of high-molecular-weight species, both distinct and indistinct, in addition to the primary band(s) characteristic of chaperones. This observation held true for both pluripotent stem cells and cancer cells (Fig. 1b, Supplementary Fig. 1a–d, and refs. 13,19,20). Notably, HSP90 immunoblotting revealed the presence of species comprising HSP90 in epichaperome assemblies in cancer cells and pluripotent stem cells, in addition to the prominent 242 kDa band, which is a characteristic of non-transformed cells[13,19,29].

Epichaperomes undergo disassembly during iPSC differentiation[20] or when cancer cells are treated with PU-H71 or PU-AD[15,23,28,30]. Therefore, next we induced the differentiation of the pluripotent stem cells under investigation. In the ZHBTc4 cell line, Oct4 expression is controlled by a Tet (tetracycline)-off *oct4* regulatory system[31]. Downregulation of Oct4 in ZHBTc4 cells has been reported to induce trophoblast differentiation, which is characterized by changes in cell morphology, specifically, cells flattening into epithelial-like cells, and is associated with slower growth[32]. Mouse embryonic E14 stem cells undergo spontaneous differentiation into embryoid bodies (EB) when cultured in suspension without antidifferentiation factors such as leukemia inhibitory factor (LIF)[33] and iPSCs differentiate into mature dopaminergic neurons using a floor plate-based differentiation protocol[34]. We confirmed that the differentiation of these pluripotent stem cells was correlated with the disassembly of epichaperomes, as observed through native PAGE immunoblotting. This disassembly is evident by a reduction in high-molecular-weight chaperone species on native PAGE observed when immunoblotting for epichaperome constituent chaperones (see HSP90α/β, HOP, HSC70, CDC37, AHA1, HSP110 in Fig. 1b and Supplementary Fig. 1), with minimal changes observed in total chaperone levels on SDS–PAGE. Notably, for HSP90, a decrease in bands other than those in the ~242 kDa range was observed upon differentiation, supportive of epichaperome disassembly (see HSP90 immunoblotting).

PU-H71 serves as an epichaperome probe that, in contrast to the tested antibodies which indiscriminately detect epichaperomes and other HSP90 pools, exhibits a preference for HSP90 when it is integrated into epichaperomes[13]. Labeled derivatives of PU-H71 can,

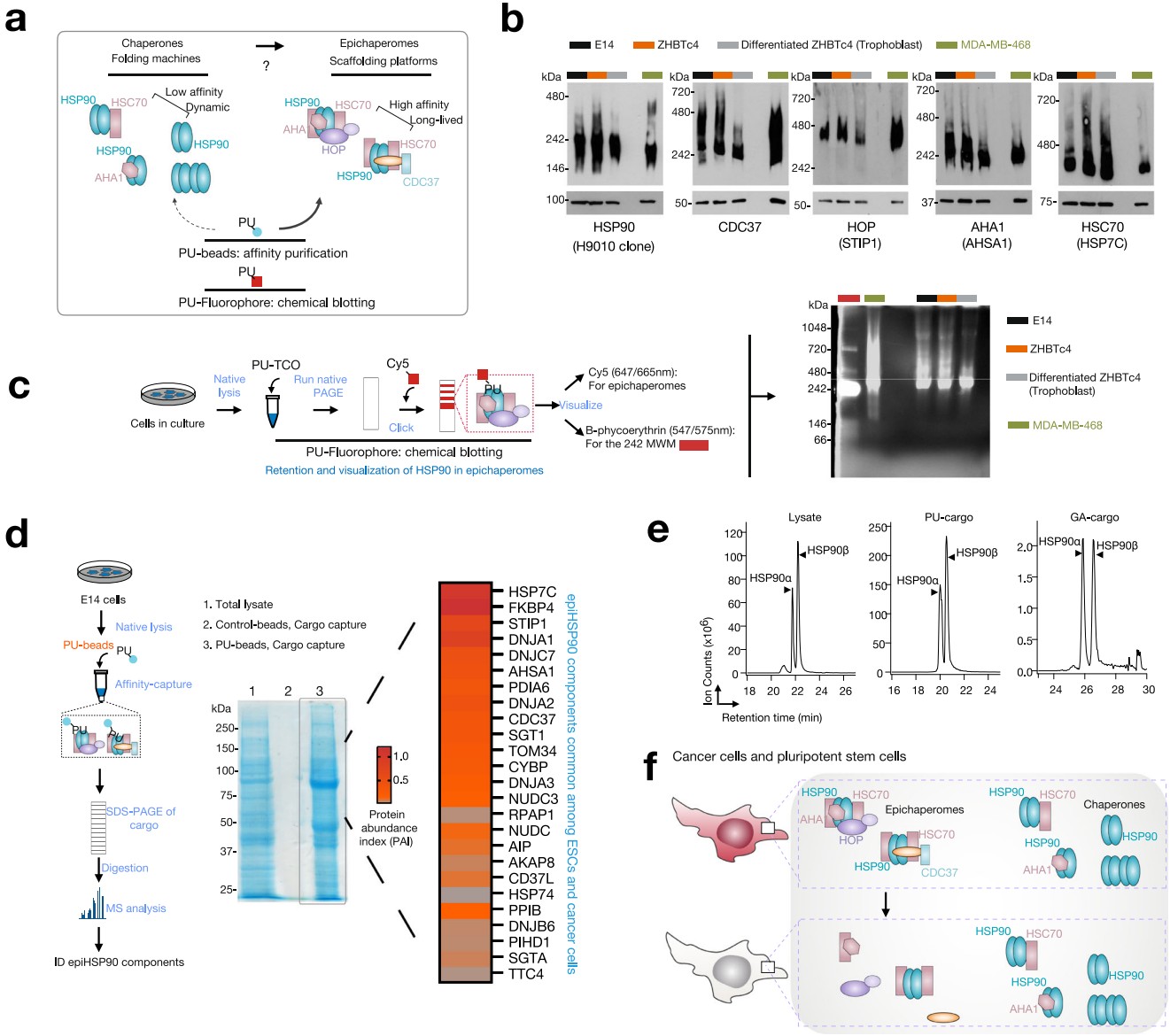

**Fig. 1 | Embryonic stem cells and cancer cells share compositionally similar epichaperomes. a** Schematic illustrating the biochemical and functional distinctions between epichaperomes, defined as long-lasting hetero-oligomeric assemblies composed of tightly associated chaperones and co-chaperones, and traditional chaperones. Unlike chaperones, which assist in protein folding or assembly, epichaperomes sequester proteins, reshaping protein–protein interactions, and consequently altering cellular phenotypes. The schematic also outlines key principles for the use of PU-probes—PU-beads and PU-clicked to a fluorophore such as cy5—in epichaperome analysis. **b** Detection of epichaperome components (chaperones and co-chaperones) through SDS–PAGE (bottom, total protein levels) and native PAGE (top), followed by immunoblotting. See also Supplementary Fig. 1.

**c** Visualization of HSP90 in epichaperomes using the PU-TCO probe clicked to cy5. See also Supplementary Fig. 2. Gel images are representative of three independent experiments. **d** Epichaperome constituent chaperones and co-chaperones identified through mass spectrometry analyses of PU-beads cargo. Representative data of two independent experiments. See Supplementary Fig. 3 for the GA cargo. HSP7C is HSC70, STIP1 is HOP, and AHSA1 is AHA1. **e** Illustration of an isobaric, discriminant peptide pair from ESC lysate samples and HSP90 captured by PU- and GA-beads. Representative data of two independent experiments. **f** Schematic summary. Both cancer cells and pluripotent stem cells harbor epichaperomes. These epichaperomes undergo disassembly during differentiation processes. Source data are provided in Supplementary Data 1 and as a Source data file.

therefore, be employed to detect HSP90 within epichaperomes, distinguishing it from other HSP90 pools (as illustrated in Fig. 1c and Supplementary Fig. 2a−c). To achieve this, we generated lysates from ZHBTc4, E14 cells, and MDA-MB-468 cells under conditions that preserve native protein assemblies. Subsequently, we labeled these homogenates with a clickable PU-probe (PU-TCO, refs. 19,35). After running these labeled samples on native PAGE gels, we conjugated the PU-probe with a Cy5 dye and visualized epichaperomes, confirming the presence of epichaperomes in both the ESCs and the cancer cells. These epichaperomes were characterized by multimers

observed at and above ~300 kDa (Fig. 1c). Moreover, the labeling of epichaperomes by the PU-probe decreased upon ESC differentiation, supportive of epichaperome disassembly (Fig. 1c and Supplementary Fig. 2b).

Additionally, we conducted labeling experiments using live E14 ESCs, instead of homogenates, employing a PU-CW800 probe (a derivative of PU-H71 conjugated with an 800 nm near-infrared dye) or a control derivative (an inactive PU-derivative that does not interact with epichaperomes) (see Supplementary Note 1). The most responsive target of the PU-probes, but not the control probe, was an HSP90

assembly of ~300 kDa, thus above the major 242 kDa band preferred by the anti-HSP90 antibody. This species was detected on native PAGE in PU-probe treated cells but not in control treated cells (Supplementary Fig. 2c).

In summary, the predominant HSP90 band characteristic of epichaperomes is a 300 kDa assembly, distinctly differing from the typical ~242 kDa band observed in non-transformed cells[13,19,32] when analyzed on native PAGE gels. Mass spectrometric (MS) analysis of the ~300 kDa assembly confirmed the presence of HSP90 and HSC70 as the primary protein components of this multimeric complex (Supplementary Data 1, 300 kDa LC–MS). This finding aligns with the composition of core epichaperome complexes previously reported in cancer cells[13]. Consequently, these findings combined confirm that both cancer cells and pluripotent stem cells share HSP90 and HSC70 as integral constituents of their core epichaperomes.

To gain further insights into epichaperome assemblies, we employed resin-based affinity purification experiments. Specifically, we utilized resins with immobilized PU-H71, referred to as PU-beads, and an inert control molecule on control beads, following established procedures[13] (Fig. 1d). As an additional control, we employed a resin containing immobilized geldanamycin (GA), known for its ability to bind and isolate predominantly un-complexed HSP90 (GA-beads, Supplementary Fig. 3 and ref. 36). Subsequently, we subjected the protein cargo isolated by these probes to unbiased MS analysis. To precisely determine the protein components of the cargo, we conducted in-gel digestion of the entire gel lanes and employed liquid chromatography/mass spectrometry (LC–MS/MS) in conjunction with the semi-quantitative spectra-counting method[37,38] for the identification and quantification of cargo proteins (Supplementary Data 1).

We observed that the cargo isolated by PU-beads from ESCs contained 26 of the 42 major chaperone and co-chaperones identified prior in cancer cells[13] as being epichaperome components (Fig. 1d). The identity of all components identified in ESCs is found in Supplementary Data 1. The interaction between PU-beads and epichaperomes was specific towards PU-H71, because control resins did not purify noticeable protein complexes. Similarly, GA-beads precipitated HSP90 but few co-purifying proteins and epichaperome components (Supplementary Fig. 3 and Supplementary Data 1) consistent with previous results that GA isolates largely an un-complexed HSP90[39].

In mammalian cells, HSP90 exists in two paralogs, HSP90α and HSP90β[40], both of which have been reported to play roles in epichaperome formation in cancer cells[13]. To assess the isoform composition of HSP90 within epichaperomes, we exploited the subtle difference between one pair of isobaric peptides, namely 88Thr-Lys100 in HSP90α and 83Thr-Lys95 in HSP90β, where a single amino acid distinguishes them (Ile in HSP90α and Leu in HSP90β) (Supplementary Fig. 4a). The assignment of HSP90 isoforms relied on co-eluting peptides obtained from the isobaric peptide present in purified HSP90β (Supplementary Fig. 4b, c). Extracted ion chromatograms of the peptide mass revealed an ~1.5 $\beta/\alpha$ ratio in the ESC lysate and the cargo isolated by PU-beads (Fig. 1e), while the GA-beads cargo exhibited an ~1.0 $\beta/\alpha$ ratio. Similar findings were obtained through spectra counting, with the HSP90β/HSP90α ratio determined using spectral counting consistent with ratios obtained through MS intensity calculations (Supplementary Data 1: 708/540 = 1.31 for the PU-beads cargo; 219/235 = 0.93 for the GA-beads cargo). This validation of spectra counting as an effective semi-quantitative method supports the conclusion that epichaperomes isolated from ESCs exhibit a predominantly unbiased HSP90 paralog composition, akin to what has been reported for cancer cells[13].

In summary, the wealth of complementary biochemical experiments presented here lends strong support to the idea that both cancer cells and pluripotent stem cells harbor epichaperomes that are compositionally similar. Notably, HSP90 and HSC70 emerge as major constituents of the core epichaperome structure, serving as a scaffold for recruiting various co-chaperones to create specific epichaperome assemblies. This shared architectural similarity between epichaperomes in ESCs and cancer cells underscores the existence of a common epichaperome-enabling HSP90 conformer that is enriched in both biological contexts.

## Epichaperome-enabling conformation of HSP90

MS identification of cross-linked residues that are in spatial proximity but not necessarily close in primary sequence, provides valuable distance restraints that can be employed for computational modeling of proteins and protein complexes[41–43]. Therefore, to determine the conformation of HSP90 in epichaperomes, we used a chemical cross-linking and mass spectrometry (CX–MS) approach to identify and quantify cross-linked peptides of PU-H71-favored HSP90 pools.

To ensure the capture of the epichaperome-enabling conformation, we first cross-linked cellular lysates using the amine-reactive cross-linker disuccinimidyl suberate (DSS) prior to HSP90 capture on the PU-beads[13,36] (Fig. 2a). DSS crosslinking stabilizes the conformation of proteins by covalently linking residues that are in close proximity, effectively "freezing" their relative positions. While crosslinking could potentially affect key residues and the binding of the assembly to the chemical inhibitor-attached resin, the DSS cross-linker primarily targets solvent-accessible surface lysine residues, minimizing the likelihood of introducing extensive conformational changes or directly perturbing the binding pocket on HSP90. Given PU-H71's preference for binding HSP90 in its epichaperome conformation, any significant alteration of HSP90's structure by DSS would likely reduce PU-H71's binding affinity. Therefore, the structure captured by PU-beads is more likely to reflect the native HSP90 conformation found in epichaperomes rather than any altered state. By applying DSS before introducing PU-H71, the experimental setup increases the likelihood that the observed conformation is representative of the functional epichaperome, prior to any potential conformational changes or epichaperome disassembly induced by PU[15,19,28]. We used SDS–PAGE to separate proteins after crosslinking and capturing HSP90 with the beads, specifically analyzing the major ~80–90 kDa band that corresponds to the HSP90 monomer (Fig. 2a). In addition, the cross-linked peptides identified were predominantly intra-monomeric, as they fit within the expected spatial constraints of the DSS cross-linker[43].

Parallel experiments were conducted using GA-beads, corresponding to solid-support immobilized GA, as a control[13,36]. The identity of cross-linked HSP90 peptides purified by PU- or GA-beads pull-down can be found in Supplementary Data 2. Notably, the alpha carbon distances between all cross-linked residues, as identified with high confidence, fell below the maximal span of DSS (30 Å). This suggests that proteins retained their native states without significant conformational perturbations during the cross-linking process.

We calculated the cross-linking percentage for each pair of cross-linked PU- or GA-bound HSP90 residues. This calculation involved normalizing the MS ion intensity of cross-linked peptides by the sum of all cross-linked peptides and cross-linker-modified peptides containing the cross-linked residues. By doing so, we could mitigate the impact of variations in the reactivity of cross-linked residues, allowing us to primarily assess the influence of the distance between cross-linked residues and their local secondary structures[44].

Most cross-linked pairs from both PU- and GA-bound samples exhibited similar cross-linking percentages, with data points evenly distributed around a trend line with a slope of 1 (dotted line, Fig. 2b). This observation suggests a broad similarity in secondary and tertiary structures between these HSP90 populations. However, clear differences emerged, revealing conformational distinctions between the PU- and GA-favored HSP90 subpopulations (highlighted by orange circles, Fig. 2b).

Notably, residues Lys58–Lys112 in HSP90α and Lys53–Lys107 in HSP90β, situated within the ligand-binding pocket, displayed a higher

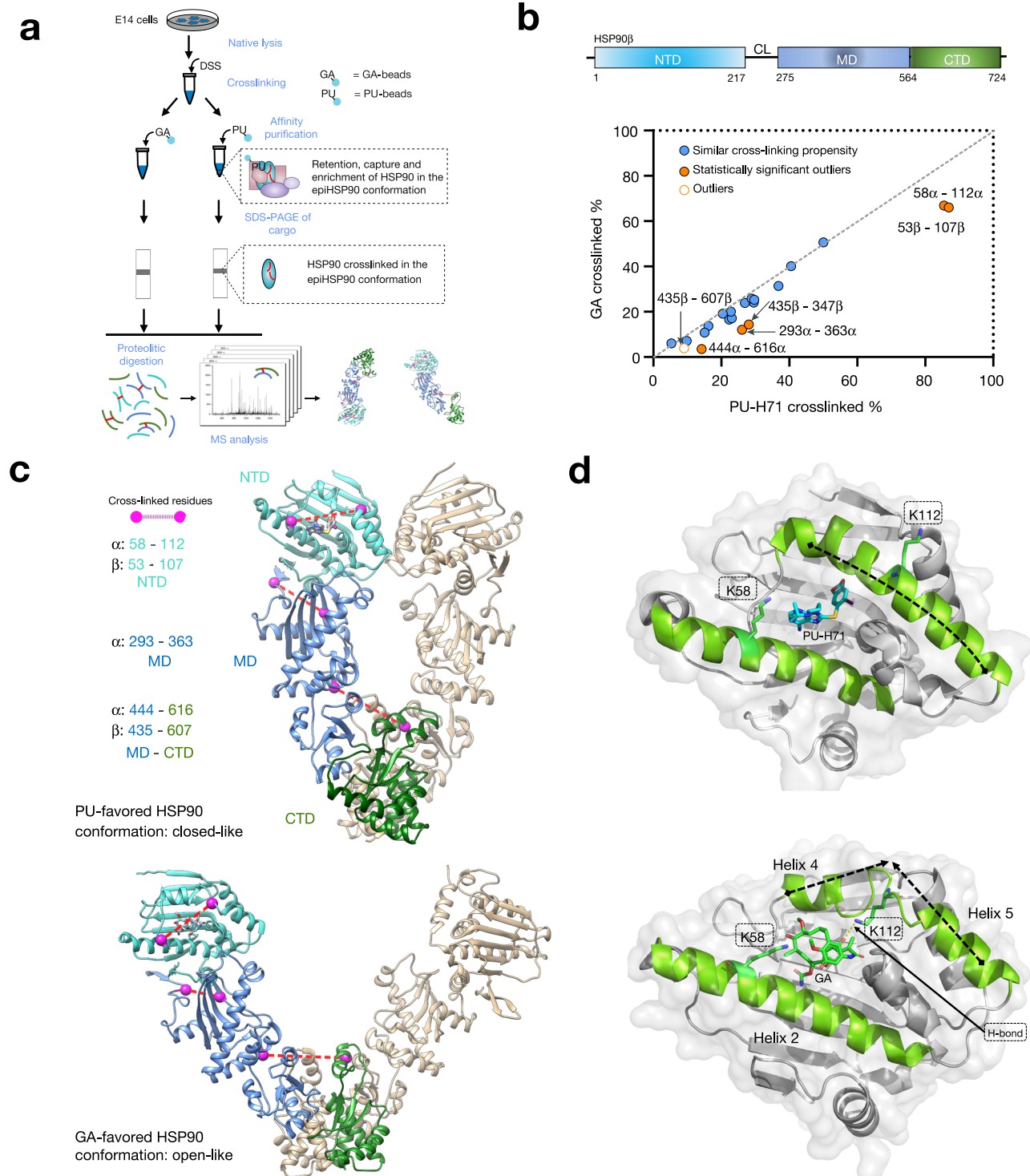

**Fig. 2 | An enrichment of the closed-like conformation of HSP90 favors epichaperomes formation. a** Experiment outline. DSS disuccinimidyl suberate crosslinker. **b** Plot comparing cross-linking propensity of Lys residues in HSP90 bound to PU-H71 or geldanamycin (GA). Average cross-linking percentage of PU-H71 (*x*-axis) and GA (*y*-axis) bound HSP90 cross-linked pairs are shown. Pairs with similar cross-linking propensity are shown along the dotted line with a slope of 1. Outlier cross-linked peptides are those with cross-linked Lys residues eight amino acids away and a cross-linking percentage difference ≥1.5 standard deviations of replicates. Statistically significant outliers ($p \leq 0.05$) were determined by two-sample *t*-test with equal variances, $n = 3$ replicate measurements. **c** Homology model illustrating the HSP90 dimer in the open conformation (template PDB: 2IOQ), favored by GA, and the closed conformation (template PDB: 2CG9), favored by PU-H71. One HSP90 protomer is colored to indicate the N-terminal domain (NTD), the middle domain (MD), and the C-terminal domain (CTD). Cross-linked residues are shown as dashed lines between labeled residues. **d** NTD structures of PU-H71 (top, PDB: 2FWZ) and GA (bottom, PDB: 1YET)-bound HSP90. Source data are provided in Supplementary Data 2.

cross-linking percentage in PU-bound HSP90 populations compared to their GA-bound counterparts (Fig. 2b). This observation aligns with distinct pocket configurations preferred by each ligand, as previously observed through X-ray crystallography[45–49]. Specifically, crystal structures show the bulkier GA binds more superficially, causing helices 4 and 5 (Fig. 2d) to move away from the nucleotide-binding site, thereby preventing full closure of the ATP lid. Moreover, the side-chain amino functional group of Lys112 forms a hydrogen bond with a benzoquinone oxygen of GA. This pocket configuration aligns with the reduced cross-linking activity of the lysine pair mentioned above. Conversely, PU-H71 binds deeply within the pocket. In this configuration, helices 4 and 5 are packed against helix 2 with Lys112 and Lys58 in HSP90α (or Lys107 and Lys53 in HSP90β) positioned more favorably for cross-linking. This arrangement of lysine residues is more likely to be found in the closed conformation of HSP90 (Fig. 2c), as proposed by crystallographic studies (PDB: 2CG9)[50].

It is essential to reiterate that the cross-linking experiments were conducted to "lock" HSP90 conformations with covalent bonds before resin-based affinity purification experiments using the PU- or GA-beads. Consequently, the X-ray structures of PU- or GA-bound HSP90 NTD closely reflect a preferred pocket configuration that each ligand may capture in the cell, and in this case, for PU-H71, it is indicative of the pocket configuration of HSP90 in the epichaperomes.

Furthermore, differences in HSP90 conformation were corroborated by cross-linked pairs located at the interfaces between NTD/MT (HSP90α: Lys293–Lys363) and MD/CTD (HSP90α: Lys444–Lys616; HSP90β: Lys435–Lys607) (Fig. 2b). These interfaces undergo significant reorientation during the HSP90 conformational cycle, implying a distinct HSP90 conformation favored by PU-H71 compared to GA. Lys444 in HSP90α (Lys435 in HSP90β) and Lys616 in HSP90α (Lys607 in HSP90β) are positioned either within the middle of the MD or in proximity to the central axis of the HSP90 homodimer (Fig. 2c). The distance between these lysine residues can provide insights into the relative placement of the monomer arms in specific HSP90 conformations (e.g., 20 Å in closed-like conformations; 29 Å in open-like conformations). The lower cross-linking percentage observed for Lys444 and Lys616 in HSP90α (Lys435 and Lys607 in HSP90β) in GA-favored HSP90 suggests a longer distance (29 Å) between them, supporting GA's preference for binding to an open-like conformation. In contrast, the moderate cross-linking percentage detected for these residues in PU-H71-favored HSP90 implies a medium distance (20 Å) between them, favoring a closed-like conformation enriched in epichaperomes (Fig. 2c).

Additionally, a third pair of cross-linked residues (Lys293 and Lys363 in HSP90α) supports this notion. Located near the interface between the NTD and the MD, their positions are sensitive to the ligand-binding state of the NTD, leading to changes in the relative positioning of secondary structures near the NTD/MD interface and altering the distance between Lys293α and Lys363α. Consistent with the cross-linked pair at MD/CTD interface, a closed-like conformation (16 Å) in PU-H71 bound HSP90 will be more amenable than an open-like conformation (13 Å) in GA-bound since the short distance might have limited the location of side chains for cross-linking reactions.

In summary, our CX−MS data, supported by several cross-linked residue pairs situated in structurally distinct regions, the nucleotide-binding pocket, and the NTD/MD and MD/CTD interfaces, shed light on the conformation adopted by HSP90 within epichaperomes. These findings underscore the notion that an enrichment of the closed-like conformation of HSP90 in specific cellular environments favors the formation of epichaperomes.

## Specific PTMs support HSP90 incorporation into epichaperomes

To uncover the factors that facilitate the enrichment of the epichaperome-favoring HSP90 conformation, we conducted a comprehensive examination of the HSP90 pools isolated by PU-H71 and GA, searching for potential differences. Notably, we identified several peptides phosphorylated on Ser231 and Ser263 in HSP90α (Ser226 and Ser255 in HSP90β) exclusively in the PU-H71 cargo from ESCs (Fig. 3a, b and Supplementary Data 3). High-quality MS/MS spectra (illustrated for Ser226 and Ser255 phosphopeptides in HSP90β, Fig. 3b) coupled with precise mass accuracy allowed for the unequivocal identification of the peptide sequences and the phosphorylation sites. In contrast, these phosphorylated peptides were notably absent in substantial quantities in the GA cargo (Supplementary Data 3).

Subsequently, we performed label-free quantitation of these phosphopeptides using ion intensity measurements and observed a significant enrichment in the PU-beads cargo, particularly in the case of Ser255 of HSP90β. For instance, the Ser255 phosphopeptide displayed a nearly threefold enrichment in the PU-H71 cargo compared to the lysate, after protein loading normalization using a representative tryptic peptide (Fig. 3c).

To gain further insights, we leveraged previously reported MS datasets of PU-H71-isolated cargo from epichaperome-positive cancer cells[13,19], including MDA-MB-468 (triple negative breast cancer), Daudi (Burkitt's lymphoma), IBL-1 (AIDS-related immunoblastic lymphoma), and NCI-H1975 (non-small cell lung carcinoma), as well as from non-transformed (NT) proliferating cells in culture (e.g., MRC5, lung fibroblast and HMEC, mammary epithelial cells) (Supplementary Data 4). This analysis revealed that phosphorylation of these serine residues is also enriched in cancer cells when compared to NT cells (Ca:NT S255 = 16; S226 = 8; S263 = 12, Fig. 3d) establishing it as a hallmark of both ESC and cancer epichaperomes. This observation further supports the idea of a shared structural and architectural foundation for epichaperomes among ESCs and cancer cells.

As HSP90 is found alongside HSC70 in epichaperomes, we conducted an additional confirmatory experiment. Here, we used YK5-B, a biotinylated probe that binds to HSC70 in epichaperomes, and thus captures HSP90 in epichaperomes via HSC70[19]. PU-H71 and YK5-B probes were used to isolate cargo from epichaperome-positive cancer cells, including MDA-MB-468 and OCI-Ly1 (breast cancer and diffuse large B-cell lymphoma, respectively), as well as from CCD-18Co colon cells in culture (i.e., non-transformed proliferating cells in culture) (Supplementary Data 4). We found that the Ser255 and S226 phosphopeptides of HSP90β were nearly four to five times more abundant in epichaperome-positive cancer cells compared to non-transformed proliferating cells in culture, for both the PU-cargo and the YK5-B cargo. Similar enrichment was noted for Ser263 and Ser231 in HSP90α (Fig. 3e). This analysis, thus, using both PU-H71 and YK5-B probes across diverse cell types, underscores the robustness of our observations and reinforces the role of phosphorylation in the acidic linker in shaping HSP90 within epichaperomes.

In light of these findings, made with two distinct probes and observed in ESCs, five cancer cell lines, each representative of a distinct cancer type, and of three non-transformed, but proliferating, cells in culture, it is evident that the epichaperome-specific agents target a subpopulation of HSP90 characterized by high phosphorylation levels in the acidic linker between the NTD and the MD, and this subpopulation predominantly assumes a closed-like conformation. In conjunction with PU's preference for HSP90 within epichaperomes, and substantiated by YK5-B, a probe that binds epichaperomes via HSC70, these results strongly indicate that phosphorylation at these two serine residues is a key driver for HSP90 incorporation into epichaperomes and, consequently, for epichaperome formation.

## Specific PTMs drive epichaperome formation and function

To explore whether the phosphorylation of these serine residues plays a pivotal role in driving, rather than merely resulting from, epichaperome formation, we next studied the phosphomimetic

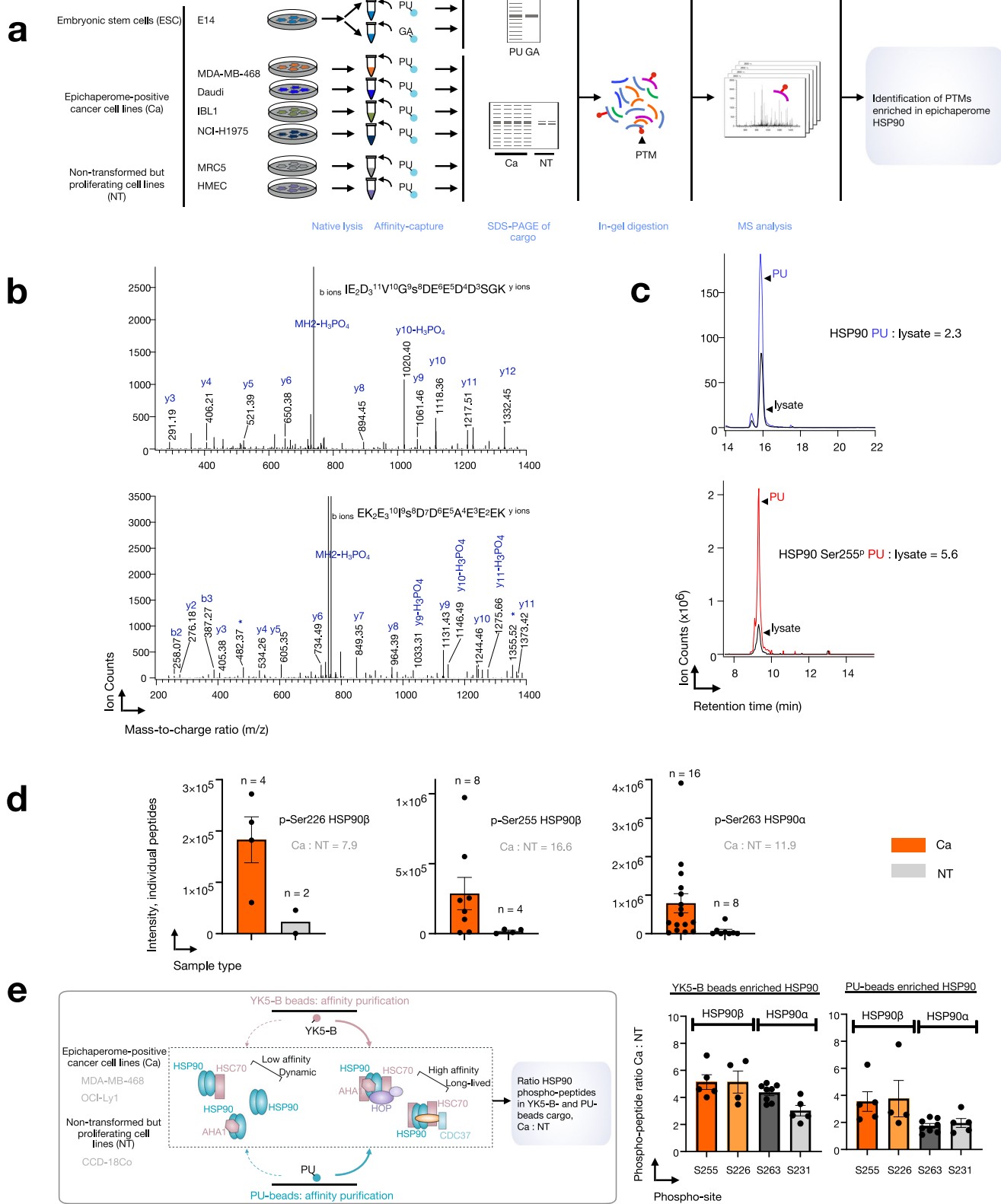

**Fig. 3 | Phosphorylation of key residues located in the charged linker supports HSP90 incorporation into epichaperomes. a** Experiment outline and expected outcomes. **b** Tandem mass spectrometry (MS) spectra of HSP90 Ser226 (bottom) and Ser255 (top) phosphorylated peptides are presented, supporting the sequence and phosphorylation site identification. **c** Comparison of the extracted ion chromatogram of HSP90 Ser255 phosphopeptide in the PU-bead cargo and ESC lysate (bottom) with a representative unmodified tryptic peptide in the PU-bead cargo and ESC lysate (top). **d** Ion intensity values of all identified phosphopeptides and the ratio of mean peptide intensity for each phosphosite in the samples described in (**a**) (i.e., $n = 4$ Ca and $n = 2$ NT cell lines). Each data point represents an individual phosphopeptide, and data are presented as mean ± s.e.m. to illustrate variability between peptides across the cell lines. **e** Ratio of individual peptide intensity for each phosphosite in the samples described in the schematic (graph: mean ± s.e.m., S255 $n = 5$; S226 $n = 4$; S263 $n = 8$; S231 $n = 5$). Source data are provided as a Source Data file and in Supplementary Data 3 and 4.

(HSP90β$^{S226E,S255E}$) and the non-phosphorylatable (HSP90$^{S226A,S255A}$) mutants.

Notably, these serine residues are located within an intrinsically disordered region (IDR) of HSP90 (Supplementary Fig. 5). IDRs are pivotal elements in the intricate network of protein–protein interactions (PPIs). These regions lack a fixed three-dimensional structure, granting them exceptional flexibility. This structural adaptability enables proteins containing IDRs to assume various conformations in response to specific cellular contexts or binding partners. Such adaptability plays a crucial role in facilitating context-dependent involvement in distinct PPIs. In the case of HSP90, these serine residues within the IDR may alter the dynamics and structure of the charged linker, contributing to stabilizing the epichaperome-enabling conformation of this chaperone, and in turn facilitating epichaperome formation.

To explore this hypothesis, we conducted computational analyses to investigate the impact of each mutation on the flexibility of the charged linker (Fig. 4a–c and Supplementary Figs. 6, 7). We constructed a model of the putative epichaperome core—namely the ~300 kDa assembly, see Fig. 1—based on the cryo-EM structure of a multimeric HSP90 assembly (PDB: 7KW7). This structure represented 2xHSP90α, protomer A and B, bound to 2xHSP70 and 1xHOP (i.e., HSP70(A)–HSP90(A)–HSP90(B)–HSP70(B)–HOP). To create the model, we substituted HSP90 with human HSP90β using the closed-state cryo-EM structure (PDB: 8EOB). Additionally, we computationally inserted the charged linker, which was missing in the cryo-EM structures (Fig. 4a).

We conducted all-atom molecular dynamics simulation of this pentameric protein assembly, with each system containing all the components along with either the EE, AA, or WT HSP90—in both protomers. These simulations are intended to qualitatively explore the immediate response of the assembly to the perturbation induced by mutations and not to provide an extensive characterization of the assemblies' dynamics. By using a comparative MD-based approach we explore how short-term changes in the structural dynamics of different components within a large assembly may influence the emergence of states relevant for assembly stabilization. The underlying premise is that nanosecond timescale residue fluctuations in regions specifically responsive to certain states may facilitate large-scale rearrangements that underlie functional changes.

These simulations revealed that the structure and conformation of the charged linker were sensitive to the phosphorylation of the serine residues. In the pentameric assembly containing the phosphomimetic EE mutant (i.e., HSP90$^{S226E/S255E}$), the linker of HSP90 protomer A (i.e., HSP90(A)), had a high probability of forming a β-strand bordering the Ser226Glu residue (2.1% of β-strand A). This strand remained stable over the duration of the simulation. This β-strand's formation significantly decreased in the pentameric assembly containing the wild-type (WT, i.e., HSP90$^{S226/S255}$) protein (0.4% of β-strand A), with no secondary structure element (SSE) found in the assembly containing the AA (i.e., HSP90$^{S226A/S255A}$) mutant (Fig. 4b). Notably, ATP binding, but not ADP binding, favored a charged linker with a high content of β-strand A formation (2.1% vs 0.3%, respectively, in the EE mutant) (Fig. 4b and Supplementary Fig. 6a). This finding emphasizes that the observed changes in the EE mutant were not merely due to the addition of charged residues; they were intricately tied to the phosphorylation status and the specific context, including the nucleotide environment permissive of the specific HSP90 conformation (i.e., closed-like). Intriguingly, the strategic formation of β-strand A not only stabilized the charged linker but also induced a conformational switch, flipping it into an up conformation, thereby fully exposing the MD of HSP90(A), where HSP70(A) binds (Fig. 4c, see HSP90 protomer A–HSP70(A) interface). While other SSEs were observed in the analyzed assemblies containing either the WT or the mutant HSP90s, no other had a similar conformational effect on the charged linker as we observed for the β-strand A (see the effect of α-helices 1 through 6 in

Supplementary Fig. 6a, b). Intriguingly, the behavior observed for the charged linker of protomer A (HSP90(A)) was not mirrored in protomer B (HSP90(B)). In the assembly, the presence of HSP70(B) and HOP results in stabilizing intermolecular hydrogen-bond interactions with the charged linker of HSP90(B). These interactions effectively lock the linker into a specific conformation, thereby limiting its potential for SSE formation and conformational rearrangement compared to protomer A (Supplementary Fig. 7).

We conducted dynamical residue cross-correlation analyses to explore how different protein units or subdomains in the pentameric HSP70(A)–HSP90(A)–HSP90(B)–HSP70(B)–HOP assemblies, featuring either the WT (HSP90$^{S226/S255}$) or mutant (HSP90$^{S226E/S255E}$ or HSP90$^{S226A/S255A}$) HSP90s, correlate in their motions throughout the simulation (Fig. 5a, b). This analysis aimed to reveal how individual components move in relation to each other. Positive dynamical cross-correlations spanning different components of the assembly within the large epichaperome core may indicate enhanced cooperative motions, suggesting increased interactions that contribute to the stability of the assembled structure. Previous studies have employed similar analyses to investigate how ligand-induced modulations influence the overall flexibility of HSP90 assemblies, facilitating progress along the chaperone cycle, thereby supporting feasibility of this approach[51].

Indeed, we observed the highest correlation among the components in assemblies containing the HSP90 EE phosphomimetic, mimicking the case where the charged linker is phosphorylated, followed by the WT, and then the non-phosphorylatable HSP90 AA mutant (Fig. 5a). The AA mutant does not fully mimic the WT in MD simulations because substituting serine with alanine alters interaction capabilities and structural dynamics. This substitution is commonly used to create non-phosphorylatable mutants because alanine's small, non-polar side chain minimizes steric hindrance and structural alteration. However, alanine lacks the hydroxyl group needed for phosphorylation and hydrogen bonding, affecting local structural dynamics and protein conformation. Thus, while the AA mutant serves as a useful non-phosphorylated baseline, it may not fully replicate the WT's behavior. Comparing WT, EE, and AA mutants, however, provides insights into how phosphorylation modulates HSP90β dynamics and epichaperome assembly. Notably, the coordinated movements observed in the assemblies containing the HSP90 phosphomimetic strongly support the idea that the HSP70(A)–HSP90(A)–HSP90(B)–HSP70(B) or HSP70(A)–HSP90(A)–HSP90(B)–HSP70(B)–HOP assemblies can be preferentially stabilized when the HSP90 charged linker is phosphorylated (Fig. 5b). This observation aligns with the prominent ~300 kDa band observed for the epichaperome core in native PAGE (see Fig. 1 showing HSP90 assemblies favored by PU-H71).

In contrast, in the WT HSP90 assembly, coordinated movements were primarily observed between the two HSP90 protomers, within HSP90, and between HSP90 and HSP70 and HOP, specifically through HSP90 protomer B (Fig. 5a, b). These movements are more consistent and favorable in the context of HSP90(A)–HSP90(B)–HSP70(B) or HSP90(A)–HSP90(B)–HOP assemblies (Fig. 5b). This observation implies that the major, broad ~242 kDa band detected by the HSP90 antibody—representing the primary HSP90-containing assembly observed in differentiated ESCs (Fig. 1) and in non-transformed cells[13–15,17,20]—may consist of such assemblies, along with HSP90 homo-oligomers.

As the correlation in movement is only one part of the bigger picture regarding dynamics, we next calculated the root mean square fluctuation (RMSF) and principal component analysis (PCA) to provide more quantitative information on the amplitude of fluctuations among the different assemblies[52,53].

RMSF offers a detailed view of the flexibility of individual residues by measuring the average deviation of each residue from its mean position over the simulation period. To gain insights into the dynamics

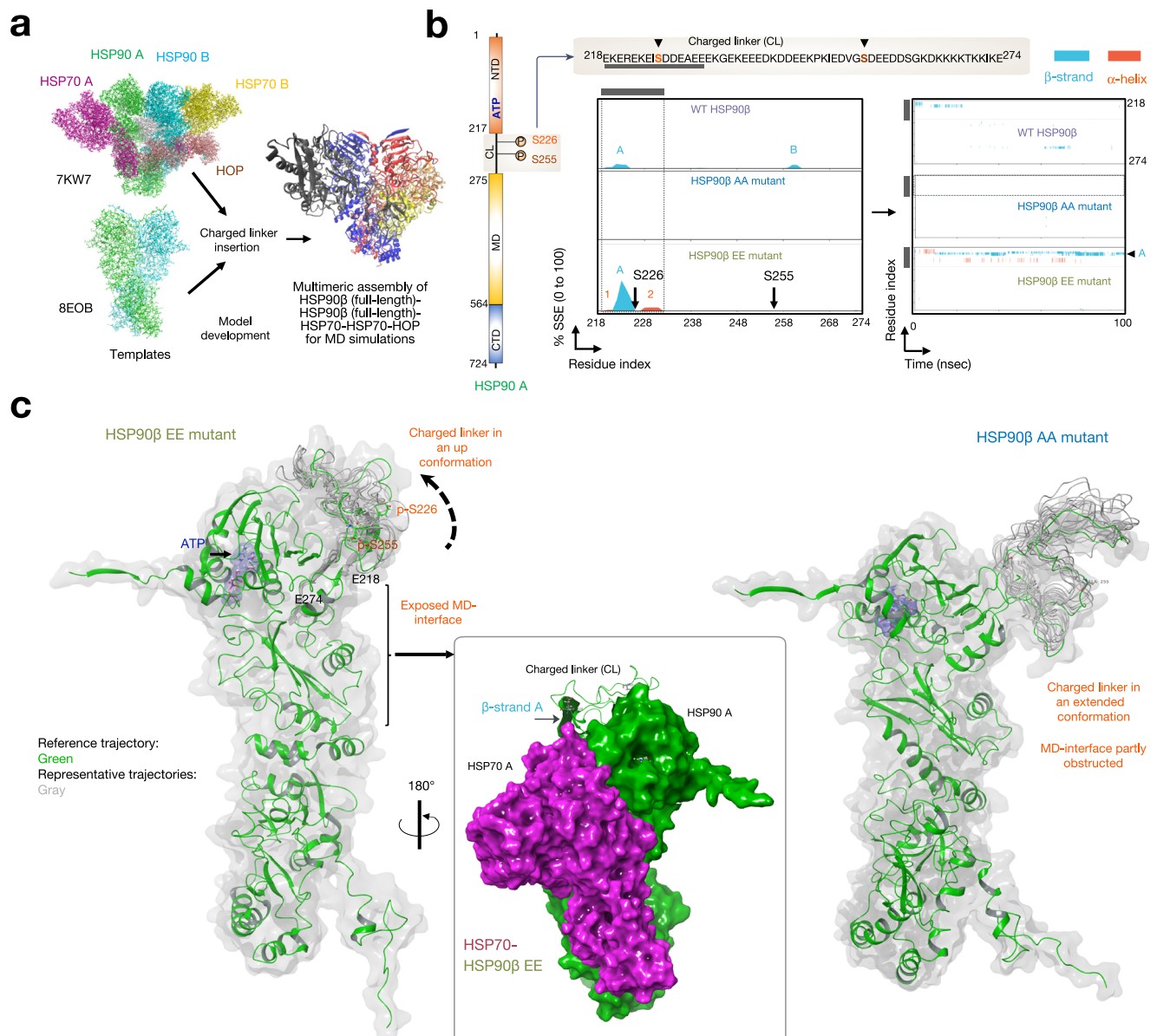

**Fig. 4 | Phosphorylation of key residues located in the charged linker of HSP90 leads to a conformational shift in the linker, exposing the middle domain of the protein. a** Model of the HSP90–HSP90–HSP70–HSP70–HOP assembly used for the molecular dynamics simulations. A and B, protomers A and B, respectively. **b** Protein secondary structure elements (SSE) like alpha-helices and beta-strands of the charged linker of protomer A of ATP-bound HSP90 monitored throughout the molecular dynamics simulation. WT (HSP90$^{S226/255}$), phosphomimetic (HSP90$^{S226E/S255E}$), and non-phosphorylatable (HSP90$^{S226A/S255A}$) mutants were analyzed. Each pentameric assembly was simulated three times for 100 ns, yielding similar results across simulations. The plot on the left reports SSE distribution by residue index throughout the charged linker and the plot on the right monitors each residue and its SSE assignment

over time. Schematic illustrating the primary structure of the full-length HSP90 with color-coded domains is also shown: NTD N-terminal domain; MD middle domain and CTD C-terminal domain. The charged linker (CL) and the location of the two key serine residues are also shown (top inset). The gray bar indicates the CL segment encompassing residues 218–232. **c** Cartoon representation of ATP-bound HSP90 protomer A in assemblies with either the phosphomimetic (HSP90$^{S226E/S255E}$) or non-phosphorylatable (HSP90$^{S226A/S255A}$) mutants. The figure depicts the reference trajectory and representative trajectories from $n = 1000$ simulation frames. The inset illustrates the surfaces available for the interaction between HSP90 A and HSP70 A when the CL is in the up conformation. The arrow indicates the location of the key beta-strand in the charged linker. See also Supplementary Figs. 5–9.

of specific components within the pentameric assemblies, we calculated the RMSF as an average of all residues for each protein component, allowing us to pinpoint regions (or components) within the assemblies that exhibit higher flexibility or rigidity (Fig. 5c and Supplementary Fig. 8).

In our ATP-bound assembly analysis, we observed that there was no significant difference in fluctuation at the regions encompassing the HSP90(A)–HSP90(B) components between the WT and phosphomimetic (EE) assemblies (WT: 1.79 ± 1.24 Å vs EE: 1.77 ± 1.32 Å) (Supplementary Fig. 8). This indicates that phosphorylation does not

significantly alter the core's flexibility. However, notable differences were detected in the regions encompassing binding to HSP70(A) and HSP70(B) (Fig. 5c). Specifically, for HSP70(A), the WT assembly showed higher flexibility (2.32 ± 0.94 Å) compared to the EE assembly (2.02 ± 0.90 Å). Similarly, in the region binding to HSP70(B), the WT assembly exhibited more fluctuation (2.57 ± 1.53 Å) than the EE assembly (2.38 ± 0.90 Å). The AA mutant had a slightly destabilizing effect on the HSP90(A)–HSP90(B) core (AA: 1.97 ± 1.22 Å vs WT: 1.79 ± 1.24 Å). Importantly, in the ADP-bound assembly, phosphorylation—as in the EE mutant—had a destabilizing effect on the HSP90 core

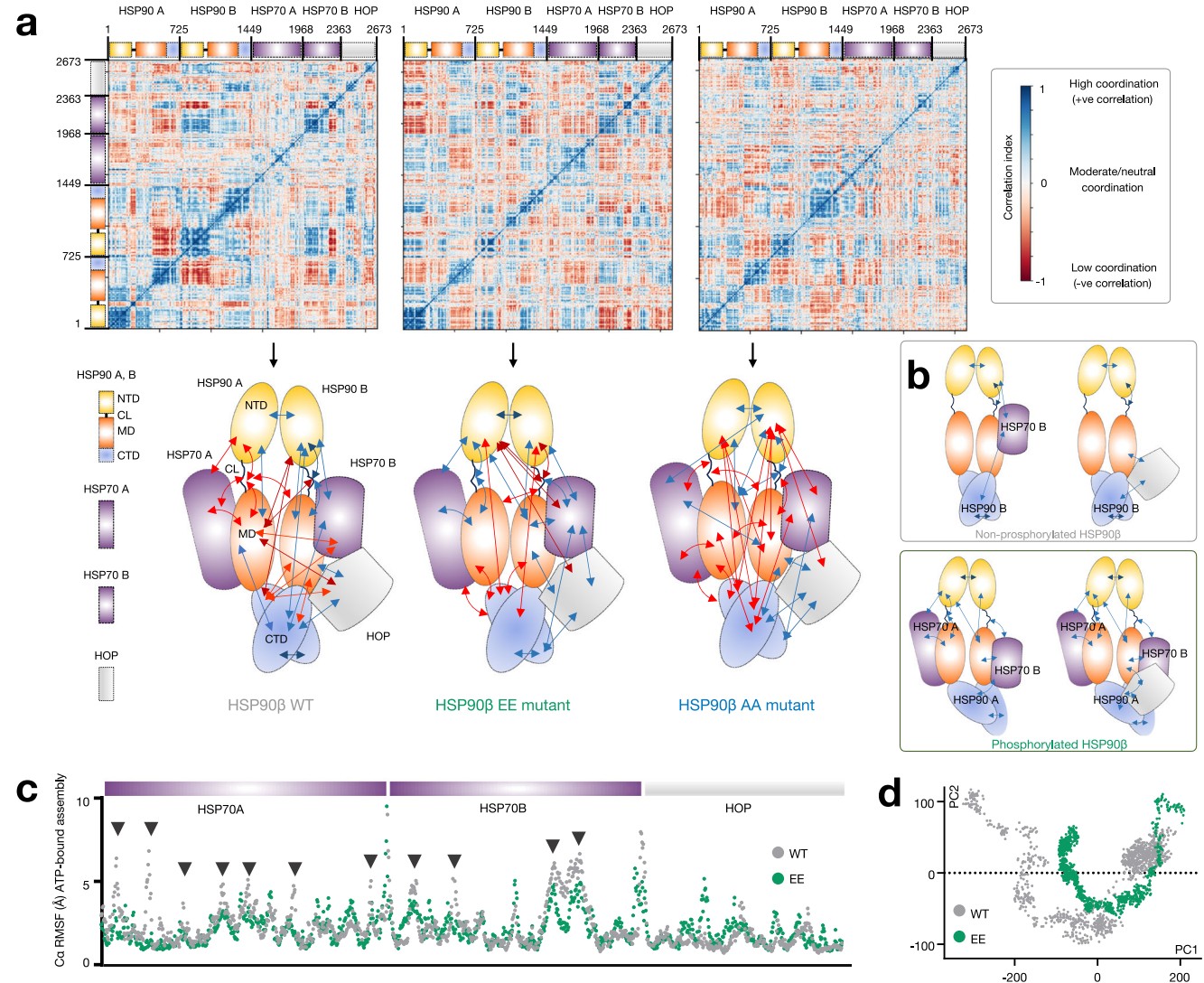

**Fig. 5 | Phosphorylation of key residues located in the charged linker of HSP90 facilitates assembly motions conducive to epichaperome core formation.**
**a** Dynamic cross-correlation matrix of Cα atoms for 100 ns molecular dynamics simulations of ATP-bound assemblies containing WT (HSP90$^{S226/S255}$), phosphomimetic (HSP90$^{S226E/S255E}$) or non-phosphorylatable (HSP90$^{S226A/S255A}$) HSP90. Correlated and anti-correlated motions are shown in the matrix and represented in the cartoon. The color of the arrows in the cartoon corresponds to the colors shown in the correlation index bar, with darker blue indicating stronger co-movement (positive correlation) and darker red indicating stronger opposite movement (negative correlation). The assembly contains two full-length HSP90β proteins (protomer A and protomer B). The two HSP70 proteins (HSP70 A and HSP70 B) and the HOP protein are of sizes reported, and as per the constructs used in 7KW7. **b** Cartoon showing assemblies that are preferentially formed when the HSP90 charged linker is either phosphorylated (as in the EE mutant) or not phosphorylated (as in the WT protein). **c** The plot depicts the root mean square fluctuation

(RMSF) values for each residue within the ATP-bound HSP90 assemblies across different conditions. Each point along the *x*-axis corresponds to a specific residue in the protein sequence of HSP70A, HSP70B, and HOP. The *y*-axis represents the RMSF value in angstroms (Å), indicating the average flexibility of each residue. Higher RMSF values suggest greater flexibility, while lower values indicate rigidity. Arrowheads highlight areas where the structural dynamics diverge significantly. See Supplementary Fig. 8 for the full assembly. **d** The plot depicts the combined global coordinated motions of all Cα atoms in ATP-bound assemblies within the PC1 and PC2 component space, representing the major directions of variance in the simulations. Each point corresponds to a frame in the simulation, illustrating the assembly's conformational state. Different sub-spaces for WT and EE mutants have been merged here for comparison. **a**–**d** Each condition was simulated three times with similar results. Source data are provided in Supplementary Data 5 and as a Source data file.

(WT: 1.76 ± 1.33 Å vs EE: 1.95 ± 1.33 Å), supporting the notion that phosphorylation does not favor the open-like conformation of HSP90. RMSF data supporting these analyses are found in Supplementary Data 5.

Thus, the RMSF data reveal that phosphorylation has a significant impact on the flexibility and stability of the pentameric assembly, particularly at the HSP70 binding sites. While the core region between HSP90 protomers A and B shows no significant difference in flexibility between WT and EE assemblies, indicating stable core dynamics, the regions interacting with HSP70(A) and HSP70(B) demonstrate notable changes. At the HSP70(A) binding site, the EE assembly shows reduced

flexibility compared to the WT. This reduced flexibility suggests that phosphorylation stabilizes the interaction of HSP70(A) with other assembly components, enhancing cooperative stability. This stabilization aligns with the increased correlated motions observed in the simulations, indicating that phosphorylation promotes tighter and more coordinated interactions at this site. Similarly, at the HSP70(B) binding site, the EE assembly exhibits reduced fluctuations, suggesting tighter binding. Thus, protomer B's interactions with HSP70(B) and HOP impose intrinsic rigidity, which is further stabilized by phosphorylation of the charged linker, resulting in reduced flexibility and increased cooperative stability.

PCA is used to reduce the complexity of the data by identifying the main modes of movement (principal components, PCs) in the protein's trajectory. By analyzing the movement of the Cα atoms, one can understand how different regions or entire proteins in the assembly move relative to each other over time. Our PCA analysis revealed distinct differences between the WT and EE mutant HSP90 assemblies (Fig. 5d). These movements are captured along the first two PCs (PC1 and PC2), which represent the major directions of movement in the protein structure during the simulation.

In the ATP-bound state, we observed that the span of PC1 is significantly larger in the WT HSP90-containing assemblies compared to the EE HSP90 mutant assemblies. PC1 captures the direction of the greatest variance in the data, correlating with the most substantial conformational changes in the protein assembly. The larger PC1 span in the WT suggests that this assembly experiences greater conformational flexibility, allowing it to adopt a broader range of structural conformations.

PC2 captures the second most significant direction of variance, orthogonal to PC1. Although the differences in PC2 are not as pronounced as in PC1, the larger PC2 span in the WT indicates additional modes of flexibility or movement. This suggests that the WT assembly not only explores more conformational space but also possesses more varied structural dynamics, which are essential for its function in protein folding. On the other hand, the EE mutant's narrower spans suggest more restricted dynamics, indicating that phosphorylation of the charged linker enhances structural stability, contributing to a robust epichaperome assembly.

To further support the stability differences observed through RMSF and PCA analyses, we examined the potential energy of the assemblies (Supplementary Fig. 9a, b). In the ATP-bound state, the EE-containing assembly exhibited the most favorable potential energy (i.e., the largest negative value), indicating greater stability compared to the WT and AA assemblies. While potential energy values can be approximations and should not be viewed as definitive evidence on their own, they are consistent with the observed reduced flexibility and tighter binding in the EE assembly. This stabilizing effect was specific to the ATP-bound state and was not observed in the ADP-bound state.

In summary, both MS evidence and computational models converge to support the conclusion that phosphorylation of the charged linker is a crucial contributor to epichaperome assembly, emphasizing its role in shaping not only HSP90, but also the stability and dynamics of the epichaperome structure. These findings highlight how phosphorylation modulates the structural dynamics and functional roles of HSP90 within the epichaperome assembly, promoting a scaffolding function through enhanced stability and reduced flexibility at critical interaction sites. Asymmetry and stabilization through intermolecular interactions lead to distinct dynamic behaviors for the charged linkers of protomers A and B. While protomer A's linker remains more flexible and capable of rearranging into different conformations—modulated by its phosphorylation status—protomer B's linker is constrained by its interaction with HSP70 and HOP. Despite these distinct local impacts of phosphorylation on the structure and conformation of individual linkers, both linkers contribute to modulating the overall stability and dynamics of the pentameric assembly. This dynamic interplay highlights the complex regulatory mechanisms at play within the epichaperome, illustrating how each protomer's interaction with its partners can influence the system's overall behavior and stability.

Next, we carried out an extensive biochemical and functional analysis to reinforce these findings. Given the well-established tight association between HSP90 and other chaperones and co-chaperones in epichaperomes[13,19,20,54], our focus shifted to a comprehensive evaluation of chaperone and co-chaperone proteins co-purified with the phosphomimetic (HSP90β$^{S226E,S255E}$) and non-phosphorylatable (HSP90$^{S226A,S255A}$) mutants. Our strategy involved the purification of

protein complexes containing N-terminally mCherry-tagged HSP90β in ESCs while retaining the endogenous WT HSP90 proteins. Distinctly labeled ESCs (i.e., labeled with heavy or light isotope lysine and arginine) expressing either the phosphomimetic or non-phosphorylatable mutant were subjected to immunoprecipitation (IP), followed by SDS–PAGE separation and quantitative analysis via MS to determine protein abundance (Supplementary Fig. 10a–d and Supplementary Data 6). It is worth noting that we performed IP separately for the phosphomimetic and non-phosphorylatable mutants to minimize subunit exchange during IP[55], thereby enhancing our ability to detect changes in co-chaperone binding more accurately than previous studies[56].

We found co-chaperones were among the most abundant co-purifying proteins, and most co-chaperones reported to participate in epichaperome formation[13,19] displayed prominent changes in the phosphomimetic mutant (Supplementary Fig. 10a–d). The increased presence of epichaperome-specific co-chaperones (such as AHA1 and FKBP4)[13] in phosphomimetic complexes compared to non-phosphorylatable complexes highlights a stronger association with Ser226$^P$/Ser255$^P$ HSP90 as opposed to the non-phosphorylatable protein. However, we observed a slight reduction in the levels of HSC70 and HOP within phosphomimetic complexes. This decrease is potentially associated with specific subpopulations of HSP90 complexes that become more prevalent when the non-phosphorylatable Ala mutant is overexpressed in cells.

While the phosphomimetic mutant (EE) increases the formation of epichaperomes, the non-phosphorylatable mutant (AA), which as we show in Fig. 6c–e can incorporate into the endogenous non-tagged HSP90 assemblies, is potentially altering the cellular composition of folding chaperone assemblies. This alteration could lead to a higher prevalence of assemblies involving HSC70 and HOP distinct from epichaperomes. Since the immunopurification experiment reports on a ratio of EE to AA, as captured by the antibody, the AA component, which may contain the non-epichaperome HSP90–HSC70 or HSP90–HOP assemblies, will skew the ratio to make it appear that there is less HSC70 and HOP in the EE component (i.e., in the epichaperomes). In fact, this is not true, as the apparent reduction is due to the presence of other HSP90 assemblies that incorporate these chaperones. Therefore, the observed reduction in HSC70 and HOP in the phosphomimetic's immunopurification does not necessarily contradict the computational findings, but rather highlights the complexity of HSP90's interactions within the cell, where multiple forms and assemblies coexist, each with distinct roles and interactions. The introduction of two Ala residues in the unstructured linker region of HSP90 may prompt the recruitment of HSC70 and HOP, chaperones recognized for their ability to bind unstructured unfolded protein stretches[57]. It is important to note that these assemblies are distinct from epichaperomes. Due to the anti-mCherry antibody capturing the entirety of the tagged HSP90, differentiation between specifically epichaperome-related HSP90 and a mixture of epichaperomes and other pools becomes challenging.

To address these limitations, we adopted a multi-pronged approach. First, we utilized immunoblotting with native cognate antibodies for chaperone assemblies retained on native PAGE, coupled with chemical blotting using PU-probes. Additionally, we employed affinity capture with PU-probes to quantify the amount of epichaperome components under each condition (Fig. 6a). For these experiments, we transfected cells with the phosphomimetic (HSP90β$^{S226E,S255E}$, EE mutant) and with the non-phosphorylatable (HSP90$^{S226A,S255A}$, AA mutant) mutants, as well as with HSP90β WT or mCherry tag only for control purposes. In this study, we chose human embryonic HEK293 cells as our cell model since they exhibit intermediate epichaperome expression levels (i.e., medium expressor, Supplementary Fig. 11), making them suitable for studying epichaperome dependence. We confirmed comparable transfection

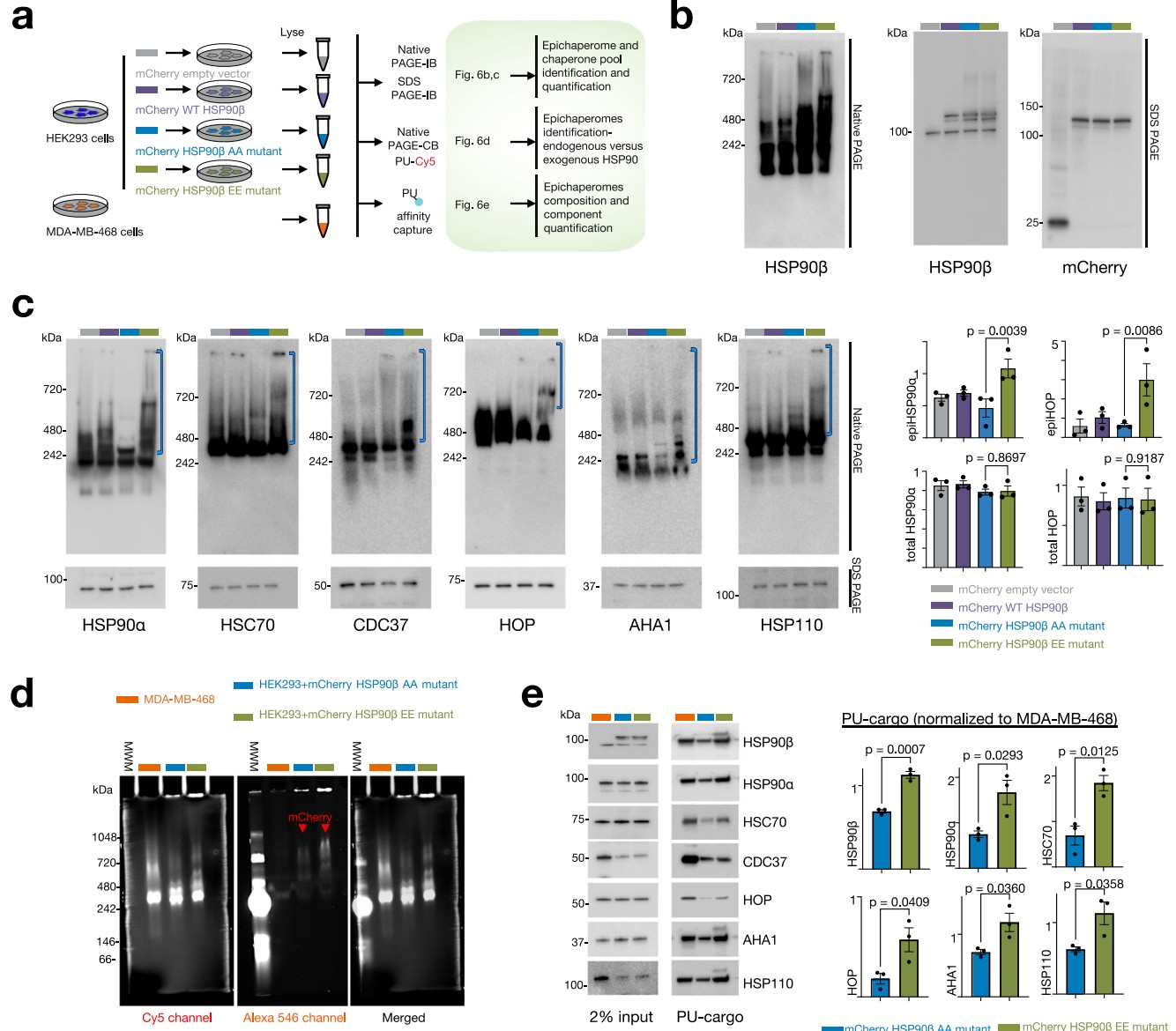

**Fig. 6 | Phosphorylation of key residues located in the charged linker supports HSP90 incorporation into epichaperomes. a** Overview of the experimental design and expected outcomes. **b** Analysis of transfection efficacy in cells transfected with HSP90β mutants, as indicated in (**a**). **c** Detection of epichaperome components (chaperones and co-chaperones) through SDS−PAGE (bottom, total protein levels) and native PAGE (top), followed by immunoblotting. Brackets indicate the approximate position of epichaperome-incorporated chaperones. Data are presented as mean ± s.e.m., *n* = 3, one-way ANOVA with Sidak's post-hoc, EE vs AA. **d** Visualization of HSP90 in epichaperomes using the PU-TCO probe clicked to Cy5 (left) and the mCherry tag (middle). Right, merged images. MWM molecular weight marker. **e** Detection and quantification of epichaperome components through PU-beads capture as indicated in (**a**). Protein amount loaded for input represents 2% of the protein amount incubated with the beads. Data are presented as mean ± s.e.m., *n* = 3, unpaired two-tailed *t*-test. Gel images are representative of three independent experiments. Source data are provided as a Source data file.

efficiency for each construct, with the tagged HSP90β protein expressed in addition to the endogenous HSP90β (Fig. 6b).

Our findings revealed that cells expressing the EE mutant exhibited higher levels of epichaperomes compared to those expressing the AA mutant, as evidenced by immunoblotting of various epichaperome components (including HSP90α, HSC70, CDC37, AHA1, HOP, and HSP110) (Fig. 6c, native PAGE) and chemical blotting with the PU-Cy5 epichaperome probe (Fig. 6d). Notably, there was no significant change in the overall concentration of these proteins in association with their incorporation into epichaperomes (Fig. 6c, SDS−PAGE).

Epichaperome isolation using PU-beads as an affinity purification probe also revealed significantly greater incorporation of chaperones, including mCherry-HSP90β, and co-chaperones into epichaperomes in

cells expressing the EE mutant compared to those containing the AA mutant HSP90 (Fig. 6e), with no substantial alterations observed in cells containing the control vectors (Supplementary Fig. 12a). In contrast, overexpression of wild-type HSP90 in HEK293 cells had a minimal impact on endogenous epichaperomes (Fig. 6c, native PAGE and Supplementary Fig. 12a, PU-beads capture). This observation aligns with previous reports[13] suggesting that factors beyond chaperone concentration play a pivotal role in driving HSP90 incorporation into epichaperomes. Notably, cargo isolated on the control probe (control beads, Supplementary Fig. 12b) showed no detection of HSP90.

Supporting the hypothesis that phosphorylation of HSP90 shifts its role from a folding function (as in chaperone complexes) to a scaffolding role (as in epichaperomes), we found that the refolding

activity was impaired in cell lysates containing the phosphomimetic EE mutant (Supplementary Fig. 13). We conducted an experiment using denatured luciferase as a substrate to assess the refolding capabilities of different HSP90 mutants present in HEK293 cell lysates. We prepared cell extracts from HEK293 cells transfected with cherry-HSP90β constructs, specifically the WT, AA (non-phosphorylatable), and EE (phosphomimetic) mutants. Denatured luciferase was mixed with equal amounts of these lysates to determine whether the distinct HSP90 species in each lysate could facilitate the refolding of luciferase. Denatured luciferase was chosen as the substrate because its spontaneous refolding is inefficient, and because its refolding is HSP90 dependent[58], providing a sensitive assay to measure the chaperone activity of HSP90. MDA-MB-468 cells and CCD-18Co cells were selected as control cell lines with endogenously high- and low-epichaperome levels, respectively.

Lysates containing WT and AA HSP90 had effective protein-folding activity (Supplementary Fig. 13). The WT lysates regained ~50% of luciferase activity, while the AA lysates regained 31.6% of activity after 60 min, demonstrating that these forms of HSP90 can support the refolding process, similar to the CCD-18Co lysates, which regained 44% of luciferase activity. However, lysates from cells expressing the EE mutant showed significantly impaired refolding activity, with only 0.2% of luciferase activity recovered after 60 min. This lack of refolding capacity was similar to what we observed in lysates from MDA-MB-468 cells, which contain high levels of epichaperomes (Supplementary Fig. 13). These findings demonstrate that phosphorylation of HSP90 at key serine residues impairs its ability to refold denatured proteins, supporting the transition from a protein-folding chaperone to a scaffolding platform role within the epichaperome.

The impaired refolding activity observed with the EE mutant (and in the epichaperome-high MDA-MB-468 cells), despite the presence of endogenous HSP90, is highly intriguing. The phosphorylated HSP90 form (EE mutant) could act as a dominant negative, interfering with the normal function of endogenous HSP90. By sequestering co-chaperones or client proteins away from native HSP90, the phosphorylated form could effectively disrupt the entire chaperone system within the lysate. This disruption would reduce the overall folding capacity despite the presence of endogenous HSP90, a hypothesis that warrants future investigation.

We further established the dependency of epichaperome function, beyond its formation, on the phosphorylation of HSP90 serine residues (Figs. 7 and 8). A key characteristic shared among high epichaperome-expressing cells in PSC, CSC, and cancer cells is the hyperactivity of the transcription factor c-MYC[13,25–27]. In cancer, c-MYC is frequently overexpressed or mutated, resulting in sustained activation, which drives uncontrolled cell proliferation[59]. In ESCs, c-MYC plays a crucial role in maintaining pluripotency and self-renewal, crucial for preserving the undifferentiated state of ESCs[60]. We therefore investigated the impact of HSP90β Ser226$^P$/Ser255$^P$ on cellular behaviors such as self-renewal and proliferation.

To assess proliferation, ESCs were transfected with plasmids containing either the phosphomimetic (HSP90β$^{S226E,S255E}$) or non-phosphorylatable (HSP90β$^{S226A,S255A}$) mutant. Notably, ESCs transfected with the HSP90β phosphomimetic mutant displayed a significantly higher proliferative rate ($P < 0.0001$, >25%) compared to those transfected with the non-phosphorylatable variant, regardless of whether medium (1×) or high (2×) plasmid concentrations were employed (Fig. 7a). This observation lends support to the notion that HSP90β Ser226$^P$/Ser255$^P$, and consequently epichaperomes, play a crucial role in ESC proliferation.

Differentiation of ESCs results in a decreased proliferative rate, as indicated by the doubling time of ZHBTc4 ES cells (~12 h) and trophoblast-differentiated cells (~25 h)[32]. Since differentiation is also closely associated with the disassembly of epichaperomes, we next

examined the phosphorylation levels of HSP90β at Ser226 and Ser255 in cells with varying self-renewal capacities. We utilized the TET-repressible *oct4* mouse ESC line ZHBTc4, where the Oct4 expression is suppressed in the presence of doxycycline for ESC differentiation into trophoblast-like cells (Troph)[31]. In this experiment, we expressed WT mCherry-HSP90β in ZHBTc4 cells and quantified phosphopeptides in both ESCs and trophoblast cells following ESC differentiation (Fig. 7b and Supplementary Data 7). After normalizing the data to mCherry-HSP90β protein loading (middle panel, ES/Troph = 0.44), we observed a 30% higher phosphorylation of HSP90β at Ser255 in stem cells compared to differentiated cells (left panel, ES/Troph = 0.57). Phosphorylation levels of HSP90β at Ser226 appeared to remain unchanged under these experimental conditions after normalizing to protein loading (right panel, ES/Troph = 0.45).

Pluripotency hinges on crucial transcription factors like Oct4. Oct4 is widely recognized as one of the principal transcription factors governing the self-renewal of both pluripotent stem cells and cancer cells[61]. We find Oct4 interacts with epichaperomes in ESCs (Supplementary Data 1) and exhibits significant enrichment in the cargo captured with the Ser226/Ser255 phosphomimetic compared to the non-phosphorylatable HSP90 (Supplementary Data 6 and Fig. 7c, 1.4-fold EE: AA). To validate the reliance of Oct4 on epichaperomes, we examined Oct4 levels in both MDA-MB-468 cancer cells and HEK293 cells transfected with the various HSP90 plasmids. Additionally, we utilized affinity capture with PU-probes (Fig. 7d–f and Supplementary Fig. 14a). Notably, we observed that cells expressing the phosphomimetic EE mutant showed significantly elevated levels of Oct4, both overall (Fig. 7e) and within epichaperomes (i.e., those sequestered within the epichaperomes, Fig. 7f), compared to cells expressing the HSP90 AA mutant. No detectable differences were observed under control conditions (WT HSP90 and empty vector only) (Supplementary Fig. 12a). Additionally, Oct4 was sequestered by epichaperomes in MDA-MB-468 cells, supporting the idea that epichaperomes play a role in regulating pluripotency through both direct and indirect regulation of Oct4.

Epichaperomes play a pivotal role in supporting enhanced proliferation by altering the regulation of various proteins involved in cell signaling[3,13,19]. Higher epichaperome levels translate to a greater number of proteins being affected, resulting in increased signaling output[13,17,62]. We therefore next assessed the signaling output of cells transfected with the various HSP90 mutants. We observed a significantly heightened epichaperome-dependent impact on key signaling effector proteins involved in cell growth and proliferation (i.e., MEK, AKT, and mTOR) in cells expressing the HSP90 EE mutant compared to those expressing the AA mutant. This was evident in both the increased phosphorylation status of these effector proteins (Fig. 8a, b) and their enhanced recruitment to epichaperome platforms (Supplementary Fig. 14a–c) in cells expressing the EE mutant, as compared to those expressing the AA mutant. Importantly, these effects occurred without notable changes in the expression levels of the proteins (Supplementary Fig. 14a, b). No measurable differences were observed under control conditions (WT HSP90 and empty vector only) (Fig. 8b and Supplementary Fig. 14a, b).

Epichaperome formation fuels aggressive behaviors in cells[54,63]. Indeed, when observed under a microscope, we noted that, in comparison to cells expressing the non-phosphorylatable AA mutant (HSP90β$^{S226A,S255A}$), those expressing the phosphomimetic EE mutant (HSP90β$^{S226E,S255E}$) displayed a higher prevalence of cells with an elongated phenotype and several protrusions (Fig. 8c), supportive of a mesenchymal-like phenotype[64]. These morphological changes suggest a shift towards a more stem cell-like state, or a more aggressive phenotype in the context of cancer, in cells harboring the EE HSP90 mutant (i.e., with a high epichaperome load), a feature not observed in cells carrying the AA HSP90 mutant (i.e., not permissive of epichaperome formation).

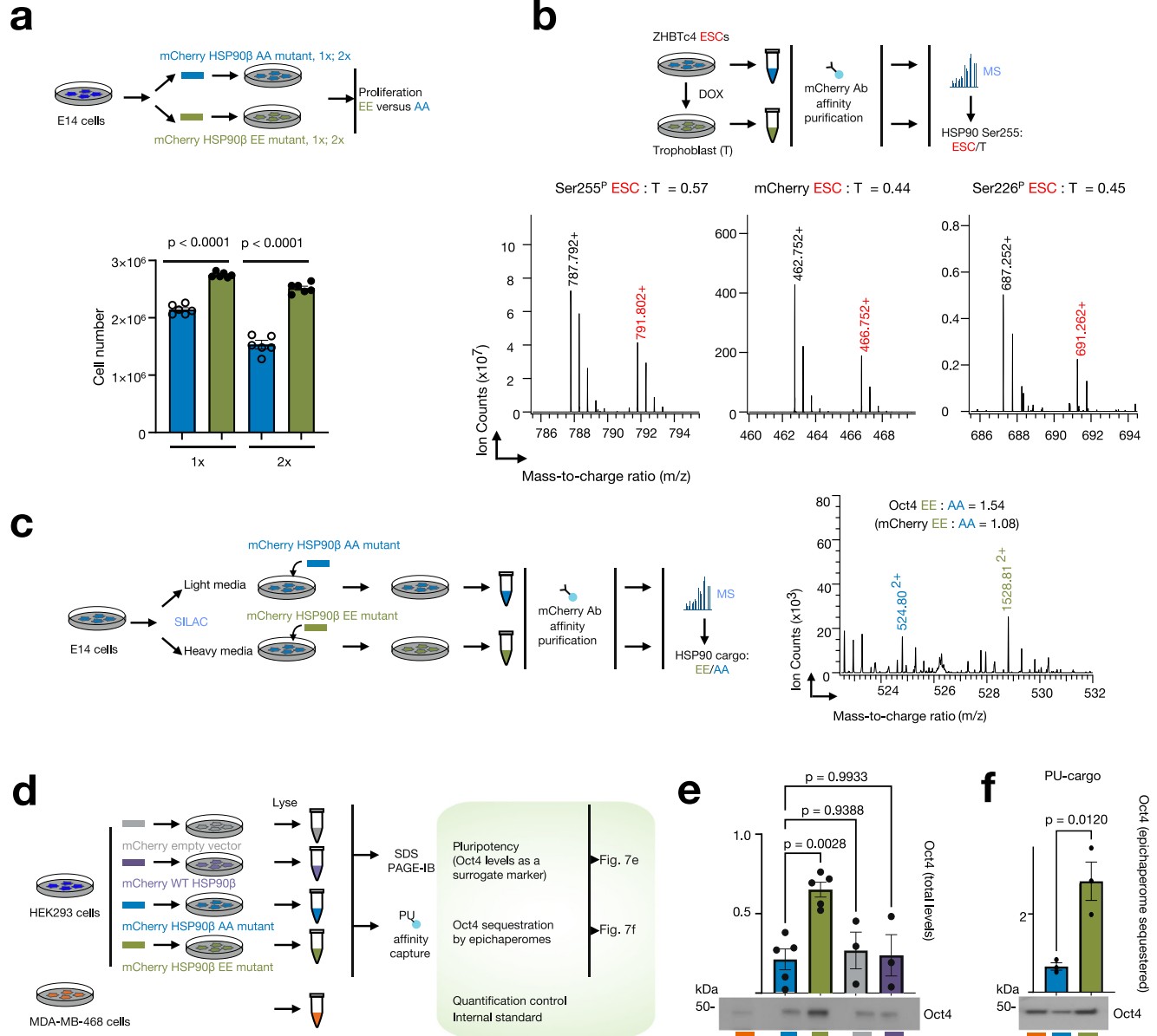

**Fig. 7 | Phosphorylation of key residues located in the HSP90 charged linker favors ESC proliferation and self-renewal potential. a** ESC proliferation at 60 h post-transfection in E14 cells transfected with either the phosphomimetic HSP90β$^{S226E,S255E}$ (EE) or the non-phosphorylatable HSP90$^{S226A,S255A}$ (AA) mutant. Medium (1×) or high (2×) plasmid concentrations were employed. Data are presented as mean ± s.e.m., $n = 6$, one-way ANOVA with Sidak's post-hoc, EE vs AA. **b** Representative spectra ($n = 3$ independent experiments) of phosphopeptides, S255P (left) and S226P (right), and a representative unmodified tryptic peptide (middle) in mCherry-tagged WT HSP90β affinity-purified from ESC or differentiated trophoblast (T) cells. **c** Representative spectra ($n = 3$ independent experiments) of a tryptic peptide from Oct4 protein co-purified from ESCs labeled with heavy or light isotope lysine and arginine expressing either the EE or the AA HSP90 mutant. Quantitative analysis via mass spectrometry (MS) to determine protein abundance is shown. **d** Overview of the experimental design and expected outcomes (for **e** and **f**). **e, f** Detection and quantification of Oct4 protein expressed in cells transfected with the indicated HSP90 mutants or vector control (**e**) and sequestered into the epichaperome platforms (identified through PU-beads capture, **f**). Data are presented as mean ± s.e.m., $n = 5$ AA, $n = 5$ EE, $n = 3$ WT, $n = 3$ empty vector, one-way ANOVA with Dunnett's post-hoc, EE vs AA, WT vs AA, empty vector vs AA (for **e**) and as mean ± s.e.m., $n = 3$, unpaired two-tailed $t$-test (for **f**). Source data are provided as a Source Data file and in Supplementary Data 6 and 7.

One intriguing question is which kinase could phosphorylate HSP90 at these serine residues? A likely candidate is casein kinase II (CK2)[65,66]. Ser226 fits well within CK2's phosphorylation consensus sequence and is a likely phosphorylation target, whereas Ser255 has a potential but weaker consensus match. CK2 is sequestered to epichaperomes in ESCs and in cancer cells[13]. Notably, CK2 is overexpressed in highly proliferative cells[67] and plays a role in phosphorylating numerous protein substrates involved in cell proliferation and survival[68]. Moreover, the mutation of CK2 has been shown to abolish the viability of both PSCs[69] and tumor cells[70,71], indicating a potential direct link

between epichaperome function and cellular physiology, possibly mediated by CK2 phosphorylation.

To investigate this potential link, we explored the impact of two CK2 inhibitors, CX4945 and CIBG-300, on epichaperome formation (Fig. 9a, b). Treatment of MDA-MB-468 epichaperome-high cancer cells with these inhibitors resulted in a dose-dependent decrease in epichaperomes, as observed by native PAGE coupled with immunoblotting against epichaperome components such as HSP90α, HSP90β, HSC70, CDC37, HOP, and HSP110. This treatment also led to a similar decrease in the levels of HSP90 phosphorylated at Ser226.

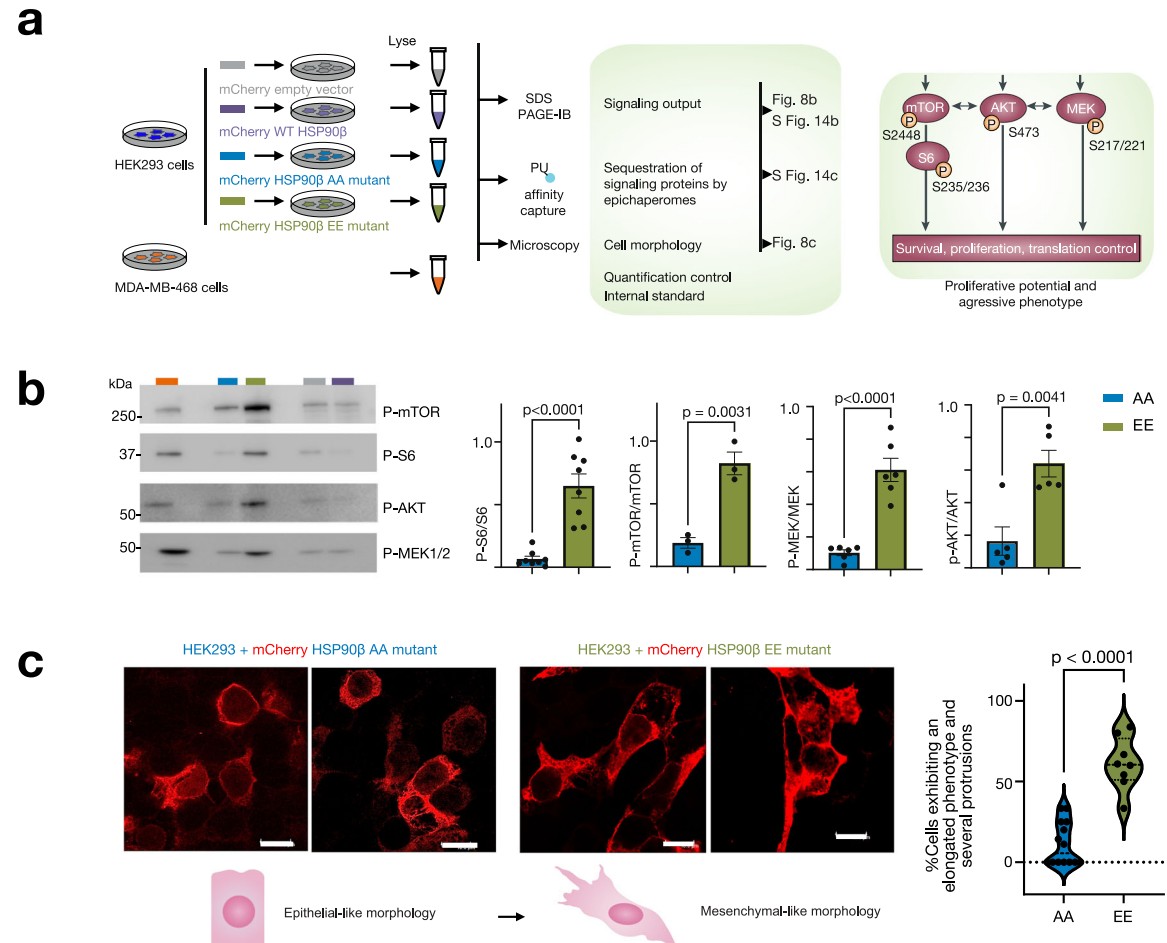

**Fig. 8 | Regulation of epichaperome processes in ESC and cancer cells hinges on the specific phosphorylation events occurring at key residues within HSP90's charged linker. a** Overview of the experimental design and expected outcomes. **b** Detection and quantification of proteins involved in transducing signaling events that lead to cell proliferation, survival, and protein synthesis control. See Supplementary Fig. 14 for total protein levels and levels sequestered into epichaperomes. Data are presented as mean ± s.e.m., p-S6 $n = 8$; p-mTOR $n = 3$; p-MEK1/2 $n = 6$; p-AKT $n = 5$, unpaired two-tailed $t$-test. **c** Confocal microscopy shows morphological differences between the cells transfected with either the AA or the EE HSP90 mutant. Micrographs are representative of 96 cells for EE and 62 cells for AA. Scale bar, 10 μm. Data are presented as mean ± s.e.m., $n = 8$ wells for EE, $n = 14$ wells for AA, unpaired two-tailed $t$-test. Source data are provided as a Source data file.

Importantly, no substantial change was observed in the total concentration of chaperones, indicating that the inhibitors specifically affected the phosphorylation state of HSP90, and in turn, epichaperome formation, rather than the expression levels of the epichaperome constituents. siRNA knockdown of CK2α—the catalytic subunit of CK2—recapitulated the effects on epichaperomes observed with CK2 inhibitors (Fig. 9c).

CK2 is a tetrameric enzyme consisting of two catalytic CK2α subunits and two regulatory CK2β subunits[72]. The α subunit serves as the catalytic unit responsible for substrate phosphorylation, while the β subunit acts as the regulatory unit, controlling the specificity and activity of CK2. Without the regulatory β subunit, the α subunit can randomly phosphorylate substrates, highlighting the importance of testing CK2 activity with both subunits present to accurately reflect its physiological role. Additionally, CK2 K68M α is known as the kinase-dead mutant, lacking catalytic activity[73]. Transfection with HA-tagged CK2α or a combination of HA-tagged CK2α and Myc-tagged CK2β resulted in an increase in epichaperome levels (Fig. 9d), suggesting that the presence of active CK2 enhances epichaperome assembly. Conversely, transfection with the HA-tagged kinase-dead mutant CK2 K68M α along with Myc-tagged CK2β led to a complete disruption of epichaperomes (Fig. 9d), underscoring the necessity of CK2's catalytic activity for maintaining epichaperome integrity. No effects were

observed in cells transfected with an empty vector, confirming the specific role of CK2 activity in modulating epichaperome levels.

These findings confirm the functional role of HSP90 phosphorylation at these specific serine residues in epichaperome formation and posit CK2 as a likely physiological candidate behind epichaperome formation (Fig. 9e). By demonstrating how CK2's phosphorylation activity directly influences the stability and assembly of epichaperomes, our study highlights a crucial regulatory mechanism in cellular proliferation and survival.

Previous studies have found that irrespective of the tumor type, 60–70% of tumors contain HSP90–HSC70 epichaperomes[13,19]. Additionally, epichaperomes are known to specifically form in diseased tissue[3]. To assess whether our observations regarding the impact of the HSP90 charged linker, derived from cell models, extend to human patients and are not artifacts specific to cultured cells, we obtained surgical specimens from breast and pancreatic cancer surgeries ($n = 18$ tissues from 9 patients, Fig. 10a–d). Both tumor ($n = 9$) and tumor-adjacent ($n = 9$) tissues, determined by gross pathological evaluation to be potentially non-cancerous, were analyzed for epichaperome levels using native PAGE. Additionally, total HSP90β and phosphorylated HSP90β at Ser226 were assessed by SDS–PAGE and immunoblotting with specific antibodies. To mitigate potential biases arising from varying HSP90 levels, each pair was normalized based on HSP90

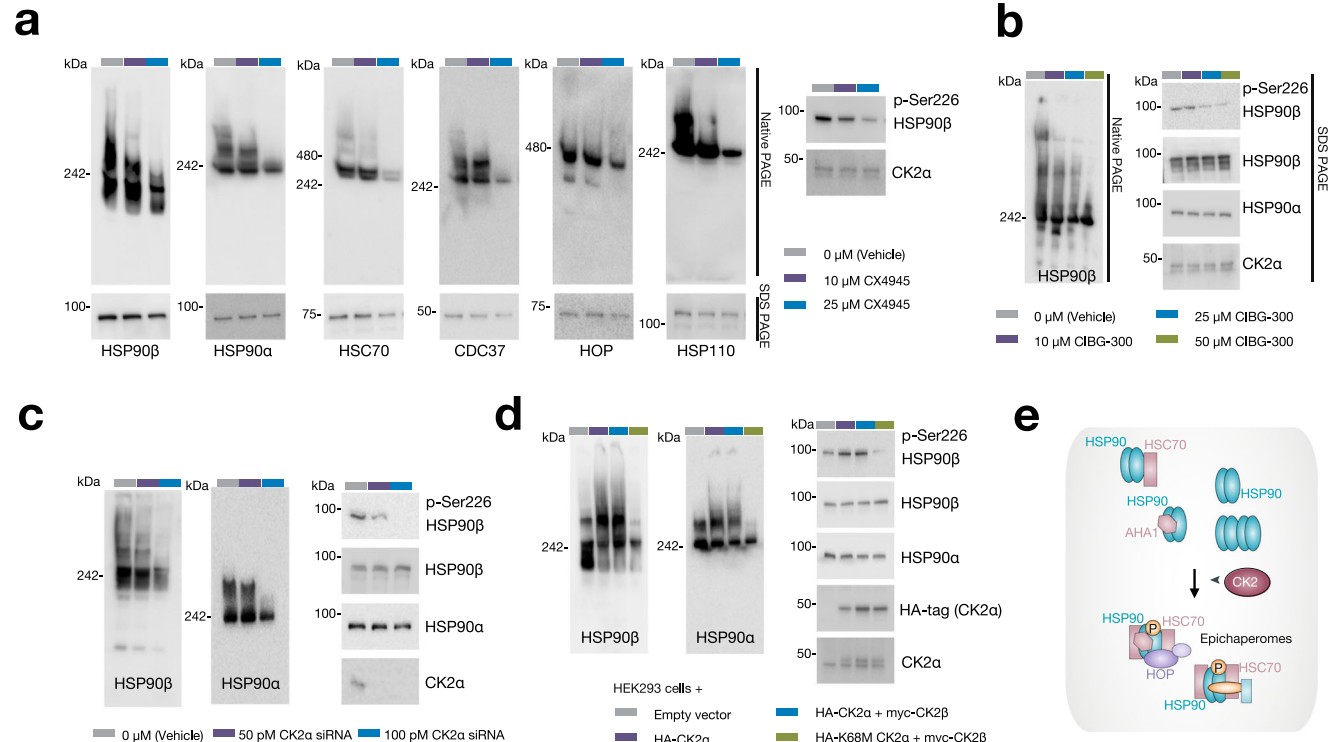

**Fig. 9 | Casein kinase 2 (CK2) is a physiologic regulator of epichaperome formation by modulating the phosphorylation of key residues within HSP90's charged linker. a**, **b** The gels illustrate the resulting epichaperome formation and the phosphorylation status of HSP90 in lysed MDA-MB-468 epichaperome-positive cancer cells treated with CK2 inhibitors. Detection of epichaperome components was done through SDS–PAGE (total protein levels) and native PAGE followed by immunoblotting. **a** The effect of CX4945 treatment, while **b** depicts the effect of CIBG-300 treatment. Vehicle-treated cells serve as controls. CK2α levels are shown to verify that inhibitor treatment effects are independent of changes in CK2α expression levels. **c** Same as in (**a**, **b**) for CK2α knockdown using dose-dependent

siRNAs in MDA-MB-468 cells. CK2α levels, knockdown efficiency control. **d** The indicated CK2 constructs were used to transfect HEK293 cells. CK2α, catalytic subunit; CK2β, regulatory subunit; kinase-dead mutant CK2 K68M α. HA tag and CK2α levels, transfection efficacy control. **a**–**d** Gel images are representative of three independent experiments. Source data are provided as a Source data file. **e** Schematic summary. CK2's phosphorylation activity directly influences the stability and assembly of epichaperomes. These findings confirm the functional role of HSP90 phosphorylation at these specific serine residues in epichaperome formation and posit CK2 as a likely physiological candidate behind epichaperome formation.

concentration. Despite challenges in obtaining high-quality epichaperome profiles from surgical samples, a robust correlation emerged between epichaperome expression and Ser226 phosphorylation (Fig. 10c, d). Tissues positive for epichaperomes exhibited p-Ser226 HSP90β positivity, and conversely, those negative for epichaperomes showed no or negligible p-Ser226 signal.

Collectively, these multifaceted biochemical and functional lines of evidence establish a compelling connection between structural features in HSP90 and the processes of epichaperome formation and function. These findings lend robust support to the hypothesis that the regulation of epichaperome processes in ESC and cancer cells—encompassing critical factors such as proliferative potential, self-renewal capacity, plasticity, and signaling output—crucially relies on the specific phosphorylation events taking place at key residues within HSP90's charged linker.

## Discussion

The intricate network of protein–chaperone interactions within cells plays a critical role in maintaining protein homeostasis and cellular function. In recent years, the discovery of epichaperomes as specialized chaperone assemblies in both cancer cells and pluripotent stem cells has opened new avenues for understanding chaperone biology. This investigation offers valuable insights into the structural and regulatory intricacies of epichaperomes, with particular attention to the pivotal role played by PTMs of HSP90 in orchestrating their formation and function.

A central discovery in this investigation is the recognition of specific PTMs on HSP90, especially at Ser226 and Ser255, as critical factors governing the assembly of epichaperomes. Our data reveal that phosphorylation of these serine residues enhances the association of HSP90 with other chaperones and co-chaperones, creating a microenvironment conducive to epichaperome formation. This PTM within the charged linker region reduces flexibility at critical interaction sites, such as those between HSP90, HSP70, and HOP, thereby enhancing the overall stability of the epichaperome assembly. Consequently, chaperones in epichaperome structures exhibit a more rigid and stable conformation with reduced variability, distinguishing them from the dynamic nature of typical chaperone complexes, where flexibility and rapid assembly/disassembly are essential for protein-folding activities. This finding underscores the significance of PTMs in regulating chaperone assemblies and highlights the potential of targeting these modifications for therapeutic intervention.

Chaperones appear to be highly susceptible to structural and functional regulation by a spectrum of PTMs. The concept of the "HSP90 PTM code" was introduced to highlight the nuanced regulation of HSP90 function by specific modifications, which transform its activity and interactions within the cell[12]. Understanding this code provides valuable insights into the mechanistic shifts that enable HSP90 to transition between different structures, assemblies and functions. For example, PTMs of HSP90 provide an important regulatory element, modulating co-chaperone and client protein binding[74–78], ATPase activity[79], conformational cycle[75,78–80], turnover[81],

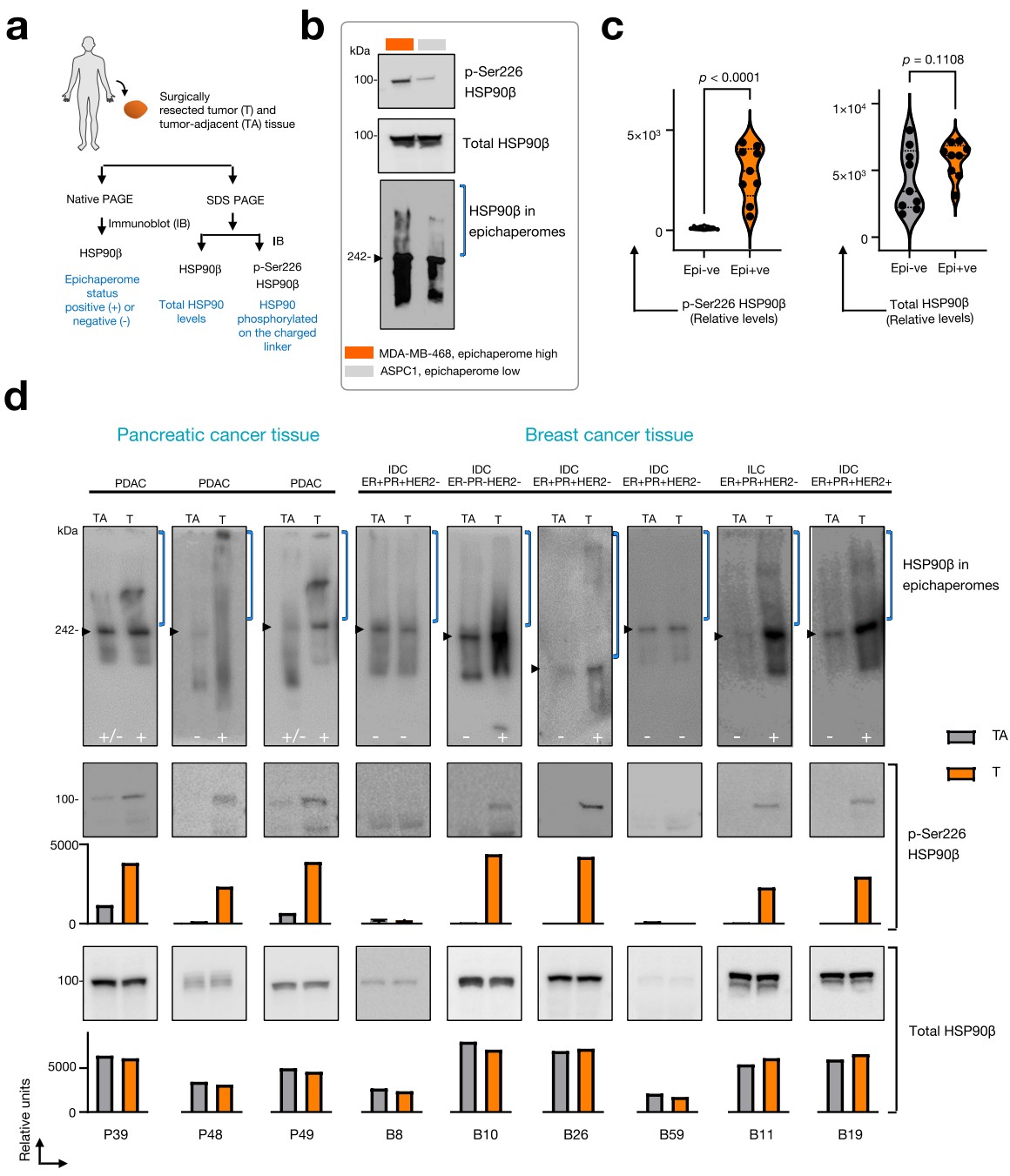

**Fig. 10 | Human tissues positive for epichaperomes exhibit p-Ser226 HSP90β positivity, and conversely, those negative for epichaperomes show no or negligible p-Ser226 signal within HSP90's charged linker. a** Cartoon illustrating the processing of human tissue for biochemical analyses. Both tumor (T) and tumor-adjacent (TA) tissues, determined by gross pathological evaluation to be potentially non-cancerous, were harvested and analyzed. **b** MDA-MB-468 breast cancer cells (epichaperome-high) and ASPC1 pancreatic cancer cells (epichaperome-low) served as controls for assessing p-Ser226 HSP90 levels. Gel images are representative of three independent experiments. **c** The graph presents the relationship between epichaperome positivity and HSP90 Ser226 phosphorylation for tissues described in (**a**). Data represent mean ± s.e.m., with $n = 9$ tumor (T) and $n = 9$ paired tumor-adjacent (TA) tissues classified based on epichaperome positivity or negativity, as determined by native PAGE (see **d**); unpaired two-tailed

*t*-test. **d** Detection of epichaperomes through native PAGE (top), and of p-Ser226 HSP90 (middle) and total HSP90 (bottom) by SDS–PAGE, followed by immunoblotting, in tissues from the indicated patient specimens, as in (**a**). Brackets indicate the approximate position of epichaperome-incorporated HSP90. Note: obtaining genuinely "normal" tissue adjacent to tumors presents challenges, especially in the case of pancreatic tissue. The relatively small size of the organ and the nature of surgical procedures for pancreatic cancer often lead to the collection of normal samples in close proximity to the tumor. It's crucial to acknowledge that, due to these challenges, we designate potentially normal tissue as tumor-adjacent tissue, recognizing that it may not entirely reflect a truly normal tissue state. PDAC pancreatic ductal adenocarcinoma, IDC invasive ductal carcinoma, ILC invasive lobular carcinoma, ER estrogen receptor, PR progesterone receptor. Source data are provided as a Source data file.

and small molecule affinity[12,39]. Similar to minor changes in primary sequence, these PTMs likely regulate the access to and occupancy of key conformational states of HSP90 for in vivo processing of some essential clients. Our investigation pinpoints crucial PTMs that remodel the functional profile of HSP90, metamorphosing it from a protein-folding entity into epichaperomes, a platform orchestrating the reorganization of PPI networks for heightened cellular adaptability and proliferation.

Our study uncovered a fascinating aspect of PTMs in HSP90 within epichaperomes—phosphorylation events occur in an IDR of the protein. The strategic placement of these PTMs in the IDR holds profound significance, suggesting that they influence HSP90's conformation and function beyond the traditional structured regions. This adaptability is crucial for HSP90's participation in distinct PPIs, allowing it to stabilize the epichaperome-enabling conformation and restructure the interactions of numerous proteins in response to cellular stressors. Intriguingly, previous studies in yeast[82], where the IDR was substituted with glycine–glycine–serine residues, align with our findings. These studies suggested that the charged linker (encompassing the IDR), influenced by the N-domain of HSP90, can adopt a structured form. This structured form, in turn, can stabilize interactions between specific HSP90 domains, influencing HSP90 dynamics, co-chaperone binding, and overall biological function, especially in conditions of cellular stress.

Changes in PPI networks play a fundamental role in cellular responses to stressors and the coordination of various biological processes[18]. These alterations, often induced by external stressors, are vital for the cell's ability to adapt and function under different conditions. Notably, <10% of human PPIs remain unaffected by stress-induced perturbations, highlighting the widespread impact of cellular stress on the interactome. These changes, influenced by factors such as PTMs and protein conformation, are essential for species-specific adaptation and contribute to PPI network malfunctions observed in diseases[18].

Our findings position CK2 as a physiological regulator of HSP90 phosphorylation that drives epichaperome formation. CK2 is a constitutively active kinase, and its overexpression has been reported in various diseases, including cancers, infectious diseases, neurological disorders, and cardiovascular conditions[83]. Future studies will be crucial in exploring whether the link between CK2 and epichaperome formation extends to these other diseases as well.

The implications of our study go beyond providing structural and mechanistic insights. We present compelling evidence that phosphorylation of HSP90 at Ser226 and Ser255 not only promotes epichaperome formation but also influences cellular behaviors, including proliferation and self-renewal. This suggests a direct link between epichaperome function and cellular physiology, particularly crucial in contexts such as cancer and pluripotent stem cell maintenance, where robust proliferation and adaptation are vital. The shared composition of epichaperome complexes between ESCs and cancer cells suggests a possible commonality in their functional roles. In both contexts, the epichaperome may facilitate rapid cellular proliferation and adaptability to environmental stress, characteristics crucial during development and tumorigenesis. This raises intriguing questions about whether the epichaperome contributes to the aberrant growth and survival of cancer cells by reactivating developmental pathways. The epichaperome might allow cancer cells to hijack developmental pathways typically active in ESCs, enabling them to maintain high proliferation rates and resist cell death.

Plasticity, a key characteristic associated with both ESCs and cancer cells[84], is also implicated in our findings. The morphological changes observed in cells expressing the phosphomimetic HSP90 mutant—specifically, the higher prevalence of cells with an elongated phenotype and several protrusions—hint at a mesenchymal-like phenotype[64]. This phenotypic shift is often associated with increased plasticity and is indicative of a more stem cell-like state. Our findings suggest a potential role for epichaperomes in modulating this dynamic process of cellular transition between different phenotypic states.

The link between pluripotency and cancer is particularly intriguing. Cellular stress is increasingly recognized as a pivotal factor that can shift the balance between cellular pluripotency and the development of malignancies. The process of dedifferentiation, observed in regeneration in plants and some vertebrates, involves the deactivation of genes responsible for cell-specific functions, re-entry into the cell cycle, proliferation, and activation of pluripotency-associated genes[85]. Tumors also undergo dedifferentiation, where cancer cells revert to a less differentiated state, re-express stem cell genes like Oct4, leading to the emergence of cancer stem-like cells with enhanced metastatic potential and treatment evasion[86]. Our study proposes epichaperomes as significant mediators of changes in cellular identity, partly through Oct4.

The revelation of HSP90's dysfunctional multimeric states carries implications for therapeutic interventions[3,16]. Instead of universally inhibiting all HSP90 pools, a paradigm shift comes to the fore with precision medicine strategies. The prospect of targeting specific pathologic conformations while preserving normal HSP90 functions emerges as a promising direction. This shift beckons researchers to navigate the intricate interplay of HSP90 conformations as they forge ahead in the quest for innovative therapeutic approaches. Our study also confirms the notion that small molecule HSP90 binders have distinct preference for HSP90 conformers in cells, reinforcing the finding that not all HSP90 inhibitors act equally well or equally selectively on specific disease-promoting HSP90 conformations or disease-associated HSP90 assemblies in comparison with HSP90 conformers found in normal cells. The first feature determines drug efficacy, whereas the latter influences the safety profile during administration.

In conclusion, our study unravels the intricate interplay between PTMs, conformational regulation, and biological functions of HSP90 within epichaperomes. These findings have implications for the development of novel therapeutic strategies targeting chaperone complexes in diseases characterized by epichaperome dysregulation, such as in cancers and neurodegenerative disorders. By deciphering the regulatory mechanisms underlying epichaperomes, we move one step closer to harnessing their potential for precision medicine and therapeutic intervention.

## Methods

### Human biospecimens research ethical regulation statement

This research complies with all relevant ethical regulations for research involving human participants. Surgical specimens were obtained in accordance with the guidelines and approval of the Institutional Review Board at Memorial Sloan Kettering Cancer Center, under the following approved protocols: Biospecimen Research Protocol# 09-121, project title: Ex-Vivo Testing of Breast Cancer Tumors for Sensitivity to Inhibitors of Heat Shock Proteins and Signaling Pathway Inhibitors, S. Modi, PI, and Biospecimen Research Protocol# 14-091, project title: Establishment and Characterization of Unique Mouse Models Using Patient-Derived Xenografts, E. de Stanchina, PI. The source of samples consisted of unused portions of surgical specimens taken for reasons other than research (i.e., for patients undergoing the procedures for medical reasons unrelated to need for research samples or to the nature of the research). No individuals were excluded on the basis of age, sex or ethnicity. Because breast cancer is a disease which overwhelmingly affects women, and is a disease that is generally not seen in children, the vast majority of breast cancer patients enrolled on protocol# 09-121 were females >18 years of age. In the case of pancreatic cancer samples ($n = 3$ patients), sex information was not available. Patient tissue samples were obtained with written informed consent and were de-identified prior to use in the studies. No compensation was provided for participation in this research.

## Reagents and chemical synthesis

All commercial chemicals and solvents were purchased from Sigma-Aldrich or Fisher Scientific and used without further purification. The identity and purity of each product was characterized by MS, HPLC, TLC, and NMR. Purity of target compounds has been determined to be >95% by LC/MS on a Waters Autopurification system with PDA, MicroMass ZQ and ELSD detector and a reversed phase column (Waters X-Bridge C18, $4.6 \times 150$ mm, $5 \, \mu m$) eluted with water/acetonitrile gradients, containing 0.1% TFA. Stock solutions of all inhibitors were prepared in molecular biology grade DMSO (Sigma-Aldrich) at $1000 \times$ concentrations. The PU-TCO, PU-CW800, and YK5-B probes and relevant control probes, and the PU-beads and the control probes were generated using published protocols[13,19,36,87–92] or as described in Supplementary Fig. 15 and Supplementary Note 1. The GA-biotin probe was purchased from Sigma (SML0985). DSS was acquired from Thermo Fisher (21655). CX4945 (Silmitasertib) was purchased from MedChemExpress (Cat No. HY-50855), and CIBG-300 was obtained from Sigma-Aldrich (Cat No. SML3143).

## Cell lines and culture conditions

Cell line selection was not based on gender, sex, or ethnicity. Cell lines were cultured according to the providers' recommended culture conditions. Cells were authenticated using short tandem repeat profiling and tested for mycoplasma. The MDA-MB-468 (female, breast cancer cell line, HTB-132, RRID: CVCL_0419), ASPC1 (female, pancreatic cancer cell line, CRL-1682, RRID: CVCL_0152), NCI-H1975 (female, non-small cell lung cancer cell line, CRL-5908, RRID: CVCL_1511), Daudi (male, B lymphoblast cell line, CCL-213, RRID: CVCL_0008), MRC5 (male, lung fibroblast cell line, CCL-171, RRID:CVCL_0440), CCD-18Co (female, colon fibroblast cell line, CRL-1459, RRID: CVCL_2379) and the Human Embryonic Kidney 293 (HEK293) cell line (CRL-1573, RRID: CVCL_0045), of female origin as determined by sequencing, were purchased from ATCC. IBL-1 (RRID:CVCL_9638) was derived from a male AIDS-related immunoblastic lymphoma patient[93]. Human mammary epithelial cells HMEC (PCS-600-010) isolated from adult female breast tissue were purchased from Lonza. B-cell lymphoma cell line OCI-LY1 (RRID:CVCL_1879), of male origin as determined by sequencing, was obtained from the Ontario Cancer Institute. E14 mouse ES cells[94] were received as frozen ampules from TG Fazzio (U Mass Med School). Cells were feed-free and verified as of male mouse origin through sequencing. ZHBTc4 mouse ES cells derived from a male mouse[31] were received from D. Levasseur (U of Iowa). Cells were cultured as ESCs without feeder cells in the absence of doxycycline. hiPSC were a gift from the Studer lab (MSKCC) and were derived from fibroblasts from a healthy male donor purchased from Coriell (#AG16146) and reprogrammed using CytoTune Sendai viruses[34].

## Mammalian cell culture and lysis

Mouse feeder-free ESCs (E14 or ZHBTc4 line) were grown on tissue culture plates coated with 0.2% gelatin. ESCs were cultured in Dulbecco's Modified Eagle Medium (DMEM; Gibco 10829018) media supplemented with 10% fetal bovine serum (FBS, HyClone SH30070.03HI), 2 mM L-glutamine, 0.1 mM nonessential amino acids (Gibco 11140050), $100 \, U \, mL^{-1}$ penicillin/streptomycin (Gibco 15140122), 0.1 mM beta-mercaptoethanol (Sigma M6250), and $103 \, U \, mL^{-1}$ LIF. Cells are grown in $37 \, °C/5\% \, CO_2$ incubator with media change every 2 days, passaged or harvested when 60–80% confluent. After harvesting, cell pellets are washed with phosphate-buffered saline (PBS, GenClone 25-508) and flash frozen before storing in $-80 \, °C$. For pull-down and chemical cross-linking experiments, frozen cells are thawed and lysed in Felts lysis buffer (20 mM HEPES pH 7.4, 50 mM KCl, 5 mM $MgCl_2$, 0.01% NP-40) in the presence of protease inhibitors, phosphatase and deacetylase inhibitors.

## ESC and hiPSC differentiation

ZHBTc4 cells were differentiated into trophoblasts through Oct4 repression. Cells were seeded at a density of $2 \times 10^5$ cells $mL^{-1}$ and grown in media with added doxycycline at a final concentration of $200 \, ng \, mL^{-1}$ for 96 h before harvest. E14 cells were spontaneously differentiated using attached EB culture. Briefly, cells were seeded at a density of $5 \times 10^4$ cells/mL in sterile bacteriological petri dishes in differentiation media (ES media without LIF) and cultured in $37 \, °C/5\% \, CO_2$ incubator for 4 days to aggregate into EBs. When turned orange, media were changed. On day 4, EBs were transferred into tissue culture dishes (without gelatin) at a density of 100–200 EBs per 10 cm tissue culture dish. Attached EBs were cultured in differentiation media in $37 \, °C/5\% \, CO_2$ incubator for 14–18 days before harvest. hiPSC differentiated in midbrain dopaminergic neurons were a gift from Dr. Lorenz Studer. Cells were differentiated into midbrain dopamine (mDA) neurons by a modified dual-SMAD inhibition protocol as described[20]. Briefly, hiPSCs were dissociated into single cells using Accutase and plated at high density on Matrigel (BD). The cells were subjected to timed exposure to LDN193189 (100 nM, Stemgent), SB431542 (10 $\mu M$, Tocris), SHH C25II (100 ng $mL^{-1}$, R&D), Purmorphamine (2 $\mu M$, Stemgent), FGF8 (100 ng $mL^{-1}$, R&D), and CHIR99021 (CHIR; 3 $\mu M$, Stemgent) to induce midbrain floor plate precursors. For mDA neuron induction, floor plate precursors were maintained in mDA differentiation media containing Neurobasal/B27/L-Glut (NB/B27; Invitrogen) supplemented with CHIR (until day 13) and with brain-derived neurotrophic factor (BDNF, 20n $mL^{-1}$; R&D), ascorbic acid (0.2 mM, Sigma), glial cell line-derived neurotrophic factor (GDNF, 20 ng $mL^{-1}$; R&D), transforming growth factor type $\beta 3$ (TGF$\beta$3, 1 ng $mL^{-1}$; R&D), dibutyryl cAMP (0.5 mM; Sigma), and DAPT (10 $\mu M$; Tocris). On day 20, cells were dissociated using Accutase and replated on dishes pre-coated with polyornithine (PO; 15 $\mu g \, mL^{-1}$)/laminin (1 $\mu g \, mL^{-1}$)/fibronectin (2 $\mu g \, mL^{-1}$) in differentiation medium (NB/B27 + BDNF, ascorbic acid, GDNF, dbcAMP, TGF$\beta$3, and DAPT). On day 30 of differentiation, cells were dissociated using Accutase and replated on dishes pre-coated with polyornithine (PO; 15 $\mu g \, mL^{-1}$)/laminin (1 $\mu g \, mL^{-1}$)/fibronectin (2 $\mu g \, mL^{-1}$) in differentiation medium (NB/B27 + BDNF, ascorbic acid, GDNF, dbcAMP, TGF$\beta$3, and DAPT) supplemented with 10 $\mu M$ Y-27632 (until day 32). Two days after plating, cells were treated with 1 $\mu g \, mL^{-1}$ mitomycin C (Tocris) for 1 h to kill any remaining proliferative contaminants. The mDA neurons were fed every 2–3 days and maintained without passaging until they were assayed at day 65. To prevent neurons from lifting off, laminin and fibronectin were supplemented into the media every 7–10 days.

## Cell culture and transfections

Monolayer cultures of MDA-MB-468 and HEK293 cells were grown in high glucose (4.5 g $L^{-1}$) DMEM containing 10% FBS and $1 \times$ antibiotic and antimycotic ($100 \times$ antibiotic-antimycotic (ABAM), GIBCO) in a $37 \, °C$ incubator supplied with 5% oxygen–air atmosphere. For native electrophoresis, and in-gel fluorescence studies, $1 \times 10^7$ cells were seeded in 100 mm dishes (Corning) at 70% confluency in DMEM supplemented with 10% FBS and $1 \times$ ABAM. Next day, spent medium was changed with fresh serum and antibiotic-free DMEM for 1 h before performing transfections. Cells were transfected using Lipofectamine 3000 (Invitrogen) with 4 $\mu g$ of mCherry empty vector, mCherry-HSP90$\beta$-Wild type (mCherry-HSP90$\beta$-WT), mCherry-HSP90$\beta$-S226A, S255A mutant (mCherry-HSP90$\beta$-AA) or mCherry-HSP90$\beta$-S226E, S255E mutant (mCherry-HSP90$\beta$-EE) plasmids. See Supplementary Tables 1–3 for plasmid sequences. Transfection mixtures were prepared in Opti-MEM (Gibco). Post 6 h of transfection, medium was changed with 10% FBS and $1 \times$ ABAM supplemented DMEM. Cells were harvested in native lysis buffer for future analyses.

## Primary specimen processing

Frozen tumor and matched tumor-adjacent tissues were cut into small pieces using surgical blades and weighed using a precision balance. Seventy-four milligrams of tissue was homogenized in 200 μL of 1× native lysis buffer in 1.5 mL microtube homogenizer for each sample. Homogenization was performed on dry ice. Post homogenization samples were incubated on ice for 30 min followed by centrifugation at $12,000 \times g$ at 4 °C for 15 min. Supernatant was collected, and protein quantification was done using BCA method. Samples were normalized using total HSP90β levels for each tissue pairs. An initial SDS–PAGE was run using 5 μg of total protein for each sample. Total protein loads were adjusted to ensure equal levels of total HSP90β in tumor and corresponding matched adjacent tissue. Samples were then processed for native PAGE and SDS–PAGE to check for HSP90β and p-Ser226 HSP90β as described below. A specific analysis of sex or gender was not conducted as part of this study because the primary focus was on the molecular mechanisms of epichaperome formation in disease states, irrespective of sex or gender. The sample set used did not include sufficient representation of different sexes to allow for meaningful sex-based comparisons. For breast cancer, the majority of samples were from female patients, while for the pancreatic cancer samples, sex data were unavailable.

## Native gel electrophoresis and western blot

Native gel electrophoresis was performed as reported[95]. Namely, $1 \times 10^7$ cells were lysed in 20 mM Tris pH 7.4, 20 mM KCl, 5 mM MgCl₂, 0.01% NP-40, and 10% glycerol buffer containing protease and phosphatase inhibitors (native lysis buffer), by a freeze–thaw procedure. Protein concentrations were measured by using the BCA assay according to the manufacturer's protocol (Pierce™ BCA Protein Assay Kit, Thermo Fisher Scientific, Waltham, MA). One hundred micrograms (100 μg) of protein were loaded in 4–10% native gel and run using native 1× Tris–Glycine buffer (25 mM Tris, 192 mM glycine) at 4 °C in a cold room at 125 V. Following electrophoresis, proteins were transferred to PVDF membrane, by wet transfer (25 mM Tris, 192 mM glycine, 20% (v/v) methanol, 0.02% SDS) at 100 V in the cold room. Membranes were then blocked for 1 h in 5% BSA in TBS/0.1% Tween 20. The blots were then probed with the following antibodies: HSP90β (SMC-107; RRID:AB_854214; 1:2000) and HSP110 (SPC-195; RRID:AB_2119373; 1:1000) from Stressmarq; HSC70 (SPA-815; RRID:AB_10617277; 1:1000), and HOP (SRA-1500; RRID:AB_10618972; 1:1000) from Enzo; HSP90α (ab2928; RRID:AB_303423; 1:6000), AHA1 (ab56721, RRID:AB_2273725, 1:1000) from Abcam; CDC37 (4793; RRID:AB_10695539; 1:1000), HOP (5670; RRID:AB_10828378; 1:1000), from Cell Signaling Technologies. The blots were washed with TBS/0.1% Tween 20 and incubated with appropriate HRP-conjugated secondary antibodies: goat anti-mouse (1030-05, RRID: AB_2619742, 1:5000), goat anti-rabbit (4010-05, RRID: AB_2632593, 1:5000), and goat anti-rat (3030-05, RRID: AB_2716837, 1:5000) (Southern Biotech, Birmingham, AL, USA). The chemiluminescent signal was detected with enhanced chemiluminescence (ECL) reagent according to manufacturer's instructions and visualized using ChemiDoc (Bio-Rad) and analyzed using Image Studio Lite Version 5.2. (LI-COR Biosciences). NativeMark unstained protein standard (Invitrogen, LC0725) was used to estimate the molecular weight of protein complexes in native gel electrophoresis and western blotting.

## SDS–PAGE and western blot

Proteins were extracted in 20 mM Tris pH 7.4, 20 mM KCl, 5 mM MgCl₂, 0.01% NP-40, and 10% glycerol buffer containing protease and phosphatase inhibitors (native lysis buffer), by a freeze–thaw procedure. Protein concentrations were measured by using the BCA assay according to the manufacturer's protocol (Pierce™ BCA Protein Assay Kit, Thermo Fisher Scientific, Waltham, MA). Ten to thirty micrograms (10–30 μg) of total protein were subjected to SDS–PAGE, transferred onto PVDF membrane, by wet transfer (Towbin buffer: 25 mM Tris, 192 mM glycine, 20% (v/v) methanol) at 100 V in cold room. Membranes were then blocked for 1 h in 5% BSA in TBS/0.1% Tween 20 and incubated overnight with the indicated antibodies. HSP90β (SMC-107; RRID:AB_854214; 1:2000) and HSP110 (SPC-195; RRID:AB_2119373; 1:1000) from Stressmarq; HSC70 (SPA-815; RRID:AB_10617277; 1:1000), HSP70 (ADI-SPA-810, RRID:AB_10616513, 1:2000) and HOP (SRA-1500; RRID:AB_10618972; 1:1000) from Enzo; HSP90α (ab2928; RRID:AB_303423; 1:6000), AHA1 (ab56721, RRID:AB_2273725, 1:1000) and anti-HA tag (ab9110, RRID:AB_307019; 1:1000) from Abcam; p-MEK1/2 (S217/221) (9154; RRID:AB_2138017; 1:1000), MEK1/2 (9122; RRID:AB_823567; 1:1000), p-mTOR (S2448) (5536; RRID:AB_10691552; 1:500), mTOR (2983; RRID:AB_2105622; 1:1000), CDC37 (4793; RRID:AB_10695539; 1:1000), HOP (5670; RRID:AB_10828378; 1:1000), p-S6 ribosomal protein (Ser235/236) (4858; RRID:AB_916156; 1:2000), S6 ribosomal protein (2217; RRID:AB_331355; 1:3000), Oct4 (2840, RRID:AB_2167691, 1:2000), p-AKT (S473) (9271, RRID:AB_329825, 1:2000), AKT (4691, RRID:AB_915783, 1:3000), CK2α (2656, RRID: AB_2236816, 1:2000) from Cell Signaling Technologies, β-actin (A1978, RRID: AB_476692, 1:3000) from Sigma-Aldrich, and mCherry (PA5-34974, RRID:AB_2552323, 1:2000) and p-Ser226 HSP90β (PA5-105480, RRID:AB_2816908, 1:1000) from Fisher Scientific. The blots were washed with TBS/0.1% Tween 20 and incubated with appropriate HRP-conjugated secondary antibodies: goat anti-mouse (1030-05, RRID: AB_2619742, 1;5000), goat anti-rabbit (4010-05, RRID: AB_2632593, 1:5000) and goat anti-rat (3030-05, RRID: AB_2716837, 1:5000) (Southern Biotech, Birmingham, AL, USA). The chemiluminescent signal was detected with ECL reagent according to manufacturer's instructions and visualized using ChemiDoc MP imaging system (Bio-Rad) and analyzed using Image Studio Lite Version 5.2. (LI-COR Biosciences). Thermo Scientific PageRuler Plus prestained protein ladder (Fisher Scientific, 26619) or Precision Plus protein standards (Bio-Rad, 161-0375) were used as size standards in protein electrophoresis and western blotting.

## Coomassie and Ponceau S staining

Where indicated, gels after native PAGE or SDS–PAGE were washed with deionized water three times for 5 min and incubated with Coomassie G-250 stain (Bio-Rad) for 1 h. The gels were washed with water after to remove the excess of the dye and imaged. Where indicated, membranes after protein transfer were incubated with Ponceau S solution (Sigma) for 10 min, then were washed with water to remove the excess of the dye and imaged.

## Primary specimen analyses

Specimens were harvested as previously reported[96]. Briefly, the surgical team delivered specimens in tightly sealed, sterile, leak-proof bags without fixatives. This maintained specimens in their fresh state, crucial for downstream analyses. Fresh specimens underwent sterile harvesting by the pathologist or assistant, using laminar flow hoods. Harvesting times were meticulously recorded, kept under 30 min postsurgery to mitigate cold ischemia effects. Primary breast tumor specimens were selectively obtained from the index lesion's periphery, avoiding central necrosis. Recognition criteria for necrotic tissue included color loss, softness, and demarcation from viable tissue. Normal breast tissue samples (e.g., normal dense/fibrous breast parenchyma) are taken from distant locations, at least 1 cm grossly away from the target lesion if feasible. In contrast, due to the relatively small size of the pancreas and the nature of surgical procedures, normal pancreas samples collected were typically in close proximity to the tumor. Whipple procedures typically involve the resection of the head of the pancreas, while distal procedures focus on the resection of the tail. Samples were initially stored in tubes with MEM and antibiotics and transported on wet ice to the laboratory immediately after procurement. Upon reaching the laboratory, samples were transferred to

cryovials, snap frozen, and stored at −80 °C for future molecular analyses.

## Chemical blotting

For in-gel blotting using PUTCO, cells were harvested in 20 mM Tris pH 7.4, 20 mM KCl, 5 mM MgCl₂, 0.01% NP-40, and 10% glycerol buffer containing protease and phosphatase inhibitors (native lysis buffer), by a freeze−thaw procedure. Protein concentrations were measured by using the BCA assay according to the manufacturer's protocol (Pierce™ BCA Protein Assay Kit, Thermo Fisher Scientific, Waltham, MA). One hundred micrograms (100 μg) of protein were incubated with 1 μM of PUTCO in a total volume of 42 μL. Post 3 h of incubation samples were loaded in 4−10% native gel and run using native 1× Tris−Glycine buffer at 4 °C in cold room at 125 V. Following electrophoresis, the gel was incubated in 30 mL of 700 nM Cy5-Tetrazine containing ice cold 1× Tris−Glycine buffer at room temperature (RT) for 15 min for the click reaction to occur. After 15 min, the gel was washed thrice (5 min each) with ice cold 1× Tris−Glycine buffer. The gel was then imaged using ChemiDoc MP imaging system (Bio-Rad). Alexa 546 channel (illumination: Epi-green, 520−545 nm excitation, Filter: 577−613 nm filter for green-excitable fluorophores and stains) was used to visualize mCherry-tagged species, and native page ladder (NativeMark™ Unstained Protein Standard, Cat. No. LC0725, Invitrogen™). The Cy5 channel (illumination: Epi-far red, 650−675 nm excitation, Filter: 700−730 nm filter for far red-excitable fluorophores and stains) was used for imaging PUTCO staining. Post capturing, the images from the two channels were merged to get the alignment of the bands with respect to the molecular weight ladder in Image Lab 6.1 (Bio-Rad). For in cell blotting using PU-CW800, E14 cells were plated at a seeding density of 1 × 10⁶/10 cm plate and grown for 44 h before treatment with either PU-CW800 or control fluorophore (C-CW800) at a concentration of 1 μM in culture media for 4 h while incubating at 37 °C, 5% CO₂. Following the treatment, cells were harvested and lysed by dounce homogenization in Felts lysis buffer (20 mM HEPES at pH 7.4, 50 mM KCl, 2 mM EDTA, and 0.01% NP-40) supplemented with protease, phosphatase, and deacetylase inhibitors. Cell lysates were buffer exchanged with fresh Felts lysis buffer containing supplements to remove any unbound drug before loading into a native gel. For visualization of PU-CW800 fluorescence and total protein, 200 μg of cell lysate was loaded onto a 4−10% native gradient gel and resolved at 4 °C for 5 h. Fluorescence was visualized on LI-COR Odyssey CLx using Image Studio™ Software (LI-COR Biosciences, v5.2) and then total protein was visualized on the same gel using Coomassie Brilliant Blue R250 stain. Band(s) with observable fluorescent signal were then processed by in-gel digestion and analyzed for LC−MS/MS to identify major proteins.

## SILAC and ESC transfection

For metabolic labeling with SILAC (stable-isotope labeling of amino acid in cell culture), ESCs were cultured and passaged five times at 48 h intervals in media containing SILAC DMEM (Thermo Fisher 88364) supplemented with 13C- and 15N-labeled heavy L-arginine (84 mg L⁻¹, Cambridge isotope CNLM-539-H) and L-lysine (146 mg L⁻¹, Cambridge isotope CNLM-291-H) or supplemented with 12C- and 14N-labeled light L-arginine (Fisher BP2505100) and L-lysine (Fisher J6222522) amino acids for five passages to ensure complete stable-isotope incorporation. For heterologous expression of HSP90 AA or EE mutants, cells were then reverse transfected with plasmid DNA using Lipofectamine™ 3000 Transfection Kit (Invitrogen #L3000015) and incubated at 37 °C, 5% CO₂ for 72 h at which point they were harvested.

## Measurement of cell proliferation

E14 cells were transfected and incubated in 37 °C/5% CO₂ incubator for 24 h. Cells were then replated to 6-well plate at the same dilution factor for each transfection treatment condition and then returned to the incubator. At 60 h post-transfection, cell proliferation was determined via cell count for all conditions.

## Casein kinase 2 (CK2) modulation

**CK2 inhibitor treatment.** MDA-MB-468 cells (1 × 10⁶) were treated with either 10, 25, or 50 μM of Silmitasertib (Cat No. HY-50855, MedChem-Express) or CIBG-300 (SML3143, Sigma-Aldrich), with DMSO serving as the control. After 24 h of incubation, cells were harvested and lysed in native lysis buffer supplemented with protease inhibitors (PI) and phosphatase inhibitor cocktail (PIC). The lysates were then processed for native and SDS−PAGE.

**siRNA treatment.** MDA-MB-468 cells were seeded in 6 cm dishes at a density of 1 × 10⁶ cells. The following day, the spent medium was replaced with fresh, serum-free, and antibiotic-free DMEM for 1 h before transfection. Cells were transfected using Lipofectamine 3000 (Invitrogen) with 50 or 100 picomoles of SignalSilence® CK2α siRNA I (CST Cat No. 6389S) for 24 h. After transfection, the medium was replaced with fresh DMEM supplemented with 10% FBS and 1× ABAM. After 72 h, cells were harvested and lysed in native lysis buffer containing PI and PIC, and samples were prepared for both native and SDS−PAGE assays.

**CK2 overexpression studies.** The following constructs were purchased from Addgene and purified using the QIAprep Spin Miniprep Kit (Cat No. 27104): pZW6, CK2α (ID: 27086), pZW12, CK2β (ID: 27088), pGV15, CK2α K68M (kinase-dead mutant, ID: 27089). HEK293 cells were cultured in high glucose (4.5 g L⁻¹) DMEM supplemented with 10% FBS and 1× ABAM (100XABAM, GIBCO) in a 37 °C incubator with a 5% CO₂ atmosphere. HEK293 cells were seeded in 100 mm dishes (Corning) at 70% confluency in DMEM supplemented with 10% FBS and 1× ABAM. The following day, the medium was replaced with fresh, serum-free, and antibiotic-free DMEM for 1 h before transfection. Cells were transfected using Lipofectamine 3000 (Invitrogen) with 5 μg CK2α, co-transfected with either 5 μg of CK2α + 5 μg of CK2β, or 5 μg CK2α K68M + 5 μg CK2β plasmids, following the manufacturer's protocol. Transfection mixtures were prepared in Opti-MEM (Gibco). After 6 h of transfection, the medium was replaced with DMEM supplemented with 10% FBS and 1× ABAM. Cells were harvested in native lysis buffer for downstream experiments.

## Luciferase refolding assay

Recombinant firefly luciferase (QuantiLum® Recombinant Luciferase, Promega, Cat No. E1701) was heat-denatured at 42 °C for 15 min in refolding buffer (25 mM HEPES/KOH pH 7.6, 100 mM KOAc, 10 mM Mg(OAc)₂, 2 mM ATP, 5 mM DTT). Refolding reaction mixtures were prepared by reconstituting 10 μM of heat-denatured luciferase with 50 μg of native extracts from MDA-MB-468, HEK293-WT, HEK293-AA, HEK293-EE, or CCD-18Co cells in refolding buffer, and then incubated at 30 °C. At the indicated time points, 1 μL of the sample was taken from each tube and added to 124 μL of assay buffer (100 mM K-phosphate buffer pH 7.6, 25 mM glycylglycine, 100 mM KOAc, 15 mM Mg(OAc)₂, 5 mM ATP), then mixed with 125 μL of 80 μM D-luciferin (Sigma-Aldrich, L6882). The final luciferase concentration for detection was 40 nM. Luminescence was measured for 10 s using a Perkin Elmer EnVision 2104 Multilabel Reader.

## Confocal microscopy

HEK293 cells transfected with mCherry-HSP90β-AA or mCherry-HSP90β-EE plasmids were seeded at a density of 1.8 × 10⁶ cells mL⁻¹ on coverslips in a monolayer in 6-well plates and then grown overnight for the cells to attach. Coverslips were mounted with ProLong™ Gold antifade mountant with DAPI. Imaging was done using Leica SP8 Stellaris microscope. Images were analyzed using Image J (version 1.54f) and Leica LAS X lite (version 2.6.0) software. Cell morphology

was manually inspected, and the percentage of cells exhibiting an elongated phenotype and several protrusions was calculated. Specifically, cells transfected with mCherry were assessed, and those displaying the described features were counted. The percentage was then determined based on the total number of mCherry-transfected cells observed.

## Chemical precipitation and cross-linking

The GA-affinity beads were prepared by incubating GA-biotin (Sigma SML0985) with Dynabeads M-280 Streptavidin (Thermo Fisher 11205D) at 4 °C for 2.5 h. The GA-bound beads were then incubated with cleared cell lysates or cross-linked cell lysates overnight at 4 °C. For PU-beads affinity capture, cell lysates were incubated with PU-beads or control beads at 4 °C for 3.5 h. Following incubation, bead conjugates were washed three times in lysis buffer before elution with sample buffer. The chemical cross-linking and HSP90 purification experiments were carried out in >3 replicates for both ligands. Samples were analyzed separately, and statistical significance was assessed.

## Chemical precipitation and immunoblotting

Cells were harvested in 20 mM Tris pH 7.4, 20 mM KCl, 5 mM MgCl$_2$, 0.01% NP-40, and 10% glycerol buffer containing protease and phosphatase inhibitors (native lysis buffer), by a freeze–thaw procedure. Protein concentrations were measured by using the BCA assay according to the manufacturer's protocol (Pierce™ BCA Protein Assay Kit, Thermo Fisher Scientific, Waltham, MA). PU-beads and control beads were washed with the native gel buffer three times prior use. Post washing, 40 μL aliquots of the beads were distributed into the sample tubes. Five hundred micrograms (500 μg) of total protein in 300 μL final volume, adjusted with native lysis buffer were added. Samples were incubated for 3 h at 4 °C on a rotor, followed by washing with native lysis buffer four times. Post washing, 30 μL of 5× Laemmli buffer was added to the beads and boiled at 95 °C for 5 min. Ten micrograms (10 μg) of the lysates (2%) was used as input for the pull-down experiment. Samples were then centrifuged at 13,000 × g for 20 min and supernatant collected was loaded on to SDS–PAGE. The protein transfer and western blotting procedures were performed as described in SDS–PAGE and western blot section.

## IUPred analysis for disorder prediction

**Sequence preprocessing.** The primary amino acid sequence of human HSP90β (P08238) and HSP90α (P07900) were extracted in FASTA format. These sequences served as the input for subsequent disorder prediction using the IUPred algorithm[97].

**Calculation of disorder scores.** The IUPred algorithm utilizes energy potentials derived from pairwise amino acid interactions to assess the local structural propensities of each residue in the protein sequence. For each residue, IUPred computes a disorder score within the range of 0–1. A score of 0 suggests a higher likelihood of being ordered, while a score of 1 indicates a higher likelihood of being disordered.

**Threshold for disorder classification.** To classify residues as either ordered or disordered, a threshold was applied to the calculated disorder scores. A common threshold of 0.5 was employed, designating residues with scores above 0.5 as disordered. The output of the IUPred analysis consisted of a disorder profile, providing disorder scores for each residue in the input protein sequence. Residues were categorized based on the applied threshold, facilitating the identification of regions with a high probability of disorder. All analyses were performed with the default parameters of the IUPred algorithm. The results presented here are based on the specific sequence input and the applied threshold for disorder classification.

## Computational analyses

**Protein complex preparation and docking calculations.** The structure comprising HSP90β–HSP70(2)–HOP proteins was developed using the molecular comparative modeling technique, employing Modeller v10.4, the Modeller Python script[98], and experimental template structures (PDB codes: 7KW7, 8EOB)[10,99]. The cryo-EM structure of human HSP90β (8EOB) served as the basis for obtaining coordinates for HSP90β (protomers A and B) in the developing model. To construct the assembly involving HSP70 and HOP, we utilized the sequences and atomic cryo-EM structure from the HSP90–HSP70–HOP–GR (7KW7) template. As these structures lacked certain residues, including those in the charged linker (Glu222–Lys273), we incorporated them as intrinsic loops during computational processing. The target sequence for each HSP90β protomer was extracted from UniProt ID: P08238. After model generation, we selected the optimal model based on the Discrete Optimized Protein Energy (DOPE) score. The final model included full-length HSP90 (excluding a ten-residue N-terminal disordered segment). For HOP and HSP70, we maintained the sequences provided in PDB:7KW7. The validated model, equipped with co-crystal ligands on each HSP90β protomer, was imported into Maestro v13.3 (Schrödinger LLC, 2022-3). Mutagenesis was performed to substitute Ser226/Ser255 with phosphomimetic conditions (Glu226/Glu255) and de-phosphorylated conditions (Ala226/Ala255) in both protomers of HSP90β. The preparation of all complexes utilized the Protein Preparation Wizard, a module for creating reliable, all-atom protein models. This involved restraining the assignment of bonds and bond orders, adding hydrogens, correcting formal charges, and filling missing side chains. Pre-processing steps included generating hetero states, H-bond assignment, and energy minimization using the optimized potentials for liquid simulations (OPLS3) force field, with a maximum root-mean-square deviation of 0.30 Å, employing the molecular mechanics engine Impact v9.6. Essential water atoms within 5 Å of the binding pocket were retained, while remaining waters were deleted. Structural refinement at neutral pH was carried out through the Epik v6.1 module[100]. The final refined structure served as the receptor for docking simulations. Ligands, such as ATP and ADP, underwent preparation with the LigPrep node, where the optimized ligand minimization algorithm yielded more conformers with numerous rotatable bonds, enhanced efficiency, and robustness. Different possible protonation states based on machine learning were generated, and ligand structures were minimized at pH values within the range of 7.0 and ±2.0, to guide the selection of protonation states on acidic/basic groups on ligands consistent with their pKa values, using the OPLS_3 force field, Premin, Truncated Newton Conjugate Gradient (TNCG), and Epik v6.1 nodes. Subsequently, a receptor grid was generated around the co-crystal ligand with default parameters. Docking experiments were executed on the nucleotide-binding pockets of both protomers using the XP (extra-precision) Glide program (Glide v9.6) and Prime-MMGBSA (molecular mechanics generalized born surface area) modules, respectively. The best poses in the resulting docked complexes served as the initial complex structure for MD simulations[101].

**Molecular dynamics simulations.** The pentameric assemblies were prepared in the following combinations: 2xHSP90(Ser226Ser255)-2xHSP70-HOP, 2xHSP90(Glu226Glu255)-2xHSP70-HOP-, 2xHSP90(Ala226Ala255)-2xHSP70-HOP, each bound to either ATP or ADP. For molecular dynamics simulations, the wild-type HSP90β (Ser226-Ser255) assembly bound to ATP was simulated for 100 nanoseconds (ns) with three independent replicas, resulting in a total simulation time of 300 ns. Similarly, the phosphomimetic mutant HSP90β (Glu226Glu255) assembly bound to ATP was also simulated for 100 ns with three independent replicas, providing a total simulation time of 300 ns. The non-phosphorylatable mutant HSP90β (Ala226Ala255)

assembly bound to ATP underwent the same simulation duration of 100 ns with three replicas, contributing to another 300 ns of total simulation time. For the simulations involving ADP-bound assemblies, the wild-type HSP90β (Ser226Ser255) assembly was simulated for 100 ns with three independent replicas, yielding a total simulation time of 300 ns. The phosphomimetic mutant HSP90β (Glu226Glu255) assembly bound to ADP was likewise simulated for 100 ns with three replicas, again leading to 300 ns of total simulation time. Lastly, the non-phosphorylatable mutant HSP90β (Ala226Ala255) assembly bound to ADP was simulated under the same conditions, resulting in another 300 ns of simulation time. In total, each assembly condition was run for 100 ns across three independent replicas, with a total simulation time of 300 ns for each condition (Supplementary Tables 4, 5). These complexes underwent molecular dynamics simulations using the Desmond v7.1 module of the MAESTRO Suite from Schrodinger (www.schrodinger.com). Before simulations, each assembly was built by embedding water molecules, adjusting temperature and pressure closer to the physiological environment through the OPLS3 force field and TIP4PEW water model. The system was neutralized with counter ions (Na$^+$/Cl$^-$) to balance the net charge in the simulation box. The particle mesh Ewald method[102] was used for electrostatics with a 10 Å cut-off for Lennard-Jones interactions, and the SHAKE algorithm[103] was applied to restrict the motion of all covalent bonds involving hydrogen atoms. The complex system underwent a six-step relaxation protocol before productive MD simulations. The solvated system was initially minimized with solute restraints and then without solute restraints, utilizing a hybrid method of steepest descent and the LBFGS (limited memory Broyden–Fletcher–Goldfarb–Shanno) algorithm[104,105]. The energy-minimized system underwent a brief 12 ps simulation within the NVT canonical ensemble at a temperature of 10 K, followed by a similar simulation in the isothermal-isobaric (NPT) ensemble at 10 K, with restraints on nonhydrogen solute atoms. Subsequently, the system was simulated for 24 ps in the NPT ensemble at 300 K with limited restraints on nonhydrogen solute atoms. In the final equilibration step, the system was simulated for 24 ps in the NPT ensemble at 300 K without constraints to reach an equilibrium state. The minimized and equilibrated system without restraints was then subjected to a 100 ns NPT simulation for production. The temperatures and pressures of the system in the initial simulations were controlled by Berendsen thermostats and barostats, respectively[104,105]. The relaxed system underwent productive simulations using the Nose'–Hoover thermostat at 300 K and the Martyna–Tobias–Klein barostat at 1.01325 bar pressure. Atomic-coordinate data for each receptor–ligand complex and system energies were recorded every 1000 ps. Residue-pair correlations were calculated along the MD trajectory using the script trj_essential_dynamics.py available in the Schrödinger suite. Additionally, the unexplored cryptic motions, distribution of secondary structural elements, and the array of protein folding in intrinsic disordered regions were thoroughly examined using the extracted meta-trajectory data from 1000 trajectories throughout the simulation period. The SSE index was computed to illustrate the percentage occurrence of alpha-helices (α) and beta-strands (β) during the simulation period, delineated by residue. The RMSF calculations were performed for the Cα atoms of each residue in the pentameric assemblies (HSP90(A)–HSP90(B)–HSP70(2)–HOP) in both ATP and ADP-bound states. These assemblies included WT, AA-mutant, and EE-mutant HSP90. The analysis was conducted using 1000 trajectory simulations loaded into the Simulation Interactions Diagram (SID) program within the Schrödinger software suite. Similarly, PCA was performed on the full assemblies, assessing three PC modes for each combination of WT, AA-mutant, and EE-mutant bound with ATP or ADP. The PCA was executed using a Python script available from the Schrödinger site. PCs that reflect dynamic (slow) global motions, derived from 1000 simulation frames, were utilized

to generate PCA plots. Potential and total energies were calculated using Desmond v7.7 in the Schrödinger software suite to assess the stability of the large assemblies with phosphorylated and non-phosphorylated mimic variants (mutant-EE, mutant-AA, and WT) in the presence of ATP and ADP.

## Immunoprecipitation of mCherry-HSP90
RFP Selector (NanoTag #N0410) resins were equilibrated with lysis buffer to prepare the resin. Cell lysates were then added and incubated with the resins at 4 °C with head-over-tail rotation for 90 min. Following incubation, resins were washed twice with lysis buffer and once with PBS before elution with 2 × sample buffer and incubated at 95 °C for 5 min. Eluents were then run on a 12.5% SDS–PAGE. For SILAC samples, heavy and light replicates ($n = 3$) were immunoprecipitated separately, then combined and separated by SDS gel electrophoresis.

## Chemical cross-linking
Cell lysates, with a concentration of ~3 µg µL$^{-1}$, underwent cross-linking using DSS (Thermo Fisher# 21655) at a concentration of 2.5 mM. This process occurred at RT for 1 h. To terminate the reaction, 0.8 M NH$_4$OH (Sigma# 09859) was added, reaching a final concentration of 25 mM, and incubated at RT for an additional 15 min. The lysates were clarified through two rounds of centrifugation at 16,200 × $g$ for 15 min at 4 °C before proceeding to separate HSP90 using immobilized PU-H71 or GA.

## SDS–PAGE and trypsin digestion
After elution from PU- or GA-beads, samples were loaded into 12.5% SDS–PAGE gel for separation. The entire lanes were cut into 10–15 bands and processed by in-gel digestion as described previously[19]. Briefly, gel bands were cut into small cubes, washed with 25 mM NH$_4$HCO$_3$/50% acetonitrile, reduced with 10 mM DTT (in 25 mM NH$_4$HCO$_3$) at 56 °C for 1 h, alkylated with 55 mM iodoacetamide (in 25 mM NH$_4$HCO$_3$) in darkness for 45 min. Gel pieces were washed again with 25 mM NH$_4$HCO$_3$/50% acetonitrile and evaporated in a speed-vac to complete dryness. The dried gel samples were proteolyzed using varied volumes of trypsin (0.6–1.0 µg depending on the intensity of the gel bands) at 37 °C for 4 h, before the extraction of tryptic peptides by 50% acetonitrile/2% acetic acid. Tryptic peptide mixture was concentrated down to ~7 µL before LC–MS/MS analysis. This experiment was done twice with similar results. For validation experiments in Fig. 3d, e, chemical precipitation and sample preparation for PTM analyses were performed as follows. For in-cell YK-B bait affinity purification, cells were plated in 10 cm plates at 6 × 10$^6$ cells per plate and treated with 50 µM YK5-B for 4 h. Cells were next collected and lysed in 20 mM Tris pH 7.4, 150 mM NaCl, and 1% NP-40 buffer. Five hundred micrograms (500 µg) of total protein were incubated with streptavidin agarose beads (Thermo Fisher Scientific) for 1 h and beads were washed with 20 mM Tris pH 7.4, 100 mM NaCl, and 0.1% NP-40 buffer (washing buffer). For in-lysate YK5-B bait affinity purification, cells were lysed in the above-mentioned lysis buffer. Streptavidin agarose beads were incubated with 50 µM YK5-biotin for 1 h, washed and added to 500 µg of total protein and incubated overnight. The beads were then washed with the washing buffer. For PU-H71 beads pull-down, 250 µg of the same protein lysates were incubated with 40 µL PU-H71 beads for 3 h and washed. Three different cell lines were used: MDA-MB-468 and OCI-Ly1 (epichaperome-positive cancer cell lines) and CCD-18Co (non-transformed but proliferating cell lines). For each cell line, three conditions were used: PU-beads incubated with lysates, YK5-B beads incubated with lysates, or YK5-B applied directly to cells in culture, resulting in a total of nine samples (three for PU and six for YK across three cell lines, $n = 9$). The nine samples were applied onto SDS–PAGE. Resulting gels were washed three times in distilled

deionized $H_2O$ for 15 min each and visualized by staining overnight with Simply Blue Coomassie stain (Thermo Fisher Scientific). Stained protein gel regions were typically excised into six gel sections per gel lane, and completely destained as described[19]. In-gel digestion was performed overnight with MS-grade trypsin (Trypsin Gold, Mass spectrometry grade, Promega) at 5 ng mL$^{-1}$ in 50 mM $NH_4HCO_3$ digestion buffer and incubation at 37 °C. After acidification with 10% formic acid (final concentration of 0.5–1% formic acid), peptides were extracted with 5% formic acid/50% acetonitrile and resulting peptides were desalted using hand-packed, reversed phase Empore C18 Extraction Disks (3 M, Cat#3M2215), following an established method[106]. Each of the six sections per sample, per gel lane, was excised and separately digested in-gel, at the same time, using the same batch and amount of trypsin. The peptides from each of these gel sections were purified and analyzed by nano-LC−MS/MS separately. For CCD-18Co samples, each experimental condition was run in technical replicates, resulting in six total MS batches. For MDA-MB-468 and OCI-Ly1 (epichaperome-positive cancer cell lines), each sample was run once, resulting in six MS batches. Each MS batch involved six gel sections, yielding a total of 72 LC−MS/MS runs (36 for CCD-18Co and 36 for the epichaperome-positive cancer cells).

## LC−MS data acquisition, protein and phosphopeptide identification

Briefly, digestion mixtures were injected into a Dionex Ultimate 3000 RSLCnano UHPLC system (Thermo Fisher Scientific), and separated by a 75 µm × 25 cm PepMap RSLC column (100 Å, 2 µm) at a flow rate of ~450 nL min$^{-1}$. The eluant was connected directly to a nanoelectrospray ionization source of an LTQ Orbitrap XL mass spectrometer (Thermo Fisher Scientific). LC−MS data were acquired in a data-dependent acquisition (DDA) mode, cycling between a MS scan (m/z 315−2000) acquired in the Orbitrap, followed by low-energy collision-induced dissociation (CID) analysis on three most intense multiply charged precursors acquired in the linear ion trap. The centroided peak lists of the CID spectra were generated using PAVA and searched against the Swiss-Prot protein database (version 2021.06.18; 17,089/565,254 entries searched for *Mus Musculus*), using Batch-Tag, a program of the University of California, San Francisco (UCSF) Protein Prospector software, version 6.5.2. For identification of proteins in pull-down experiments, a precursor mass tolerance of 15 ppm and a fragment mass tolerance of 0.5 Da were used for protein database searches (trypsin as enzyme; 1 miscleavage; carbamidomethyl [C] as constant modification; acetyl [protein N-term], acetyl+oxidation [protein N-term M], Gln->pyro-Glu [N-term Q], Met-loss [protein N-term], Met-loss + acetyl [protein N-term], oxidation [M] as variable modifications). Protein hits were reported with a Protein Prospector protein score ≥ 22, a protein discriminant score ≥ 0.0, and a peptide expectation value ≤ 0.01[107]. This set of thresholds of protein identification parameters does not return any substantial false positive protein hits from the randomized half of the concatenated database. We used a label-free, spectral counting-based quantitation strategy to estimate the abundance relationship between identified proteins[108,109]. Protein abundance index values were determined as a ratio of the number of detected peptides ($N_{obsd}$) and observable peptides ($N_{obsbl}$)[108]. The number of observable peptides were calculated using MS-Digest module of the UCSF Protein Prospector version 6.5.2 in the Mr range 700−2800, trypsin as enzyme, and 0 miscleavage allowed. After protein identification, PTM search was carried out with S/T/Y phosphorylation included in variable modifications among the identified proteins. A threshold of SLIP score >6 was imposed for false phosphorylation site assignment <5%[110]. Identified phosphopeptides were manually inspected by confirming the quality of MS/MS spectra and mass accuracy. Cross-linked peptides were identified using an integrated module in Protein Prospector, based on a

bioinformation strategy developed in the UCSF Mass Spectrometry Facility[42,43,111,112]. Key cross-linked peptides were identified and confirmed by manually examining the returned spectrum, peptide scores (>20), FDR (<5%), mass accuracy (<10 ppm), and absence from uncross-linked samples. For validation experiments in Fig. 3e, MS data acquisition and processing were performed as follows. Desalted peptides were concentrated to a very small droplet by vacuum centrifugation and reconstituted in 10 mL 0.1% formic acid in $H_2O$. Approximately 90% of the peptides were analyzed by nano-LC−MS/MS. A Q Exactive HF mass spectrometer was coupled directly to an EASY-nLC 1000 (Thermo Fisher Scientific) equipped with a self-packed 75 mm × 18 cm reverse phase column (ReproSil-Pur C18, 3M, Dr. Maisch GmbH, Germany) for peptide separation. Analytical column temperature was maintained at 50 °C by a column oven (Sonation GmBH, Germany). Peptides were eluted with a 3–40% acetonitrile gradient over 60 min at a flow rate of 250 nL min$^{-1}$. The mass spectrometer was operated in DDA mode with survey scans acquired at a resolution of 120,000 (at m/z 200) over a scan range of 300−1750 m/z. Up to 15 of the most abundant precursors from the survey scan were selected with an isolation window of 1.6 Th for fragmentation by higher-energy collisional dissociation with normalized collision energy of 27. The maximum injection time for the survey and MS/MS scans was 20 and 60 ms, respectively; the ion target value (Automatic Gain Control) for survey and MS/MS scan modes was set to $3e^6$ and $1e^6$, respectively.

## Quantitation of phosphopeptides and cross-linked peptides

Manually confirmed, high-confidence phosphopeptides and cross-linked peptides were quantified by the peak height of the extracted ion chromatogram of each peptide monoisotope mass. For phosphopeptide quantitation, the protein loading of HSP90 peptides in lysates or from pull-down experiments was normalize to a representative, isoform-specific tryptic peptide, ELISNSSDALDK for HSP90α and ELISNASDALDK for HSP90β. Phosphopeptides with different charge state or miscleavages were considered as different measurements for quantitation of each phosphosite. To assess the relative phosphorylation levels of different phosphosites in cancer cells and non-transformed cells, the ion intensity values of all phosphopeptides for each phosphosite were summed. The average ion intensities of each phosphosite between cancer and non-transformed cells were compared. Cross-linked peptides were identified using an integrated module in Protein Prospector, based on a bioinformation strategy developed in the UCSF Mass Spectrometry Facility[42,43,111,112]. Key cross-linked peptides were identified and confirmed by manually examining the returned spectrum, peptide scores, mass accuracy and absence from uncross-linked samples. Cross-linked peptides identified from various samples were pooled together, and the cross-linking propensity of each cross-linked peptide was assessed by its cross-linking percentage[44]. Cross-linking percentage for each peptide pair was calculated using the following formula:

$$\%XL = \frac{\text{cross} - \text{linked lpeptide lPeak lHeight (PH)}}{\sum \text{cross} - \text{linked lpeptide lPH} + \text{Dead} - \text{end lXL1PH} + \text{Dead} - \text{end lXL2PH}}$$

(1)

where the peak height is the apex peak height in LC−MS/MS runs. Dead-end XLs are cross-linker modified peptides where only one NHS-ester function of DSS is cross-linked to a Lys residue and the other NHS-ester function is hydrolyzed by water.

## Homology modeling

The mouse HSP90 sequences for both alpha and beta isoforms were aligned and the models were built using an open conformation template (PDB: 2IOQ), a closed conformation template (PDB: 2CG9), and

an HSP70-bound model (derived from a cryo-EM structure of HSP90·HSP70·GR complex[10] using UCSF Modeller (version 10.4)). Structural visualization and analysis were carried out using UCSF Chimera (version 1.18).

## Statistics and reproducibility

Unless as specified above under protein identification and bioinformatics analyses, statistics were performed, and graphs were generated, using Prism 10 software (GraphPad). Statistical significance was determined using Student's $t$-tests or ANOVA, as indicated. Means and standard errors were reported for all results unless otherwise specified. Effects achieving 95% confidence interval (i.e., $p < 0.05$) were interpreted as statistically significant. No statistical methods were used to pre-determine sample sizes, but these are similar to those generally employed in the field. No samples were excluded from any analysis unless explicitly stated.

## Reporting summary

Further information on research design is available in the Nature Portfolio Reporting Summary linked to this article.

## Data availability

The source data underlying all main and Supplementary Figs.—raw data, statistical analyses and uncropped gels—are provided with this paper as a Source Data file and were deposited in the Figshare repository under accession code 27075415[113]. Datasets and analytics associated with epichaperomics and proteomics analyses are available in the Supplementary Information as Supplementary Data 1 through 7 and were deposited in the Figshare repository under accession code 26662333[114]. LC–MS data (i.e., proteomics and epichaperomics raw mass spectrometry data, peak lists, and results) that support the findings of this study are deposited to the ProteomeXchange Consortium via the PRIDE partner repository with the dataset identifier PXD050251. Protein sequences (FASTA files) were obtained from UniProt (https://www.uniprot.org/). MD simulations data were deposited in Zenodo entry 10800912[115]. Source data are provided with this paper.

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

## Acknowledgements

This work was supported by the NIH (R01 CA172546, P01 CA186866, R56 AG061869, R01 HD09783, R01 AG067598, R01 AG074004, R01 AG072599, R56 AG072599, RF1 AG071805, P30 CA08748, P20 GM113131), NSF MRI 2320241, UNH Hamel Center (HTN), UNH Graduate School (G.Chiosis., F.C., and S.D.G.). L.B. acknowledges support from the National Science Foundation Graduate Research Fellowship Program. G.Colombo acknowledges funding from Fondazione AIRC (Associazione Italiana Ricerca Sul Cancro) under IG 2022—ID. 27139. S.S. would like to acknowledge funding support from BrightFocus Foundation (Award ID: A2022020F). We thank Dr. David A. Agard for providing the model of HSP90·HSP70·GR complex derived from a cryo-EM density map[116], Dr. Thomas G. Fazzio (U Mass Med School) for the E14 cells, Dr. Lorenz Studer for the human iPSCs and iPSC-derived neurons, and Dr. Dana Levasseur (U Iowa) for the ZHBTc4 cells. We thank the Molecular Cytology Core, the Antitumor Assessment Core, and our colleagues in the Departments of Surgery and Medicine at Memorial Sloan Kettering for providing the biospecimens for research.

## Author contributions

T.R. performed biochemical and functional studies in human cells and tissues. S.W.M. performed the MS studies and biochemical and functional studies in mouse ESCs. C.P. performed the MD simulations. H.T.N. and D.T.T. performed MS studies of cargos and cross-linking experiments. S.S. performed chemical synthesis, compound identity and purity evaluations for the epichaperome probes. L.B. and N.Y. generated ESC culture samples and MS sample preparation. A.R., P.P., S.J., S.C., S.B., and H.E.-B. performed experiments. C.S.D. provided reagents. V.M., C.K., J.L., P.Y., E.deS., A.C., S.M., and M.L.A. were involved in various aspects of biospecimen handling, including recruitment, procurement, or processing at different stages from surgery to delivery to the laboratory. R.J.C. and P.R.B. provided Protein Prospector and supported data analysis. F.C., T.A.N., G.Chiosis, and A.L.B. participated in the design and analysis of various experiments. H.E.-B., A.R., S.D.G., G.Colombo, and T.A.N. assisted with manuscript writing and data analysis. F.C. and G.Chiosis. developed the concept and wrote the paper.

## Competing interests

Memorial Sloan Kettering Cancer Center holds the intellectual rights to the epichaperome portfolio. G.Chiosis., A.R., and S.S. are inventors on the licensed intellectual property. All other authors declare no competing interests.

## Additional information

Tanaya Roychowdhury[1,14], Seth W. McNutt[2,14], Chiranjeevi Pasala [1,14], Hieu T. Nguyen[2], Daniel T. Thornton[2], Sahil Sharma [1], Luke Botticelli[2], Chander S. Digwal [1], Suhasini Joshi[1], Nan Yang[2], Palak Panchal[1], Souparna Chakrabarty [1], Sadik Bay [1], Vladimir Markov[3], Charlene Kwong[3], Jeanine Lisanti [3], Sun Young Chung [1], Stephen D. Ginsberg [4,5], Pengrong Yan[1], Elisa De Stanchina[3], Adriana Corben[6,12], Shanu Modi[7], Mary L. Alpaugh [1,13], Giorgio Colombo [8], Hediye Erdjument-Bromage [9], Thomas A. Neubert[9], Robert J. Chalkley [10], Peter R. Baker[10], Alma L. Burlingame [10], Anna Rodina [1], Gabriela Chiosis [1,7,15] ✉ & Feixia Chu [2,11,15] ✉

[1]Chemical Biology Program, Memorial Sloan Kettering Cancer Center, New York, NY, USA. [2]Department of Molecular, Cellular & Biomedical Sciences, University of New Hampshire, Durham, NH, USA. [3]Antitumor Assessment Core Facility, Memorial Sloan Kettering Cancer Center, New York, NY, USA. [4]Departments of Psychiatry, Neuroscience & Physiology & the NYU Neuroscience Institute, NYU Grossman School of Medicine, New York, NY, USA. [5]Center for Dementia Research, Nathan Kline Institute, Orangeburg, NY, USA. [6]Department of Pathology, Memorial Sloan Kettering Cancer Center, New York, NY, USA. [7]Department of Medicine, Division of Solid Tumors, Memorial Sloan Kettering Cancer Center, New York, NY, USA. [8]Department of Chemistry, University of Pavia, Pavia, Italy. [9]Department of Neuroscience and Physiology and Neuroscience Institute, NYU Grossman School of Medicine, New York, NY, USA. [10]Mass Spectrometry Facility, University of California, San Francisco, CA, USA. [11]Hubbard Center for Genome Studies, University of New Hampshire, Durham, NH, USA. [12]Present address: Maimonides Medical Center, Brooklyn, NY, USA. [13]Present address: Rowan University, Glassboro, NJ, USA. [14]These authors contributed equally: Tanaya Roychowdhury, Seth W. McNutt, Chiranjeevi Pasala. [15]These authors jointly supervised this work: Gabriela Chiosis, Feixia Chu. ✉e-mail: chiosisg@mskcc.org; feixia.chu@unh.edu

