## [Transparent Peer Review file · Nature Communications]

Phosphorylation-driven epichaperome assembly is a regulator of cellular adaptability and proliferation

Corresponding Author: Professor Gabriela Chiosis

Version 0:

Reviewer comments:

Reviewer #1

(Remarks to the Author)

In their study, McNutt et al. investigate the assembly of the epichaperome in both embryonic stem cells (ESCs) and cancer cells, focusing on the role of post-translational modifications (PTMs) of Hsp90 in the linker domain. They find that ESCs and cancer cells surprisingly share a similar epichaperome composition, especially in core components like Hsp70 and DNJC proteins. Through Mass spectrometry, they identify phosphorylation sites Ser231 and Ser263, that are phosphorylated in the closed conformation of Hsp90, which they induce by the PU-H71 inhibitor to isolate Hsp90 from ESCs.

Their molecular dynamics simulations then convincingly show that phosphorylation at these sites in the charged linker induces a conformational shift, which exposes the middle domain of Hsp90 and facilitates formation of a core epichaperome, that includes HSP70 proteins and the co-chaperone HOP. By comparing biochemistry and phosphorylation-mimetic Hsp90 mutants to non-phosphorylatable mutants, they reinforce the importance of these phosphorylation sites. Furthermore, they demonstrate that the regulation of epichaperome in ESCs and cancer cells depends on phosphorylation of Hsp90 Serine 226.

Finally, their data from pancreatic and breast cancer tissues underscore that Hsp90 epichaperome complexes are primarily phosphorylated at this site, further highlighting its significance in cellular disease states. Overall, this study significantly advances our understanding of the epichaperome, with Hsp90 phosphorylation on S226 playing a crucial role in this process. I have two comments to increase clarity of study. 1) The authors might want to elaborate more in the discussion on the significance of similar epichaperome complexes in ESCs, which could indicate a role of the epichaperome during development. 2) To further emphasize the crucial role of PTMs in the formation of epichaperome they might want to consider usage of the term "Hsp90 chaperone code".

Reviewer #2

(Remarks to the Author)

McNutt et al. use a combined experimental and computational study to investigate epichaperome networks and the role of Hsp90 in epichaperome formation. The authors show that epichaperomes from different cell lines share similar composition, containing chaperones like Hsp90, Hsp70, and cochaperones that form high molecular weight complexes. Using cross-linking combined with mass spectrometry, the authors assign conformations to Hsp90 within epichaperomes. Importantly, the authors show that the phosphorylation of sites within the unstructured charged linker of Hsp90 enhance the incorporation of Hsp90 into epichaperomes and result in increased presence of co-chaperones in epichaperomes. Computational modeling suggests that phosphorylation at S226 and S255 of Hsp90 induces a conformational change of the charged linker, flipping it into the "up" position and exposing the Hsp90 middle domain interface. The authors also show functional consequences of Hsp90 phosphorylation and how it enhances proliferation by altering the levels of signaling proteins involved in proliferation, like mTOR and AKT. Lastly, they analyzed tumors from human patients and showed that tissues positive for epichaperomes showed Hsp90 phosphorylation at the S226 site. This comprehensive study helps in understanding the physiological consequences of Hsp90's post-translational modifications and offers insight into the role of Hsp90 in facilitating the formation of epichaperomes. The study is of high technical quality and these findings will help researchers in targeting specific pathological conformations of Hsp90. However, some issues need to be addressed.

1. The authors identify 26 of the 42 major chaperones and co-chaperones that are known to localize to epichaperomes from studies with cancer cells. Were any new interacting proteins able to be identified in embryonic stem cells?
2. In the cross-linking mass spec experiments, it seems plausible that one might obtain both inter- and intra-Hsp90 crosslinks. How are these accounted for in the experiments/data and models shown in Figure 2?
3. The discussion of treating with PU-H71 on Page 4 of the manuscript suggests that this compound promotes disassembly of epichaperomes, while the experiments use PU-H71 to probe for the Hsp90 in epichaperomes and assign conformations in Figure 2. It is not apparent whether the assigned conformations of Hsp90 are those in the epichaperomes or the conformations that promote disassembly of epichaperomes.
4. The authors state that epichaperome assemblies are not active remodeling complexes but instead are inactive scaffolding complexes. Their cross-linking data assigns a conformation of Hsp90 in epichaperomes similar to that of the Hsp90 loading complex. Given this and the large molecular weight of epichaperomes, the authors speculate that the complex may resemble Hsp90 client loading complexes. However, it is important to distinguish the features that would cause the switch from productive folding to scaffolding complexes. Does phosphorylation of the two linker residues result in inhibition of client folding? Alternatively, could the cross-linking strategy be used in the complexes to identify whether the Hsp70 and HOP interactions are preserved in scaffolding vs client-loading complexes or whether epichaperome complexes have a similar stoichiometry but different arrangement?
5. In modeling the epichaperome assembly, the authors used the existing structure of an Hsp90(2)-Hsp70(2)-Hop multimeric assembly. The effects on protomer A linker are discussed in detail. How does the linker on protomer B behave? Given the asymmetry of Hsp70s in this complex (one nearly FL and the second with an NBD portion), I'm curious whether that has any effect on the dynamics of the system.
6. The findings from immunopurification showing a reduction in Hsc70 and HOP with the Hsp90 phosphomimetic seem to contradict the findings from the computational study suggesting stabilization of larger complexes with the phosphomimetic. One possibility is that Hsp90's conformational change could alter accessibility of chaperone/cochaperone binding sites on Hsp90 or that different binding sites are utilized relative to the client loading complex. The authors could examine their linker-docked Hsp90 structure and compare to other known binary Hsp90-chaperone complexes (Hsp82-Aha1, Hsp90-HOP, Hsp90-Hsp70) to determine whether the interacting region on Hsp90 may be obstructed by the presence of the linker. Conversely, it is challenging to draw strong conclusions regarding stability from correlations alone. Perhaps the amplitude of the fluctuations in the simulations can provide more insight, as the correlation is only one part of the bigger picture regarding dynamics. The authors should consider calculating the b-factors or RMSF, perturbations, or PCA, with the representative error, to provide more quantitative information (see General et al. Plos 2014, Meireles et. Al Prot Sci 2011).

Minor Points

- Fig 1c (right) lane 1 is not labeled – I assume this is B phycoerythrin treatment.
- HSP7C is identified as Hsc70, ST11P as HOP, and AHA1 as Aha1 in Figure 1. It might be worth noting it in the text since this notation also appears in subsequent figures.
- Arrows pointing to the location of the phosphorylated residues in Figure 4B (residue index axis) would be helpful.
- The number of simulations for each system is missing in the methods. Perhaps the authors could include a table to depict the number of replicas, simulation time, and ligands bound to each Hsp90 β WT and mutant for clarity.
- In the legend of Fig 5. Hsp90beta should be Hsp90 β
- Fig S6B is confusing with the residue index from 1-100, the sequence and gradient from green to red, and the highlight of S226A. Perhaps more explanations in the legend would improve the clarity.
- A description of the modeling studies should also be included in the last paragraph of the intro describing this study.

Reviewer #3

(Remarks to the Author)

The manuscript by Seth McNutt and the colleagues described a in-depth analysis of epichaperome assembly in various cancer and embryonic stem cell lines and tissues. The authors took advantage of specific chemical inhibitor that only recognizes either complexed HSP90 or unbound HSP90 and derivatized the chemical inhibitor with biotin affinity tag. Using these derivatized chemicals, the authors employed a chemical proteomics strategy by precipitating proteins that specifically bound to the chemicals for identification and PTM analysis. From this analysis, the authors identified specific components of the epichaperome complex and key phosphorylation sites on HSP90. By applying S/T to E or S/T to A mutations on the critical phosphosites that mimicked constant phosphorylation or no phosphorylation, the authors determined the functional of phosphorylation on mediating protein complex formation and the interaction to other proteins.

The overall study leveraged label-free quantitative protein analysis with functional validation strategies such as interactome assay and molecular dynamic simulation. The study presented a detailed view of the epichaperome assembly in solution and elucidated its potential interaction mechanism. A few major and minor concerns are listed below:

Major concerns:

1. Site-directed mutagenesis is an effective strategy to assess the functional role of phosphorylation but it can only represent either complete or none phosphorylation states, which may not fully represent physiological situations. Are the upstream enzymes of the phosphorylation events on HSP90 known? If so, it would be straightforward to analyze the phosphorylation-dependent change in protein interaction using either kinase overexpression or in vitro enzymatic reactions. Alternatively, phosphatase treatment is an efficient in vitro method. It would be interesting to see if removing phosphorylation by phosphatase after the complex has been pulled down can confirm the changes in the protein interaction compared to untreated control as we learned from the interactome analysis of the mutants.
2. Crosslinking is widely used to capture the transient protein-protein interaction in solution. In this study, crosslinking has been applied to the cell lysate prior to the pull-down with chemical inhibitor-attached resin. A potential concern is whether the crosslinking may affect key residues and binding of the complex to the chemical inhibitor, which will generate biased

results. An alternative strategy would be to pull down the complex first and then apply the crosslinking treatment on beads.

Minor concerns:

1. Fig 1d, a heat map was used to represent the numbers of peptides identified for each protein. But this representation does not take into account the size of protein. As larger proteins are likely to contribute more peptides, such representation is naturally biased against smaller proteins. Therefore, it would be better to use some sort of normalization method that considers the protein size information.

2. Fig 5a, the crosslinking matrix uses blue color to represent positive correlation and red color to represent negative correlation. As red color is more prominent, it may be best to use red color to represent positive correlation and blue color to represent negative correlation. In addition, what do the red and blue colors of the double arrows refer to in the cartoon?

Fig. 5b, why two different forms were depicted in either left or right complex formation? Is HSP90b AA showed similar movement as HSP90b WT?

It's a bit hard to interpret the correlation data in Fig 5 and how to determine phosphomimic mutants had more positive correlation than WT and none-phospho mutants.

3. Fig. 6c, since three replicate analyses were performed, it would be best to represent the SILAC interactome analysis of different mutant forms with volcano plots and highlight statistically significant changes.

4. Fig. 9b, when presenting the blots for the site-specific phosphorylation events, typically it would be necessary to show the blots for the corresponding proteins such as mTOR, S6, AKT etc, to indicate that the changes in phosphorylation was not due to the changes in overall protein abundance.

Reviewer #4

(Remarks to the Author)

I co-reviewed this manuscript with one of the reviewers who provided the listed reports.

Version 1:

Reviewer comments:

Reviewer #1

(Remarks to the Author)

In this revised version of the manuscript, the authors have addressed all comments of all reviewers. This has significantly improved the study and I have no further concerns.

Reviewer #2

(Remarks to the Author)

McNutt et al. have done an excellent job in addressing the reviewers' concerns in a point by point manner. The authors have put a significant amount of effort into revising their manuscript and carrying out additional experiments, including luciferase reactivation assays with the phosphomimetic mutants and CK2 phosphorylation assays. Additionally, the authors have performed further analysis of their simulations and added clarifying text throughout their manuscript. Together, these changes have further strengthened their findings. Overall, this is a very high impact research article that convincingly shows the structural details regarding epichaperome complexes and how PTMs of Hsp90 facilitate epichaperome formation. This study is of high technical quality and of broad interest to the Hsp90 and chaperone community.

Reviewer #3

(Remarks to the Author)

This is an excellent work. The authors have addressed the concerns raised by this reviewer. The manuscript is recommended for publication.

Reviewer #4

(Remarks to the Author)

REVIEWER COMMENTS AND POINT-BY-POINT RESPONSE

We sincerely thank the referees for their insightful suggestions and detailed review of the manuscript. We appreciate that all four reviewers have recognized the significance and technical quality of our work, highlighting its substantial contribution to advancing our understanding of the epichaperome and the pivotal role of HSP90 phosphorylation in this process. "Overall, this study significantly advances our understanding of the epichaperome, with Hsp90 phosphorylation on S226 playing a crucial role in this process." "This comprehensive study helps in understanding the physiological consequences of Hsp90's post-translational modifications and offers insight into the role of Hsp90 in facilitating the formation of epichaperomes. The study is of high technical quality and these findings will help researchers in targeting specific pathological conformations of Hsp90." "The overall study leveraged label-free quantitative protein analysis with functional validation strategies such as interactome assay and molecular dynamic simulation. The study presented a detailed view of the epichaperome assembly in solution and elucidated its potential interaction mechanism."

In response to the reviewers' critiques and comments, we have carefully addressed each point in the below point-by-point responses and have made the necessary revisions to the manuscript, which are highlighted in blue font.

Reviewer #1 (Remarks to the Author):

In their study, McNutt et al. investigate the assembly of the epichaperome in both embryonic stem cells (ESCs) and cancer cells, focusing on the role of post-translational modifications (PTMs) of Hsp90 in the linker domain. They find that ESCs and cancer cells surprisingly share a similar epichaperome composition, especially in core components like Hsp70 and DNJC proteins. Through Mass spectrometry, they identify phosphorylation sites Ser231 and Ser263, that are phosphorylated in the closed conformation of Hsp90, which they induce by the PU-H71 inhibitor to isolate Hsp90 from ESCs.

Their molecular dynamics simulations then convincingly show that phosphorylation at these sites in the charged linker induces a conformational shift, which exposes the middle domain of Hsp90 and facilitates formation of a core epichaperome, that includes HSP70 proteins and the co-chaperone HOP. By comparing biochemistry and phosphorylation-mimetic Hsp90 mutants to non-phosphorylatable mutants, they reinforce the importance of these phosphorylation sites. Furthermore, they demonstrate that the regulation of epichaperome in ESCs and cancer cells depends on phosphorylation of Hsp90 Serine 226.

Finally, their data from pancreatic and breast cancer tissues underscore that Hsp90 epichaperome complexes are primarily phosphorylated at this site, further highlighting its significance in cellular disease states. Overall, this study significantly advances our understanding of the epichaperome, with Hsp90 phosphorylation on S226 playing a crucial role in this process.

Response: Thank you for finding our paper of interest and for your detailed analysis. We appreciate your acknowledgment of the strengths and significance of our study.

I have two comments to increase clarity of study. 1) The authors might want to elaborate more in the discussion on the significance of similar epichaperome complexes in ESCs, which could indicate a role of the epichaperome during development. 2) To further emphasize the crucial role of PTMs in the formation of epichaperome they might want to consider usage of the term "Hsp90 chaperone code".

Response1.1-1.2: Thank you for suggesting these salient points.

1. We addressed this comment by adding to the Discussion the following text: "The shared composition of epichaperome complexes between ESCs and cancer cells suggests a possible commonality in their functional roles. In both contexts, the epichaperome may facilitate rapid cellular proliferation and adaptability to environmental stress, characteristics crucial during development and tumorigenesis. This raises intriguing questions about whether the epichaperome contributes to the aberrant growth and survival of cancer cells by reactivating developmental pathways. The epichaperome might allow cancer cells to hijack developmental pathways typically active in ESCs, enabling them to maintain high proliferation rates and resist cell death."

2. We appreciate your suggestion to use the term "Hsp90 chaperone code" to highlight the role of PTMs in epichaperome formation. We have incorporated this term in the manuscript - along with relevant references -

(see page 17) to emphasize the regulatory role of PTMs in the structural and functional regulation of HSP90, including in the formation and function of the epichaperome.

Reviewer #2 (Remarks to the Author):

McNutt et al. use a combined experimental and computational study to investigate epichaperome networks and the role of Hsp90 in epichaperome formation. The authors show that epichaperomes from different cell lines share similar composition, containing chaperones like Hsp90, Hsp70, and cochaperones that form high molecular weight complexes. Using cross-linking combined with mass spectrometry, the authors assign conformations to Hsp90 within epichaperones. Importantly, the authors show that the phosphorylation of sites within the unstructured charged linker of Hsp90 enhance the incorporation of Hsp90 into epichaperomes and result in increased presence of co-chaperones in epichaperomes. Computational modeling suggests that phosphorylation at S226 and S255 of Hsp90 induces a conformational change of the charged linker, flipping it into the “up” position and exposing the Hsp90 middle domain interface. The authors also show functional consequences of Hsp90 phosphorylation and how it enhances proliferation by altering the levels of signaling proteins involved in proliferation, like mTOR and AKT. Lastly, they analyzed tumors from human patients and showed that tissues positive for epichaperomes showed Hsp90 phosphorylation at the S226 site. This comprehensive study helps in understanding the physiological consequences of Hsp90’s post-translational modifications and offers insight into the role of Hsp90 in facilitating the formation of epichaperomes. The study is of high technical quality and these findings will help researchers in targeting specific pathological conformations of Hsp90.

Response: Thank you for finding our paper valuable and for your detailed review. We appreciate your recognition of the strengths of our study and the significance of our findings regarding the role of HSP90 in epichaperome formation. Below, we address your specific comments:

However, some issues need to be addressed.

1. The authors identify 26 of the 42 major chaperones and co-chaperones that are known to localize to epichaperomes from studies with cancer cells. Were any new interacting proteins able to be identified in embryonic stem cells?

Response2.1: Reviewer #2 is making a salient point. While the primary purpose of this manuscript is to identify key drivers in the assembly of the core epichaperome components, the identity of all components identified in ESCs is found in Supplementary Data 1. We added a paragraph to the text to clarify this point (see page 5). Other components are indeed context-dependent, but characterizing these context-dependent components and how they contribute to the specific functions of epichaperomes in ESCs is the subject of a paper in itself.

2. In the cross-linking mass spec experiments, it seems plausible that one might obtain both inter- and intra-Hsp90 crosslinks. How are these accounted for in the experiments/data and models shown in Figure 2?

Response2.2: Thank you for raising this point. Our experimental design was tailored to differentiate between intra- and inter-monomeric crosslinks in the cross-linking mass spectrometry experiments. After crosslinking the lysate and capturing HSP90 complexes with PU-beads, we performed SDS-PAGE to separate the proteins. We focused on the major ~80-90 kDa band, which corresponds to the HSP90 monomer. In addition, the crosslinked peptides identified were predominantly intra-monomeric, as they fit within the expected spatial constraints of the DSS cross-linker. The analysis of cross-linked lysine residues showed distances that are consistent with intra-monomeric links, supporting our conclusion that these are indeed within a single HSP90 monomer. We have added a clarifying sentence to the manuscript (see page 6) and Figure 2a schematic to explain our methodology and rationale, emphasizing how SDS-PAGE and the inherent constraints of DSS support the notion that our data primarily reflect intra-monomeric crosslinking.

3. The discussion of treating with PU-H71 on Page 4 of the manuscript suggests that this compound promotes disassembly of epichaperomes, while the experiments use PU-H71 to probe for the Hsp90 in epichaperomes and assign conformations in Figure 2. It is not apparent whether the assigned conformations of Hsp90 are those in the epichaperomes or the conformations that promote disassembly of epichaperomes.

Response2.3: Thank you for requesting further clarification on this point. In the experiments shown in Figure 2, and to ensure the capture of the epichaperome-enabling conformation, we first cross-linked cellular lysates using DSS before capturing HSP90 on the PU-beads. Given that PU has a preference for HSP90 in epichaperomes, as noted here and in prior publications, this approach ensures that the conformation 'frozen' by DSS and captured

by PU-beads is more characteristic of epichaperomes, rather than a conformation induced by PU to promote disassembly. We have added the following text to page 6 to clarify this point: “DSS crosslinking stabilizes the conformation of proteins by covalently linking residues that are in close proximity, effectively 'freezing' their relative positions within the protein complex. Given PU-H71's preference for binding HSP90 in its epichaperome conformation, any significant alteration of HSP90's structure by DSS would likely reduce PU-H71's binding affinity. Therefore, the structure captured by PU-beads is more likely to reflect the native HSP90 conformation found in epichaperomes rather than any altered state. By applying DSS before introducing PU-H71, the experimental setup increases the likelihood that the observed conformation is representative of the functional epichaperome, prior to any potential disassembly”.

4. The authors state that epichaperome assemblies are not active remodeling complexes but instead are inactive scaffolding complexes. Their cross-linking data assigns a conformation of Hsp90 in epichaperomes similar to that of the Hsp90 loading complex. Given this and the large molecular weight of epichaperomes, the authors speculate that the complex may resemble Hsp90 client loading complexes. However, it is important to distinguish the features that would cause the switch from productive folding to scaffolding complexes. Does phosphorylation of the two linker residues result in inhibition of client folding? Alternatively, could the cross-linking strategy be used in the complexes to identify whether the Hsp70 and HOP interactions are preserved in scaffolding vs client-loading complexes or whether epichaperome complexes have a similar stoichiometry but different arrangement?

Response2.4: Thank you for your insightful comment. We respectfully disagree with the statement that epichaperomes are inactive. While epichaperomes do not participate in the active folding of client proteins, they are highly active in other roles, as shown here and in our prior publications. The primary function of epichaperomes is to stabilize and organize proteins and protein complexes as scaffolding platforms, thereby promoting the rewiring of protein-protein interaction (PPI) networks. This scaffolding activity is crucial for various cellular processes, including signal transduction, stress response, and adaptation to changing environmental conditions. To address the reviewer's point about phosphorylation, we have conducted luciferase refolding assays, which demonstrate that the phosphomimetic mutants indeed impair folding activity. These findings support our conclusion that epichaperomes have a scaffolding rather than a folding role. The data are now included in Supplementary Figure 13 and associated text.

Regarding the distinction between scaffolding and client-loading complexes, our cross-linking data assigns a conformation of HSP90 in epichaperomes similar to that of the HSP90 loading complex. This suggests that while the architecture may resemble client-loading complexes, the phosphorylation of the two linker residues mediates a functional switch from productive folding to scaffolding, inhibiting client folding and facilitating the formation of scaffolding platforms. Both MS evidence and computational models support the conclusion that phosphorylation of the charged linker is a crucial contributor to epichaperome assembly, emphasizing its role in shaping not only HSP90 but also the stability and dynamics of the epichaperome structure. These findings highlight how phosphorylation modulates the structural dynamics and functional roles of HSP90 within the epichaperome assembly, promoting a scaffolding function through enhanced stability and reduced flexibility of the assembly at critical interaction sites. The text was revised on pages 10-12 and in Discussion to better highlight these findings.

The exact architecture and stoichiometry of epichaperome components in these platforms are the subject of our current studies and require sophisticated chemical biology, super-resolution microscopy, and structural biology techniques, which are beyond the scope of this paper. These ongoing investigations will provide a more detailed understanding of whether epichaperome complexes have similar stoichiometry but different arrangements compared to client-loading complexes.

5. In modeling the epichaperome assembly, the authors used the existing structure of an Hsp90(2)-Hsp70(2)-Hop multimeric assembly. The effects on protomer A linker are discussed in detail. How does the linker on protomer B behave? Given the asymmetry of Hsp70s in this complex (one nearly FL and the second with an NBD portion), I'm curious whether that has any effect on the dynamics of the system.

Response2.5: In our model, HSP90 protomer B is bound to HSP70(B) and HOP partners, and this binding configuration significantly influences the behavior of the linker on protomer B. The presence of HSP70B and HOP in the complex results in stabilizing intermolecular hydrogen-bond interactions between the charged linker

of HSP90 protomer B and HOP and HSP70B. These interactions lock the linker in a particular conformation, limiting its potential for secondary structural element formation and conformational rearrangement compared to protomer A. Supplementary Fig. 7, and associated text, were added in support. This asymmetry and stabilization through intermolecular interactions lead to distinct dynamic behaviors for the charged linkers of protomers A and B. While protomer A's linker remains more flexible and capable of rearranging into different conformations—modulated by its phosphorylation status—protomer B's linker is constrained by its interaction with HSP70 and HOP. Despite these distinct local impacts of phosphorylation on the structure and conformation of individual linkers, both linkers contribute to modulating the overall stability and dynamics of the pentameric assembly. This dynamic interplay highlights the complex regulatory mechanisms at play within the epichaperome, illustrating how each protomer's interaction with its partners can influence the system's overall behavior and stability.

6. The findings from immunopurification showing a reduction in Hsc70 and HOP with the Hsp90 phosphomimetic seem to contradict the findings from the computational study suggesting stabilization of larger complexes with the phosphomimetic. One possibility is that Hsp90's conformational change could alter accessibility of chaperone/cochaperone binding sites on Hsp90 or that different binding sites are utilized relative to the client loading complex. The authors could examine their linker-docked Hsp90 structure and compare to other known binary Hsp90-chaperone complexes (Hsp82-Aha1, Hsp90-HOP, Hsp90-Hsp70) to determine whether the interacting region on Hsp90 may be obstructed by the presence of the linker. Conversely, it is challenging to draw strong conclusions regarding stability from correlations alone. Perhaps the amplitude of the fluctuations in the simulations can provide more insight, as the correlation is only one part of the bigger picture regarding dynamics. The authors should consider calculating the b-factors or RMSF, perturbations, or PCA, with the representative error, to provide more quantitative information (see General et al. Plos 2014, Meireles et. Al Prot Sci 2011).

Response2.6: Thank you for your insightful comment. We respectfully disagree that the immunopurification showing a reduction in HSC70 and HOP with the HSP90 phosphomimetic contradicts the findings from the computational study suggesting stabilization of larger complexes with the phosphomimetic. This apparent contradiction only arises if we assume that epichaperomes are the sole form and assembly of HSP90 in the cells, which is not the case. While the phosphomimetic mutant (EE) increases the formation of epichaperomes, the non-phosphorylatable mutant (AA), which as we show in Figure 6c-e can incorporate into the endogenous non-tagged HSP90 assemblies, is potentially altering the cellular composition of folding chaperone assemblies. This alteration could lead to a higher prevalence of assemblies involving HSC70 and HOP distinct from epichaperomes. Since the immunopurification experiment reports on a ratio of EE to AA, as captured by the antibody, the AA component, which may contain the non-epichaperome HSP90-HSC70 or HSP90-HOP assemblies, will skew the ratio to make it appear that there is less HSC70 and HOP in the EE component (i.e., in the epichaperomes). In fact, this is not true, as the apparent reduction is due to the presence of other HSP90 assemblies that incorporate these chaperones. Therefore, the observed reduction in HSC70 and HOP in the phosphomimetic's immunopurification does not necessarily contradict the computational findings, but rather highlights the complexity of HSP90's interactions within the cell, where multiple forms and assemblies coexist, each with distinct roles and interactions. Text was added to provide further clarity on this matter (see page 12).

Regarding the second point, we agree, and we thank the reviewer for suggesting we perform these studies. Both the RMSF and PCA analyses strongly support the notion that the EE phosphomimetic mutant enhances epichaperome formation by stabilizing the assembly. The RMSF analysis shows reduced flexibility in key regions related to HSP70A and HSP70B binding, indicating a more rigid and stable structure that supports the maintenance of the epichaperome. Similarly, the PCA results demonstrate that the EE mutant has narrower spans along the principal components, reflecting reduced conformational variability and a more cohesive assembly. These findings align with our conclusion that phosphorylation at Ser226 and Ser255 promotes a structural environment conducive to epichaperome stabilization, thus highlighting the critical role of these post-translational modifications in enhancing epichaperome formation and function. These findings are included in Figure 5c,d, Supplementary Fig. 8 and pages 10-11.

Minor Points

- Fig 1c (right) lane 1 is not labeled – I assume this is B phycoerythrin treatment.

Response2.7: Label was added for clarity.

- HSP7C is identified as Hsc70, STI1P as HOP, and AHSA1 as Aha1 in Figure 1. It might be worth noting it in the text since this notation also appears in subsequent figures.

Response2.8: Clarification provided in the figure legend of Figure 1.

- Arrows pointing to the location of the phosphorylated residues in Figure 4B (residue index axis) would be helpful.

Response2.9: Arrows and additional labeling was added to Figure 4B.

- The number of simulations for each system is missing in the methods. Perhaps the authors could include a table to depict the number of replicas, simulation time, and ligands bound to each Hsp90 β WT and mutant for clarity.

Response2.10: For each system, we performed three independent 100-nanosecond (ns) simulations, each with 1000 frames, to ensure the reliability of our results. This strategy was employed to account for variability and enhance the statistical significance of our findings. By running these simulations three times, we aimed to minimize errors and confirm that our observations were consistent across different runs. Each 100 ns simulation was designed to provide a comprehensive view of the dynamics of the multimeric complexes. The decision to conduct multiple runs with adequate computational power allowed us to achieve robust and reliable results, which we found to be consistent across the three replicas. We acknowledge the reviewer's suggestion to include more details about our simulation strategy in the methods section. We added a table that specifies the number of replicas, simulation time, and ligands bound to each WT and mutant HSP90 containing assemblies. This additional information enhances clarity and transparency regarding our methodological approach.

- In the legend of Fig 5. Hsp90beta should be Hsp90 β

Response2.11: Edited as suggested.

- Fig S6B is confusing with the residue index from 1-100, the sequence and gradient from green to red, and the highlight of S226A. Perhaps more explanations in the legend would improve the clarity.

Response2.12: We agree with the reviewer's observation about Fig S6B. To improve clarity, we have reformatted the figure to conform with the format used in the main Figure 4. We hope these changes eliminate any confusion and enhance the interpretability of the figure.

- A description of the modeling studies should also be included in the last paragraph of the intro describing this study.

Response2.13: We added a paragraph to the Introduction, as suggested.

Reviewer #3 (Remarks to the Author):

The manuscript by Seth McNutt and the colleagues described a in-depth analysis of epichaperome assembly in various cancer and embryonic stem cell lines and tissues. The authors took advantage of specific chemical inhibitor that only recognizes either complexed HSP90 or unbound HSP90 and derivatized the chemical inhibitor with biotin affinity tag. Using these derivatized chemicals, the authors employed a chemical proteomics strategy by precipitating proteins that specifically bound to the chemicals for identification and PTM analysis. From this analysis, the authors identified specific components of the epichaperome complex and key phosphorylation sites on HSP90. By applying S/T to E or S/T to A mutations on the critical phosphosites that mimicked constant phosphorylation or no phosphorylation, the authors determined the functional of phosphorylation on mediating protein complex formation and the interaction to other proteins. The overall study leveraged label-free quantitative protein analysis with functional validation strategies such as interactome assay and molecular dynamic simulation. The study presented a detailed view of the epichaperome assembly in solution and elucidated its potential interaction mechanism.

Response: Thank you for your thoughtful evaluation of our work. Your feedback reinforces the significance of our findings in understanding the role of epichaperomes in cancer and embryonic stem cell biology, and the potential for developing targeted therapeutic strategies.

A few major and minor concerns are listed below:

Major concerns:

1. Site-directed mutagenesis is an effective strategy to assess the functional role of phosphorylation but it can only represent either complete or none phosphorylation states, which may not fully represent physiological situations. Are the upstream enzymes of the phosphorylation events on HSP90 known? If so, it would be straightforward to analyze the phosphorylation-dependent change in protein interaction using either kinase overexpression or in vitro enzymatic reactions. Alternatively, phosphatase treatment is an efficient in vitro method. It would be interesting to see if removing phosphorylation by phosphatase after the complex has been pulled down can confirm the changes in the protein interaction compared to untreated control as we learned from the interactome analysis of the mutants.

Response3.1: Thank you for your insightful comment. We agree that site-directed mutagenesis provides only partial insights into phosphorylation states. To address this comment, we investigated casein kinase II (CK2) as a potential upstream kinase for HSP90, particularly at Ser226, which fits well within CK2's phosphorylation consensus sequence. We conducted experiments using CK2 inhibitors, siRNA knockdown, overexpression, and a kinase-dead mutant to confirm the functional role of phosphorylation. These studies highlight CK2 as a physiological kinase that may phosphorylate HSP90, thereby inducing epichaperome formation. These data are incorporated into the new Figure 9a-e and associated text (pages 15,16).

2. Crosslinking is widely used to capture the transient protein-protein interaction in solution. In this study, crosslinking has been applied to the cell lysate prior to the pull-down with chemical inhibitor-attached resin. A potential concern is whether the crosslinking may affect key residues and binding of the complex to the chemical inhibitor, which will generate biased results. An alternative strategy would be to pull down the complex first and then apply the crosslinking treatment on beads.

Response3.2: Thank you for raising this important point regarding the use of crosslinking in our study. We acknowledge the potential concern that crosslinking could affect key residues and the binding of the complex to the chemical inhibitor-attached resin, potentially generating biased results. However, the DSS cross-linker primarily targets solvent-accessible surface lysine residues. This selectivity minimizes the likelihood of introducing extensive conformational changes or directly perturbing the pocket on HSP90 to which the inhibitor binds to. Applying crosslinking after the pull-down with the inhibitor-attached resin may also introduce limitations. The PU-H71 inhibitor itself can induce a conformational change in HSP90, and crosslinking at this stage might not reflect the physiological conformation of the protein complexes.

Furthermore, our molecular dynamics (MD) simulations consistently support the closed-like conformation detected by our crosslinking approach. This suggests that the conformation we captured accurately represents the physiological state. The convergence of experimental and computational data strengthens our confidence that our approach provides a reliable representation of the epichaperome complex in its native-like state. By combining several complementary approaches, we have ensured that our data faithfully reflect the interactions and conformations present in vivo, thereby alleviating concerns about potential bias introduced by crosslinking. Clarification was added to page 6.

Minor concerns:

1. Fig 1d, a heat map was used to represent the numbers of peptides identified for each protein. But this representation does not take into account the size of protein. As larger proteins are likely to contribute more peptides, such representation is naturally biased against smaller proteins. Therefore, it would be better to use some sort of normalization method that considers the protein size information.

Response3.3: Thank you for your valuable feedback regarding the heatmap in Figure 1d. We understand your concern about the potential bias against smaller proteins due to the lack of normalization for protein size. While our initial analysis indicated that this representation did not significantly affect our results, we appreciate the importance of accounting for protein size. To address this, we have included a revised heatmap, where normalization has been applied to account for protein size. We hope this provides a clearer representation of the data and addresses your concern.

2. Fig 5a, the crosslinking matrix uses blue color to represent positive correlation and red color to represent negative correlation. As red color is more prominent, it may be best to use red color to represent positive correlation and blue color to represent negative correlation. In addition, what do the red and blue colors of the double arrows refer to in the cartoon?

Response3.4: Thank you for your comment regarding the color scheme used in the crosslinking matrix in Figure 5a. While we understand the suggestion to use red for positive correlation and blue for negative correlation, the current color scheme (blue for positive correlation and red for negative correlation) is a widely accepted standard in scientific visualization, making it intuitive for most readers. Additionally, we have clarified in the figure legend that the blue arrows in the cartoon point to protein regions that move together correlatively, while the red arrows indicate regions that move oppositely, reflecting negative correlation. We hope this clarification helps convey the intended interpretation of the data.

Fig. 5b, why two different forms were depicted in either left or right complex formation? Is HSP90b AA showed similar movement as HSP90b WT?

It's a bit hard to interpret the correlation data in Fig 5 and how to determine phosphomimic mutants had more positive correlation than WT and none-phospho mutants.

Response3.5: Thank you for your comment regarding Figure 5b. The cartoon in Figure 5b illustrates the assemblies that are favored—either with or without HOP—when the charged linker serines on HSP90 β are phosphorylated (as in the EE mutant) or not phosphorylated (as in the WT, Ser/Ser).

The AA mutant does not mimic the WT in MD simulations because the substitution of serine with alanine leads to differences in interaction capabilities and structural dynamics. The substitution of serine with alanine is a widely used strategy in biological research to create non-phosphorylatable mutants. Alanine is chosen because its small, non-polar side chain is unlikely to introduce significant steric hindrance or alter the overall protein structure, making it a reasonable approximation for studying the functional role of phosphorylation without affecting the protein's overall architecture. However, despite its utility, the AA mutant cannot completely mimic the WT's natural serine behavior because it lacks the hydroxyl group necessary for phosphorylation and the associated hydrogen bonding interactions. These differences can affect local structural dynamics, potentially leading to changes in the protein's conformation and interaction patterns. Therefore, the AA mutant provides a non-phosphorylated baseline for comparison but may not capture all nuances of WT behavior under physiological conditions. By comparing WT, EE, and AA mutants, we gain comprehensive insights into how phosphorylation modulates HSP90 β dynamics and epichaperome assembly. The AA mutant is not included in the cartoon shown in Fig. 5b because it is not a physiological form. We have added clarification to the manuscript text (page 10) to explain that the AA mutant does not fully mimic the WT.

The data on correlative motion among the components of the pentameric assembly is encoded in the dynamic cross-correlation matrix. While the matrix is complex, it provides valuable insights into the coordinated movements of different protein regions. To simplify and convey this information effectively, we used the cartoon to distill the various motions among the proteins and their components, making the complex correlation data more accessible and easier to grasp. We added clarification to the figure and figure legends.

We hope this explanation, along with the revised figure, figure legend, and text, clarifies the rationale behind the representation in Figure 5b and aids in understanding the correlation data.

3. Fig. 6c, since three replicate analyses were performed, it would be best to represent the SILAC interactome analysis of different mutant forms with volcano plots and highlight statistically significant changes.

Response3.6: Added as suggested (see new Supplementary Fig. 10d).

4. Fig. 9b, when presenting the blots for the site-specific phosphorylation events, typically it would be necessary to show the blots for the corresponding proteins such as mTOR, S6, AKT etc, to indicate that the changes in phosphorylation was not due to the changes in overall protein abundance.

Response3.6: Thank you for your comment regarding the need to show blots for the corresponding proteins, such as mTOR, S6, and AKT, to indicate that changes in phosphorylation were not due to changes in overall protein abundance. We apologize if this was not clear in our initial submission, but these blots are included in Supplementary Figure 14a,b (was SFig. 9). We believe these supplementary figures provide the necessary information to support our findings. Thank you for bringing this to our attention.

Reviewer #4 (Remarks to the Author):

I co-reviewed this manuscript with one of the reviewers who provided the listed reports.

Response: Thank you for your thorough review and for collaborating with your colleague to provide valuable insights and feedback on our manuscript. We appreciate the time and effort you invested in evaluating our work, and your comments have been instrumental in enhancing the quality and clarity of our study.